# Finite Population Regression Adjustment and Non-asymptotic Guarantees for Treatment Effect Estimation

**Mehrdad Ghadiri**
MIT
mehrdadg@mit.edu

**David Arbour**
Adobe Research
arbour@adobe.com

**Tung Mai**
Adobe Research
tumai@adobe.com

**Cameron Musco**
UMass Amherst
cmusco@cs.umass.edu

**Anup B. Rao**
Adobe Research
anuprao@adobe.com

## Abstract

The design and analysis of randomized experiments is fundamental to many areas, from the physical and social sciences to industrial settings. *Regression adjustment* is a popular technique to reduce the variance of estimates obtained from experiments, by utilizing information contained in auxiliary covariates. While there is a large literature within the statistics community studying various approaches to regression adjustment and their asymptotic properties, little focus has been given to approaches in the finite population setting with non-asymptotic accuracy bounds. Further, prior work typically assumes that an entire population is exposed to an experiment, whereas practitioners often seek to minimize the number of subjects exposed to an experiment, for ethical and pragmatic reasons. In this work, we study the problems of estimating the sample mean, individual treatment effects, and average treatment effect with regression adjustment. We propose approaches that use techniques from randomized numerical linear algebra to sample a subset of the population on which to perform an experiment. We give non-asymptotic accuracy bounds for our methods and demonstrate that they compare favorably with prior approaches.

## 1 Introduction

Randomized experiments play a crucial role in estimating counterfactual outcomes. The strength of randomization lies in its ability to ensure independence between the treatment group and pre-treatment covariates [14, 21]. This independence, in turn, results in unbiased estimation of treatment effects through the comparison of sample averages. However, in finite sample settings, the use of simple randomization techniques result in high variance of the estimated quantities. There have been two broad approaches to address this issue by reducing the variance of the estimated effects when subject level covariates are available: (1) *design-based* approaches, which focus on the mechanism for assigning individuals to treatment and control groups in order to minimize imbalance (c.f. [21, 18, 27, 34, 5, 15]), and (2) *regression adjustment* based approaches, which correct for imbalances post hoc by incorporating a regression of the covariates on the outcome (c.f. [9, 24, 7, 13, 29]).

Much of the prior work examining variance reduction for experimentation has focused on asymptotic properties of approaches [23, 24, 13, 21, 23]. Recently, there has been work focusing on design-based variance reduction, which ties the treatment assignment problem to discrepancy minimization and gives finite sample error bounds for estimating causal estimands such as the average treatment effect (ATE) [18, 1, 5] and individual treatment effect (ITE) [1]. In particular, Harshaw et al. propose a

37th Conference on Neural Information Processing Systems (NeurIPS 2023).

design based on the *Gram-Schmidt random walk (GSW)* and give finite sample guarantees on the mean squared error for average treatment effect estimation when using this design with the classic Horvitz-Thompson estimator. Addanki, et al. [1] use GSW and leverage score sampling to provide algorithms and finite sample guarantees for both average and individual treatment effect estimation in the *partial observation* setting where only a subset of subjects are included in the experiment.

However, little work has examined the finite sample properties for experimental designs based on regression adjustment. Mou, et al. [28] provide non-asymptotic guarantees, but the analysis is in infinite population settings. The main contribution of this work is to analyze the finite population behavior of regression adjustment for variance reduction in treatment effect estimation, for ATE and ITE estimation in both the full and partial (sub-population) settings.

## 1.1 Problem Statement: Finite Population Treatment Effect Estimation

In the finite population treatment effect estimation problem, we have $n$ units (or individuals) associated with covariates $\mathbf{x}_1, \ldots, \mathbf{x}_n \in \mathbb{R}^d$ which are the rows of a covariate matrix $\mathbf{X} \in \mathbb{R}^{n \times d}$. Each unit $i$ is associated with two real numbers $y_i^{(1)}, y_i^{(0)}$ called the potential outcomes under the *treatment* and *control*, respectively. We denote the corresponding potential outcome vectors with $\mathbf{y}^{(1)}, \mathbf{y}^{(0)} \in \mathbb{R}^n$. The *Individual Treatment Effect (ITE)* vector $\mathbf{t} \in \mathbb{R}^n$ is defined as $\mathbf{t} := \mathbf{y}^{(1)} - \mathbf{y}^{(0)}$ and the *Average Treatment Effect (ATE)* $\tau$ is defined as the mean of this vector, i.e., $\tau := \frac{1}{n} \sum_{i=1}^n t_i$.

In treatment effect estimation, we seek to estimate either the ITE or ATE, under the constraint that at most one of the potential outcomes, either $y_i^{(1)}$ or $y_i^{(0)}$, can be observed for each unit $i$. E.g., that unit may be assigned to a treatment group in a controlled trial, and thus $y_i^{(1)}$ is observed while $y_i^{(0)}$ is not.

In *full observation* treatment effect estimation, we observe one potential outcome for each unit in the population [9, 13, 24, 18]. In *partial observation* treatment effect estimation, we further restrict ourselves to observing outcomes for just a small subsample of the population. This setting is important when the goal is to minimize experimental costs by e.g., limiting the size of a controlled trial [30, 20, 32, 1].

To reduce variance, many approaches to ATE and ITE estimation, including our own, attempt to leverage the population covariates $\mathbf{X}$ to make inferences about the treatment outcomes. In particular, throughout this work, we will target error bounds that depend on the best linear fit of the treatment outcomes to our covariates. Such linear effects models are common in the literature on treatment effect estimation [33, 18, 1] and provide a useful baseline for establishing theoretical bounds.

## 1.2 Our Contributions

We give new approaches to both ITE and ATE estimation in the full and partial observation settings that combine the classic statistical technique of regression adjustment with techniques from randomized numerical linear algebra. Our algorithms give natural, non-asymptotic error bounds depending on the best linear fit for either the ITE vector $\mathbf{t} := \mathbf{y}^{(1)} - \mathbf{y}^{(0)}$ or the sum of the potential outcome vectors $\boldsymbol{\mu} := \mathbf{y}^{(1)} + \mathbf{y}^{(0)}$ to our covariates. $\boldsymbol{\mu}$ is a natural quantity that arises e.g., in variance bounds for classic approaches to ATE estimation, such as the Hovitz-Thompson estimator [19].

### 1.2.1 Individual Treatment Effect Estimation

Our first result is for individual treatment effect estimation in the full and partial observation settings.

**Theorem 1** (ITE Estimation). *For any $\epsilon, \delta, \alpha \in (0, 1]$, there exists a randomized algorithm (Algorithm 3) that observes a potential outcome for $O(d \log(d/\delta)/\epsilon^2 + \alpha \cdot n)$ units and outputs a vector $\hat{\mathbf{t}}$. Moreover there is an event $\mathcal{E}$ such that $\Pr(\mathcal{E}) \geq 1 - \delta$ (over the randomness of the algorithm) and*

$$\mathbb{E}\left[\|\hat{\mathbf{t}} - \mathbf{t}\|_2^2 | \mathcal{E}\right] \leq (1 + \epsilon) \cdot \min_{\mathbf{b} \in \mathbb{R}^d} \|\mathbf{X}\mathbf{b} - \mathbf{t}\|_2^2 + \frac{d}{\alpha} \cdot (1 + \epsilon) \cdot \left\|\mathbf{y}^{(1)} + \mathbf{y}^{(0)}\right\|_\infty^2.$$

Theorem 1 shows that with high probability (at least $1 - \delta$), our algorithm achieves an expected error bound that nearly matches the best linear fit of the ITE vector to our covariates (i.e., $\min_{\mathbf{b} \in \mathbb{R}^d} \|\mathbf{X}\mathbf{b} - \mathbf{t}\|_2^2$), up to an additive term depending on the maximum magnitude of the sum of potential outcomes for any individual. In the full observation setting, when $\alpha = 1$, this additive

term is $O(d)$, assuming that the maximum magnitude does not scale with the population size $n$. In contrast, we expect the best fit error ($\min_{\mathbf{b} \in \mathbb{R}^d} \|\mathbf{X}\mathbf{b} - \mathbf{t}\|_2^2$) to grow as $\Theta(n)$, and thus the additive term is lower order. In the partial observation setting, as long as we set $\alpha = \omega(1/n)$, we still expect this additive term to be lower order. Thus, the case when $d \ll n$, we are able to nearly match the best fit error while only observing a very small subset of the full population.

We sketch the proof of Theorem 1 in Section 2. Our algorithm is based on *pseudo-outcome* regression, which is studied in the asymptotic setting by Kennedy et al. for conditional ATE estimation [22]. Roughly, in the full observation setting, we construct a vector $\mathbf{v}$ where $v_i = 2y_i^{(1)}$ with probability $1/2$ and $v_i = -2y_i^{(0)}$ with probability $1/2$. We can see that $\mathbb{E}[\mathbf{v}] = \mathbf{t}$ (i.e., $\mathbf{v}$ is equal to the ITE vector in expectation), and importantly, that constructing $\mathbf{v}$ only requires observing one potential outcome for each individual. By regressing $\mathbf{v}$ onto our covariates, we obtain our estimate $\widehat{\mathbf{t}}$ for $\mathbf{t}$.

In the partial observation setting, we further subsample individuals according to the *leverage scores* of the covariate matrix $\mathbf{X}$. This is a standard technique in randomized linear algebra [26, 10, 35], which allows us to approximately solve the above regression problem while only observing a subset of $\widetilde{O}(d/\epsilon^2)$ entries of $\mathbf{v}$ (and thus only observing outcomes for a subset of the population). Addanki et al. [1] similarly propose an algorithm for the partial observation setting based on leverage score sampling. However, instead of pseudo-outcome regression, they learn separate approximations to both outcome vectors $\mathbf{y}^{(1)}$ and $\mathbf{y}^{(0)}$. Thus, their final error depends on the best linear fit error for both these vectors, rather than the more natural best linear fit error for the ITE vector $\mathbf{t}$.

### 1.2.2 Average Treatment Effect Estimation

We can extend our general approach to give bounds for average treatment effect estimation. In the full observation setting we show:

**Theorem 2** (ATE Estimation – Full Observation). *For any $\epsilon, \delta \in (0, 1]$, there exists a randomized algorithm (Algorithm 1) that computes an unbiased estimate $\widehat{\tau}$ of the ATE $\tau$, and for which there is an event $\mathcal{E}$ with $\Pr(\mathcal{E}) \geq 1 - \delta$ (over the randomness of the algorithm) such that*

$$\mathbb{E}\left[(\widehat{\tau} - \tau)^2 | \mathcal{E}\right] \leq \frac{8(1+\epsilon)}{n^2} \min_{\mathbf{b} \in \mathbb{R}^d} \left( \|\mathbf{X}\mathbf{b} - \boldsymbol{\mu}\|_2^2 + 100 \log(n/\delta) \cdot \zeta^2 \cdot \|\mathbf{b}\|_2^2 \right) + \frac{32d}{n^2} \cdot \|\mathbf{y}^{(1)} - \mathbf{y}^{(0)}\|_\infty^2,$$

*where $\zeta := \max_{i \in [n]} \|\mathbf{x}_i\|_2$ and $\boldsymbol{\mu} := \mathbf{y}^{(0)} + \mathbf{y}^{(1)}$ is the total outcome vector.*

As with our result in ITE estimate, the error bound of Theorem 2 matches the error of the best linear fit of the total outcome vector $\boldsymbol{\mu}$ to our covariates up to a constant factor plus additive terms depending on 1) the maximum magnitude of the sum of potential outcomes for any individual, and 2) the norm of the coefficient vector used to approximately reconstruct $\boldsymbol{\mu}$ from the covariates. Again, we expect these additive terms to scale as $O(1/n^2)$ while we expect the term depending on the best fit error to scale as $O(1/n)$. Thus, we generally expect the additive error to be lower order.

Harshaw et al. [18] proposed an algorithm based on the *Gram-Schmidt walk (GSW) design* (see Algorithm 5 in the appendix) for balancing assignments to treatment and control groups that achieves an unbiased estimator for ATE with variance of $\frac{1}{n^2} \min_{\mathbf{b}} \left[ \frac{1}{\phi} \|\boldsymbol{\mu} - \mathbf{X}\mathbf{b}\|_2^2 + \frac{\zeta^2}{1-\phi} \|\mathbf{b}\|_2^2 \right]$, where $\phi \in (0, 1)$ is chosen by the experiment designer. This guarantee is comparable to but stronger than ours, e.g., when one sets $\phi = 1/2$. If we ignore the additive error terms and focus just on the best linear fit error $\|\mathbf{X}\mathbf{b} - \boldsymbol{\mu}\|_2^2$, then GSW is better than our guarantees by about a factor of $8$. But the GSW design is computationally more expensive and, since is based on balancing the covariates, requires availability of all covariates before running the experiment and hence cannot be applied in an online setting [4]. In contrast, our approach can be applied directly in the online setting since we place each unit independently in the treatment or control groups with equal probability. Moreover, as we show in our experiments, the empirical performance of our algorithm is much closer to GSW than the theoretical bounds we are able to show.

We sketch the ideas behind Theorem 2 in Section 2. Our algorithm is based on the classic Horvitz-Thompson estimator [19]. Roughly, this estimator randomly assigns individuals to control and treatment groups, and estimates ATE as the difference in the average outcome between these groups, appropriately weighted by the assignment probabilities. It is well known that the variance of this estimator is bounded by $\frac{1}{n^2} \|\boldsymbol{\mu}\|_2^2$. To reduce this variance, we introduce *full-observation regression*

*adjusted Horvitz-Thompson (RAHT) estimator*, which estimates ATE as the following. We partition the units into two groups $S$ and $\overline{S}$ (this is a different partitioning than the partitioning into control and treatment), and we regress $\boldsymbol{\mu}_S$ and $\boldsymbol{\mu}_{\overline{S}}$ onto their corresponding covariates. As in the ITE case, since we cannot directly form $\boldsymbol{\mu}$, we instead perform the above regressions using a random vector $\mathbf{u}$ with $\mathbb{E}[\mathbf{u}] = \boldsymbol{\mu}$, and where $\mathbf{u}$ can be formed by only observing one potential outcome per individual. We then use the solution vector of group $S$ to adjust the outcomes of the group $\overline{S}$ and vice versa, and we apply the HT estimator to the adjusted outcomes. This gives the bound in Theorem 2.

Again, following a similar approach as in the ITE case, we can apply leverage score based subsampling to tackle ATE estimation in the partial observation setting as well, obtaining:

**Theorem 3** (ATE Estimation – Partial Observation). *For any $\epsilon, \delta \in (0, 1]$, $\phi \in (0, 1)$, and $m \in [n]$, there exists a randomized algorithm (Algorithm 4) that observes a potential outcome for $O(d \log(d/\delta)/\epsilon^2 + m)$ units and outputs an unbiased estimate $\hat{\tau}$ of the ATE $\tau$, for which there is an event $\mathcal{E}$ with $\Pr(\mathcal{E}) \geq 1 - \delta$ (over the randomness of the algorithm), such that*

$$\mathbb{E}\left[(\widehat{\tau} - \tau)^2 | \mathcal{E}\right] \leq \frac{1}{mn}\frac{1}{\phi}\|\mathbf{X}\mathbf{b}^* - \boldsymbol{\mu}\|_2^2 + \frac{1}{m^2}\frac{\zeta^2}{(1-\phi)}\|\mathbf{b}^*\|_2^2 + \frac{100 d \cdot \log(d/\delta)}{n^2 \epsilon^2}\|\boldsymbol{\mu}\|_\infty^2$$
$$+ (1 + \epsilon) \cdot \left(\frac{1}{m}\|\boldsymbol{\mu}\|_\infty^2 + \frac{1}{mn}\|(\mathbf{X} - \overline{\mathbf{X}})\widehat{\mathbf{b}} - (\mathbf{t} - \overline{\mathbf{t}})\|_2^2 + \frac{\lambda}{mn}\|\widehat{\mathbf{b}}\|_2^2\right),$$

*where $\zeta := \max_{i \in [n]} \|\mathbf{x}_i\|_2$, $\boldsymbol{\mu} := \mathbf{y}^{(0)} + \mathbf{y}^{(1)}$, $\lambda = \frac{6 \cdot \log(d/\delta)}{\epsilon^2} \cdot \zeta^2$, $\overline{\mathbf{t}} \in \mathbb{R}^n$ is a vector where all the entries are equal to $\overline{t} = \frac{1}{n}\sum_{i=1}^n t_i$, and*

$$\mathbf{b}^* = \arg\min_{b \in \mathbb{R}^d}\left[\frac{m}{n}\|\mathbf{X}\mathbf{b} - \boldsymbol{\mu}\|_2^2 + \zeta^2\|\mathbf{b}\|_2^2\right], \widehat{\mathbf{b}} = \arg\min_{\mathbf{b} \in \mathbb{R}^d}\left[\|(\mathbf{X} - \overline{\mathbf{X}})\mathbf{b} - (\mathbf{t} - \overline{\mathbf{t}})\|_2^2 + \lambda\|\mathbf{b}\|_2^2\right].$$

The error bound of Theorem 3 depends on 1) the error of best linear fits of the covariates onto $\boldsymbol{\mu}$ and $\mathbf{t} - \overline{\mathbf{t}}$, 2) the norm of the coefficient vector used to approximately reconstruct $\boldsymbol{\mu}$ and $\mathbf{t} - \overline{\mathbf{t}}$ from the covariates, and 3) the largest component of the total outcome vector $\boldsymbol{\mu}$. We expect both the best linear fit and $\|\boldsymbol{\mu}\|_\infty$ terms to scale as $O(\frac{1}{m})$. All other terms are lower order. Namely, The terms depending on the norms of the coefficient vectors scale as $O(\frac{1}{mn})$ and $O(\frac{1}{m^2})$. For the case where $md \ll n$, by increasing the number of samples to $O(d \log(d/\delta)/\epsilon^2 + md)$, we can make the $\|\mu\|_\infty$ term to scale as $O(1/m^2)$ as well.

We deploy a different regression adjustment technique for achieving Theorem 3, which we call *partial-observation regression adjusted Horvitz-Thompson (RAHT) estimator*. In the partial observation setting, a simple approach is just to apply the full observation algorithm to a uniformly selected subset of $m$ units and then report the ATE estimate for these $m$ units. However, this leads to a variance bound depending on $\left\|\mathbf{t} - \overline{\mathbf{t}}\right\|_2^2$. To reduce this variance, the partial observation RAHT adjusts the estimate obtained from the $m$ units using regression adjustment techniques. We give the details behind the full algorithm in Section 2.3 and provide the full analysis in the appendix.

### 1.2.3 Experimental Evaluation

We compliment the above theoretical bounds with experiments using both synthetic and real data, focusing on the full observation ATE and ITE estimation problems. For ATE, we compare our method with classical regression adjustment [24], the GSW method [18], and the algorithm of [28]. We observe that our estimator (which is unbiased) performs as well or better than all other estimators except the classic regression adjustment, which is a biased estimator. For ITE, we compare our approach with the method of [1], and observe superior performance. A discussion about the running time of our algorithm and comparison to GSW design is included in the appendix.

### 1.3 Roadmap

We present the high-level ideas and techniques for ITE and ATE estimation for the full and partial observation settings in Section 2. We then present notation and preliminaries required for a more in-depth discussion of our result in Section 3. In Section 4, we give a detailed non-asymptotic analysis of the random vector (i.e., pseudo-outcome) regression technique, which forms the basis for all of our algorithms. We present our algorithm for the full observation ATE estimation and sketch the proof of its accuracy (Theorem 2). Finally, we present our experimental results in Section 5.

We relegate the details of other algorithms and proofs to the appendix, along with additional details and results for the experimental results. In the appendix, we also include a warm-up result for the mean estimation problem in the partial observation setting (where there is only one outcome $\mathbf{y}$ and we seek to estimate $\frac{1}{n}\sum_{i=1}^{n} y_i$ from a small number of samples). This result illustrates the key ideas behind leverage score sampling, which is used for our results on both ITE and ATE estimation. In addition we discuss how to remove the $1 - \delta$ probability from our results in Section C.1.

## 2 Technical Overview

In this section, we sketch the key ideas behind our main results on ITE estimation (Theorem 1) and ATE estimation (Theorems 2 and 3).

### 2.1 ITE Estimation

We first consider ITE estimation in the full observation setting. The key challenge here is that we must infer the full treatment effect vector $\mathbf{t} = \mathbf{y}^{(1)} - \mathbf{y}^{(0)}$, while only observing one outcome, $y_i^{(1)}$ or $y_i^{(0)}$ for any unit $i$. To do so, we use the idea of *pseudo-outcome* regression in [22]. We construct a random vector $\mathbf{v} \in \mathbb{R}^n$ that is equal to $\mathbf{t}$ in expectation, i.e., $\mathbb{E}[\mathbf{v}] = \mathbf{t}$ but only requires observing one outcome per unit. Specifically, we independently set $v_i = 2y_i^{(1)}$ with probability $0.5$ and $v_i = -2y_i^{(0)}$ with probability $0.5$. We then regress $\mathbf{v}$ onto our covariates, and use the result to estimate $\mathbf{t}$. In particular, we set $\widehat{\mathbf{b}} := \arg\min_{\mathbf{b} \in \mathbb{R}^d} \|\mathbf{X}\mathbf{b} - \mathbf{v}\|_2^2$ and construct our ITE estimate as $\widehat{\mathbf{t}} = \mathbf{X}\widehat{\mathbf{b}}$.

To give a non-asymptotic error bound for this approach, we observe that we can write $\mathbf{v} = \mathbf{t} + \mathbf{z} \odot \boldsymbol{\mu}$, where $\mathbf{z} \in \{1, -1\}^n$ is a random sign vector, $\boldsymbol{\mu} = \mathbf{y}^{(1)} + \mathbf{y}^{(0)}$ is the sum of potential outcome vectors, and $\odot$ denotes the Hadamard (entrywise) product. Thus, letting $\pi_{\mathbf{X}} \in \mathbb{R}^{n \times n}$ be the projection matrix onto the column span of $\mathbf{X}$, we can write $\widehat{\mathbf{b}} = \pi_{\mathbf{X}}\mathbf{v} = \pi_{\mathbf{X}}\mathbf{t} + \pi_{\mathbf{X}}(\mathbf{z} \odot \boldsymbol{\mu})$. In turn, we have

$$\mathbb{E}\left[\|\widehat{\mathbf{t}} - \mathbf{t}\|_2^2\right] = \mathbb{E}\left[\|\mathbf{X}\widehat{\mathbf{b}} - \mathbf{t}\|_2^2\right] = \mathbb{E}\left[\|\pi_{\mathbf{X}}\mathbf{t} - \mathbf{t}\|_2^2 + \|\pi_{\mathbf{X}}(\mathbf{z} \odot \boldsymbol{\mu})\|_2^2 + 2\mathbf{z}^T\mathbf{X}^T\mathbf{X}(\mathbf{z} \odot \boldsymbol{\mu})\right].$$

The first term on the righthand side is simply the best linear fit error for $\mathbf{t}$, $\|\pi_{\mathbf{X}}\mathbf{t} - \mathbf{t}\|_2^2 = \min_{\mathbf{b} \in \mathbb{R}^d} \|\mathbf{X}\mathbf{b} - \mathbf{t}\|_2^2$. The third term is $0$ in expectation since $\mathbb{E}[\mathbf{z} \odot \boldsymbol{\mu}] = 0$. Finally, since the entries of $\mathbf{z} \circ \boldsymbol{\mu}$ are independent and mean $0$, the expectation of the second term can be bounded by $\mathrm{tr}(\pi_{\mathbf{X}}) \cdot \|\boldsymbol{\mu}\|_\infty^2 = d\|\boldsymbol{\mu}\|_\infty^2$. Putting these all together, we have

$$\mathbb{E}\left[\|\widehat{\mathbf{t}} - \mathbf{t}\|_2^2\right] \leq \min_{\mathbf{b} \in \mathbb{R}^d} \|\mathbf{X}\mathbf{b} - \mathbf{t}\|_2^2 + d\|\boldsymbol{\mu}\|_\infty^2, \tag{1}$$

which gives the bound of Theorem 1 in the full observation setting. We note that this expected error bound can also be turned into a high probability bound using concentration inequalities, i.e., Hanson-wright inequality [31].

**ITE Estimation with Partial Observation.** Our next step is to extend the above to the partial observation setting. To do so, we apply leverage score sampling, which is a standard technique for approximate regression via subsampling [10, 26, 35]. It is well known that if we sample $O(d\log(d/\delta)/\epsilon^2)$ rows of $\mathbf{X}$ and the corresponding entries of $\mathbf{v}$ according to the leverage scores of $\mathbf{X}$, then we can solve a reweighted regression problem on just these rows to find $\widehat{\mathbf{b}}$ satisfying $\left\|\mathbf{X}\widehat{\mathbf{b}} - \mathbf{v}\right\|_2^2 \leq (1 + \epsilon) \cdot \min_{\mathbf{b} \in \mathbb{R}^d} \|\mathbf{X}\mathbf{b} - \mathbf{v}\|_2^2$. Observe that to compute $\widehat{\mathbf{b}}$, we only need to observe potential outcomes for the units (i.e., the entries of $\mathbf{v}$) that are sampled. Thus, this almost allows us to recover a similar bound to (1), up to a multiplicative $(1 + \epsilon)$ factor, in the partial observation setting. However, the leverage score of any one row may be very small, and if sampled, that row may be scaled up by a large factor in our reweighted regression problem (to preserve expectation). It thus becomes difficult to control to error introduced by approximating $\mathbf{t}$ by $\mathbf{v}$.

To handle this issue, we mix leverage score sampling, with uniform sampling at rate $\alpha$. This increases our sample complexity to $O(d\log(d/\delta)/\epsilon^2 + \alpha n)$, but allows us to bound the error due to approximating $\mathbf{t}$ by $\mathbf{v}$ by $\frac{(1+\epsilon)d}{\alpha} \|\boldsymbol{\mu}\|_2^2$. This yields the final sample complexity and error bound of Theorem 1. For a full proof, see Appendix B.

## 2.2 ATE with Full Observation

We build on our techniques for ITE estimation to tackle the ATE estimation problem. We start with the classic Horvitz-Thompson (HT) estimator for ATE, defined below.

**Definition 4** (Horvitz–Thompson estimator). *Given two outcome vectors* $\mathbf{y}^{(1)}, \mathbf{y}^{(0)} \in \mathbb{R}^n$, *let* $Z^+$ *and* $Z^- = [n] \setminus Z^+$ *be a random partitioning of the units to two groups under a distribution* $\mathcal{P}$. *Then the* Horvitz–Thompson *(HT) estimator is* $\widehat{\tau} := \frac{1}{n}(\sum_{i \in Z^+} \frac{y_i^{(1)}}{\mathbb{P}_{\mathcal{P}}[i \in Z^+]} - \sum_{i \in Z^-} \frac{y_i^{(0)}}{\mathbb{P}_{\mathcal{P}}[i \in Z^-]})$, *where* $\mathbb{P}_{\mathcal{P}}[i \in Z^+]$ *denotes the probability of placing* $i$ *in* $Z^+$ *when the partitioning is performed according to distribution* $\mathcal{P}$.

Note that if units are assigned independently with equal probability to treatment and control, the HT estimator is equivalent to outputting the average of the random vector $\mathbf{v}$ used in our regression-based estimate for ITE. In this case, the HT estimator is unbiased and well known to have variance bounded by $\frac{1}{n^2}\|\boldsymbol{\mu}\|_2^2$, where $\boldsymbol{\mu} := \mathbf{y}^{(1)} + \mathbf{y}^{(0)}$. Our goal is to improve this variance to instead depend on the error of the best linear fit to $\boldsymbol{\mu}$, $\min_{\mathbf{b} \in \mathbb{R}^d} \|\mathbf{X}\mathbf{b} - \boldsymbol{\mu}\|_2^2$, as in Theorem 2. To do so, we will employ the classic technique of regression adjustment.

**Horvitz-Thompson on Regression Adjusted Outcomes.** Suppose we were given a vector $\mathbf{b} \in \mathbb{R}^d$ determined independently of the randomness in the HT estimator. Then applying the HT estimator directly on the regression adjusted vectors $\widetilde{\mathbf{y}}^{(i)} := \mathbf{y}^{(i)} - \mathbf{X}\mathbf{b}$ (which have the same ATE as the original vectors), would yield an unbiased estimate for the ATE with variance at most $\frac{1}{n^2}\|\mathbf{X}\mathbf{b} - \boldsymbol{\mu}\|_2^2$. Naturally, we would like to find such a $\mathbf{b}$ making this variance bound as small as possible.

To do so, we will apply the same approach as in ITE estimation. We cannot directly regress $\boldsymbol{\mu}$ onto our covariates to find an optimal $\mathbf{b}$, as we cannot observe any entry of $\boldsymbol{\mu}$ since it is the sum of the two potential outcomes. Instead, we can construct a vector $\mathbf{u}$ with $\mathbb{E}[\mathbf{u}] = \boldsymbol{\mu}$, and use it as a surrogate in our regression problem. In particular, we let $u_i = 2y_i^{(1)}$ with probability $1/2$ and $u_i = 2y_i^{(0)}$ with probability $1/2$. As in the ITE setting, we can bound the error incurred by approximating $\boldsymbol{\mu}$ by $\mathbf{u}$ in our regression problem, and thus bound the variance of our regression adjusted HT estimator. We note that to the best of our knowledge, this idea has not been leveraged in ATE estimation.

Unfortunately, a difficulty arises in this approach. We need to assign each unit to a treatment or control group both when computing $\mathbf{u}$ to solve our regression problem, and later when applying the HT estimator. If we use the same assignment for both steps (as is required since we can only observe one of $y_i^{(1)}$ or $y_i^{(0)}$ for each unit), we introduce bias in our estimator.

**Avoiding Estimator Bias.** To alleviate this issue, we use *ridge leverage score sampling* to first partition the units into two groups. Ridge leverage score sampling is a technique similar to leverage score sampling that gives similar approximation guarantees to linear regression but for ridge regression problems (i.e., linear regression problems with an $\ell_2$ regularization term). We pick a ridge (regularization) parameter that guarantees that each unit only needs to be in the first or second group with a probability of at most $0.5$ while guaranteeing that the solution obtained from this sampling is a $(1 + \epsilon)$ approximation of the optimal ridge regression solution. Then we solve ridge regression problems on the groups separately to obtain two vectors $\widehat{\mathbf{b}}^{(1)}$ and $\widehat{\mathbf{b}}^{(2)}$, for the first and the second group, respectively. We then use $\widehat{\mathbf{b}}^{(1)}$ to adjust the outcomes of the second group and use $\widehat{\mathbf{b}}^{(2)}$ to adjust the outcomes of the first group. Since the vector of each group is not used to adjust the outcomes of itself, this gives an unbiased estimator.

## 2.3 ATE with Partial Observation

We now consider ATE estimation where we desire to only observe $y_i^{(1)}$ or $y_i^{(0)}$ for a small subset of the population. If we draw a subset of size $m$ uniformly at random and observe the outcomes of the samples in the subset independently and with equal probability and apply the Horvitz-Thompson estimator, the variance is $\mathbb{E}\left[(\widehat{\tau} - \tau)^2\right] \leq \frac{\|\boldsymbol{\mu}\|_2^2}{mn} + \frac{\|\mathbf{t} - \bar{\mathbf{t}}\|_2^2}{m}$. We would like to replace terms $\|\boldsymbol{\mu}\|_2^2$ and $\|\mathbf{t} - \bar{\mathbf{t}}\|_2^2$ with $\min_{\mathbf{b}} \|\boldsymbol{\mu} - \mathbf{X}\mathbf{b}\|^2$ and $\min_{\mathbf{b}} \|\mathbf{t} - \bar{\mathbf{t}} - (\mathbf{X} - \overline{\mathbf{X}})\mathbf{b}\|^2$, respectively. To deal with the $\|\boldsymbol{\mu}\|_2^2$ term, we use the HT estimator applied to the GSW design [18] on the uniformly selected subset of size $m$ of the data. Then for $\phi \in (0, 1)$ that the experiment designer selects, The variance reduces to $\frac{1}{m^2} \min_{b \in \mathbb{R}^d} \left[ \frac{m}{n} \frac{1}{\phi} \|\mathbf{X}\mathbf{b} - \boldsymbol{\mu}\|_2^2 + \frac{\zeta^2}{1-\phi} \|\mathbf{b}\|_2^2 \right] + \frac{\|\mathbf{t} - \bar{\mathbf{t}}\|_2^2}{mn}$. To replace the $\frac{\|\mathbf{t} - \bar{\mathbf{t}}\|_2^2}{mn}$ term with the error

of the best linear fit, we use the partial observation RAHT estimator that given a solution vector $\widetilde{\mathbf{b}}$ adjusts the estimate obtained by the GSW design on the sample of size $m$. Similar to our approach for partial observation ITE estimation, we compute $\widetilde{\mathbf{b}}$ by sampling about $d$ units according to leverage scores of the matrix and performing the random vector regression (pseudo-outcome regression) on a vector that is equal to $\mathbf{t}/2$ in expectation. Let $S$ be the set of units selected by leverage score sampling and $\overline{S} = [n] \setminus S$. We denote the estimate obtained by sampling $m$ units from $\overline{S}$ (we denote this set with $T$) and applying GSW design with $\widehat{\tau}_{\overline{S}}$. Then, the RAHT estimator estimates the ATE as $\widehat{\tau}_{\overline{S}} - \frac{1}{m} \sum_{i \in T} (\mathbf{x}_i - \overline{\mathbf{x}}^{\overline{S}}) \widetilde{\mathbf{b}}$. This estimate is biased because some units have a higher probability of being included in $S$ and, therefore, would contribute to the estimate with a lower probability. To resolve this issue, we also estimate the ATE on the set $S$ with the same random vector of outcomes that is used in regression to learn $\widetilde{\mathbf{b}}$. We use HT estimator for this estimate, and we denote it by $\widehat{\tau}_S$. Then, our final estimate is a convex combination of the two estimates as the following.

$$\frac{|\overline{S}|}{n} \cdot \left( \widehat{\tau}_{\overline{S}} - \frac{1}{m} \sum_{i \in T} (\mathbf{x}_i - \overline{\mathbf{x}}^{\overline{S}}) \widetilde{\mathbf{b}} \right) + \frac{|S|}{n} \cdot \widehat{\tau}_S.$$

This estimator is unbiased and achieves the variance bounds stated in Theorem 3.

## 3  Preliminaries

**Notation.** Vectors and matrices are denoted with bold lowercase and capital letters, respectively. the Hadamard (entrywise) product of $\mathbf{x}, \mathbf{y}$ is denoted by $\mathbf{x} \odot \mathbf{y}$. We denote the vectors of all ones and all zeros in $\mathbb{R}^n$ by $\mathbf{1}_n$ and $\mathbf{0}_n$, respectively. We drop the subscript if the dimension is clear. We denote the identity matrix of dimension $n$ by $\mathbf{I}_n$. For $\mathbf{X} \in \mathbb{R}^{n \times d}$ we denote the projection matrix of $\mathbf{X}$ by $\pi_{\mathbf{X}} := \mathbf{X}(\mathbf{X}^\top \mathbf{X})^+ \mathbf{X}$, where $+$ denotes the pseudoinverse. For $\mathbf{x} \in \mathbb{R}^n$ and $i \leq n$, $x_i$ denotes its $i^{th}$ entry and $\mathbf{x}_{1:i}$ denotes a vector in $\mathbb{R}^i$ that is equal to its first $i$ entries. We use $\mathbf{X}_{i,:}$ and $\mathbf{x}_i$ interchangeably to denote row $i$ of $\mathbf{X}$. Similarly for $S \subseteq [n]$, $\mathbf{X}_{S,:} \in \mathbb{R}^{|S| \times d}$ denotes the submatrix with row indices in $S$ and for $\mathbf{y} \in \mathbb{R}^n$, $\mathbf{y}_S \in \mathbb{R}^{|S|}$ denotes the vector restricted to indices in $S$.

**Definition 5** (Ridge leverage scores [2]). *Let $\mathbf{X} \in \mathbb{R}^{n \times d}$, $\lambda \geq 0$, and $\mathbf{x}_i \in \mathbb{R}^d$ be the $i^{th}$ row of $\mathbf{X}$. Then the $\lambda$-ridge leverage score of row $i$ of $\mathbf{X}$ is defined as $\ell_i^{(\lambda)} := \mathbf{x}_i^\top (\mathbf{X}^\top \mathbf{X} + \lambda \mathbf{I})^{-1} \mathbf{x}_i$. We denote the leverage scores (i.e., $\lambda = 0$) with $\ell_i$. If the corresponding matrix $\mathbf{X}$ is not clear from the context, we denote the $\lambda$-ridge leverage scores with $\ell_i^{(\lambda)}(\mathbf{X})$.*

Note that $\ell_i$ is the $i$'th diagonal entry of the projection matrix $\pi_{\mathbf{X}} = \mathbf{X}(\mathbf{X}^\top \mathbf{X})^{-1} \mathbf{X}^\top$. Thus, since $\mathrm{tr}(\pi_{\mathbf{X}}) = \mathrm{rank}(\mathbf{X})$, $\sum_{i=1}^n \ell_i = \mathrm{rank}(\mathbf{X})$. Since in our approach, we need to sample different sets of population for different purposes, we require a bound on the ridge leverage scores that can be achieved by taking the regularization factor $\lambda$ to be large enough.

**Theorem 6.** *Let $\zeta := \max_{i \in [n]} \|\mathbf{x}_i\|_2$. Then for $\lambda \geq c \cdot \zeta^2$, for $c \geq 1$, $\ell_i^{(\lambda)}(\mathbf{X}) \leq \frac{1}{c}$ for all $i \in [n]$.*

The following theorem provides a sampling procedure based on $\lambda$-ridge leverage scores to solve a linear ridge regression problem approximately.

**Theorem 7** (6, 12). *Let $\epsilon, \delta \in (0, 1]$, and $\mathbf{u} \in \mathbb{R}^n$ be a vector that upper bounds the $\lambda$-ridge leverage scores of a matrix $\mathbf{X} \in \mathbb{R}^{n \times d}$, i.e., $\ell_i^{(\lambda)} \leq u_i$. Let $\mathbf{S} = \mathbb{R}^{n \times n}$ be a random diagonal matrix in which $\mathbf{S}_{ii} = 1/\sqrt{p_i}$ with probability $p_i = \min\{1, 3 \cdot u_i \cdot \log(d/\delta)/\epsilon\}$, and $\mathbf{S}_{ii} = 0$, otherwise. Then for any $\mathbf{y} \in \mathbb{R}^n$, letting $\widehat{\mathbf{b}} = \arg\min_{\mathbf{b} \in \mathbb{R}^d} \left[ \|\mathbf{S}\mathbf{X}\mathbf{b} - \mathbf{S}\mathbf{y}\|_2^2 + \lambda \|\mathbf{b}\|_2^2 \right]$, with probably at least $1 - \delta$,*

$$\|\mathbf{X}\widehat{\mathbf{b}} - \mathbf{y}\|_2^2 + \lambda \|\widehat{\mathbf{b}}\|_2^2 \leq (1 + \epsilon) \min_{\mathbf{b} \in \mathbb{R}^d} \left[ \|\mathbf{X}\mathbf{b} - \mathbf{y}\|_2^2 + \lambda \|\mathbf{b}\|_2^2 \right].$$

## 4  Average Treatment Effect Estimation

In this section, we first analyze the random vector regression (i.e., pseudo-outcome regression) approach which is used both for our ITE and ATE estimation. We then discuss our full observation ATE estimation using this approach and provide a sketch of the proof of Theorem 2.

### 4.1  Random Vector Regression

For both ATE and ITE estimation, regression adjustment allows us to reduce the variance of estimation to the error of the best linear fit on vectors $\boldsymbol{\mu}$ and $\mathbf{t}$, respectively. Since we do not have access to the

entries of these vectors, we cannot compute a solution vector directly. Therefore, for ATE (or ITE), our approach is to instead compute a solution vector $\mathbf{b}$ by performing regression on a random vector that, in expectation, is equal to $\boldsymbol{\mu}$ (or $\mathbf{t}$). The following theorem characterizes the error of the solution vector obtained in this way.

The following is a more general result that combines this random assignment technique with leverage score sampling to allow us to observe only a small subset of the population. Algorithm 3 and the proof of the following theorem are presented in the appendix.

**Theorem 8** (Random vector regression). *Let $\mathbf{y}^{(0)}, \mathbf{y}^{(1)} \in \mathbb{R}^n$ and $\mathbf{y} \in \mathbb{R}^n$ be a random vector such that for each $i \in [n]$, $y_i$ is independently and with equal probability is either equal to $y_i^{(0)}$ or $y_i^{(1)}$. Moreover, let $\mathbf{b}^* = \arg\min_{\mathbf{b}} \|\mathbf{X}\mathbf{b} - \mathbf{y}\|_2^2$. Let $\boldsymbol{\mu} := \mathbf{y}^{(1)} + \mathbf{y}^{(0)}$. Then*

$$\mathbb{E}\left[\|2\mathbf{X}\mathbf{b}^* - \boldsymbol{\mu}\|_2^2\right] \le d\|\mathbf{y}^{(1)} - \mathbf{y}^{(0)}\|_\infty^2 + \min_{\mathbf{b}} \|\mathbf{X}\mathbf{b} - \boldsymbol{\mu}\|_2^2.$$

*Proof.* Let $\mathbf{z} \in \{-1, +1\}^n$, where $z_i = +1$ if $y_i = y_i^{(1)}$, and $z_i = -1$, otherwise. First, note that $2\mathbf{y} = \boldsymbol{\mu} + \mathbf{z} \odot \mathbf{t}$, where $\boldsymbol{\mu} = \mathbf{y}^{(1)} + \mathbf{y}^{(0)}$, and $\mathbf{t} = \mathbf{y}^{(1)} - \mathbf{y}^{(0)}$. Therefore

$$\boldsymbol{\mu} - 2\mathbf{X}\mathbf{b}^* = \boldsymbol{\mu} - 2\pi_{\mathbf{X}}\mathbf{y} = \boldsymbol{\mu} - \pi_{\mathbf{X}}(\boldsymbol{\mu} + \mathbf{z} \odot \mathbf{t}) = (\boldsymbol{\mu} - \pi_{\mathbf{X}}\boldsymbol{\mu}) - \pi_{\mathbf{X}}(\mathbf{z} \odot \mathbf{t}).$$

Moreover since $\mathbf{I} - \pi_{\mathbf{X}}$ and $\pi_{\mathbf{X}}$ are orthogonal to each other,

$$\|\boldsymbol{\mu} - 2\mathbf{X}\mathbf{b}^*\|_2^2 = \|(\boldsymbol{\mu} - \pi_{\mathbf{X}}\boldsymbol{\mu})\|_2^2 + \|\pi_{\mathbf{X}}(\mathbf{z} \odot \mathbf{t})\|_2^2.$$

Now note that $\|(\boldsymbol{\mu} - \pi_{\mathbf{X}}\boldsymbol{\mu})\|_2^2 = \min_{\mathbf{b}} \|\mathbf{X}\mathbf{b} - \boldsymbol{\mu}\|_2^2$. So we only need to bound $\mathbb{E}\left[\|\pi_{\mathbf{X}}(\mathbf{z} \odot \mathbf{t})\|_2^2\right]$. We have $\|\pi_{\mathbf{X}}(\mathbf{z} \odot \mathbf{t})\|_2^2 = (\mathbf{z} \odot \mathbf{t})^\top \pi_{\mathbf{X}}(\mathbf{z} \odot \mathbf{t}) = \mathbf{z}^\top \mathbf{T}\pi_{\mathbf{X}}\mathbf{T}\mathbf{z}$, where $\mathbf{T}$ is the diagonal matrix associated with the vector $\mathbf{t}$. Since $z_i$ and $z_j$ are independent for $i \ne j$, we have

$$\mathbb{E}\left[\mathbf{z}^\top \mathbf{T}\pi_{\mathbf{X}}\mathbf{T}\mathbf{z}\right] = \sum_{i=1}^n t_i^2 (\pi_{\mathbf{X}})_{ii} \le \|\mathbf{t}\|_\infty^2 (\pi_{\mathbf{X}})_{ii}.$$

Then the result follows by noting that $(\pi_{\mathbf{X}})_{ii}$ is equal to the leverage score of row $i$, and the sum of leverage scores is less than or equal to $d$. □

We can use the same theorem to characterize the regression error for $\mathbf{t}$ by changing the sign of $\mathbf{y}^{(0)}$.

## 4.2 Full Observation ATE

We now sketch a proof of our result for ATE estimation with full observation.

*Proof sketch of Theorem 2.* Here, we only sketch our proof for the variance bound. For $S \subseteq [n]$ and $\mathbf{z} \in \{-1, +1\}^n$ (where $Z^+ = \{i : z_i = +1\}$ and $Z^- = \{i : z_i = -1\}$), we define

$$\tau_S := \frac{1}{|S|} \cdot \sum_{i \in S}(y_i^{(1)} - y_i^{(0)}), \quad \text{and} \quad \widehat{\tau}_S := \frac{2}{|S|} \cdot \left(\sum_{i \in Z^+ \cap S} \widetilde{y}_i^{(2,1)} - \sum_{i \in Z^- \cap S} \widetilde{y}_i^{(2,0)}\right).$$

Similarly, define $\tau_{\overline{S}}$ and $\widehat{\tau}_{\overline{S}}$. Then for $\widehat{\tau}$ as defined in Algorithm 1, we have,

$$\mathbb{E}_{\mathbf{z}}\left[\mathbb{E}_S\left[(\widehat{\tau} - \tau)^2\right]\right] = \mathbb{E}_{\mathbf{z}}\left[\mathbb{E}_S\left[\left(\frac{|S| \cdot \widehat{\tau}_S + |\overline{S}| \cdot \widehat{\tau}_{\overline{S}}}{n} - \frac{|S| \cdot \tau_S + |\overline{S}| \cdot \tau_{\overline{S}}}{n}\right)^2\right]\right]$$

$$= \mathbb{E}_S\left[\frac{|\overline{S}|^2}{n^2} \cdot \mathbb{E}_{\mathbf{z}}\left[(\widehat{\tau}_{\overline{S}} - \tau_{\overline{S}})^2\right] + \frac{|S|^2}{n^2} \cdot \mathbb{E}_{\mathbf{z}}\left[(\widehat{\tau}_S - \tau_S)^2\right]\right] + \mathbb{E}_S\left[2 \cdot \frac{|S| \cdot |\overline{S}|}{n^2} \cdot \mathbb{E}_{\mathbf{z}}\left[(\widehat{\tau}_{\overline{S}} - \tau_{\overline{S}}) \cdot (\widehat{\tau}_S - \tau_S)\right]\right].$$

By Cauchy-Schwarz inequality, $\mathbb{E}_{\mathbf{z}}\left[(\widehat{\tau}_{\overline{S}} - \tau_{\overline{S}}) \cdot (\widehat{\tau}_S - \tau_S)\right] \le \sqrt{\mathbb{E}_{\mathbf{z}}\left[(\widehat{\tau}_{\overline{S}} - \tau_{\overline{S}})^2\right] \cdot \mathbb{E}_{\mathbf{z}}\left[(\widehat{\tau}_S - \tau_S)^2\right]}$. Therefore we only need to bound $\mathbb{E}_{\mathbf{z}}\left[(\widehat{\tau}_{\overline{S}} - \tau_{\overline{S}})^2\right]$ and $\mathbb{E}_{\mathbf{z}}\left[(\widehat{\tau}_S - \tau_S)^2\right]$. We can bound $\mathbb{E}_{\mathbf{z}}\left[(\widehat{\tau}_S - \tau_S)^2\right]$ by Theorem 7 (ridge leverage score sampling) and using the fact that $\widehat{\tau}_S$ is obtained by $\widetilde{\mathbf{y}}^{(2,1)}$ and $\widetilde{\mathbf{y}}^{(2,0)}$ which are adjusted by $\widehat{\mathbf{b}}^{(2)}$, a vector that is learned from the units in the set $\overline{S}$. Since $S$ and $\overline{S}$ are disjoint, $\widehat{\mathbf{b}}^{(2)}$ is independent of $\mathbf{z}_S$ and this allows us to bound $\mathbb{E}_{\mathbf{z}}\left[(\widehat{\tau}_S - \tau_S)^2\right]$. The bound on $\mathbb{E}_{\mathbf{z}}\left[(\widehat{\tau}_{\overline{S}} - \tau_{\overline{S}})^2\right]$ follows from a similar argument. □

# 5 Experiments

In this section, we compare our method for estimating ATE and ITE with full observation with prior art and baselines on synthetic and real-world datasets. For ATE estimation, our experiments demonstrate that our approach either outperforms other unbiased estimation approaches or has comparable performance in terms of the achieved variance.

---
**Algorithm 1:** ATE estimation with leverage score sampling and cross adjustment
---
1 **Input:** $\mathbf{X} \in \mathbb{R}^{n \times d}$, $\mathbf{y}^{(0)}, \mathbf{y}^{(1)} \in \mathbb{R}^n$, $\epsilon, \delta \in (0, 1]$.

2 Set $\mathbf{y} \in \mathbb{R}^n$ to a random vector such that for each $i \in [n]$, $y_i$ is independently and with equal probability either equal to $y_i^{(0)}$ or $y_i^{(1)}$.

3 Set $Z^+$ to the set of entries $i$ such that $y_i$ is equal to $y_i^{(1)}$ and $Z^- = [n] \setminus Z^+$.

4 Set $\lambda = \frac{6}{\epsilon^2} \cdot \zeta^2 \log(n/\delta)$, where $\zeta = \max_{i \in [n]} \|\mathbf{x}_i\|_2$.

5 Set $S \subseteq [n]$ to a random set of indices where each $i \in [n]$ is included in $S$ independently and with probability 0.5. Set $\overline{S} = [n] \setminus S$.

6 Set $\mathbf{S}$ to an $n \times n$ diagonal matrix corresponding to $S$ where each entry $\mathbf{S}_{ii} = 2$ if $i \in S$, and $\mathbf{S}_{ii} = 2$, otherwise. Similarly, set $\overline{\mathbf{S}}$.

7 Set $\widehat{\mathbf{b}}^{(1)} = \arg\min_{\mathbf{b}} \|\mathbf{SXb} - \mathbf{Sy}\|_2^2 + \lambda \cdot \|\mathbf{b}\|_2^2$, $\widehat{\mathbf{b}}^{(2)} = \arg\min_{\mathbf{b}} \|\overline{\mathbf{S}}\mathbf{Xb} - \overline{\mathbf{S}}\mathbf{y}\|_2^2 + \lambda \cdot \|\mathbf{b}\|_2^2$.

8 Set $\widetilde{\mathbf{y}}^{(1,0)} = \mathbf{y}^{(0)} - \mathbf{X}\widehat{\mathbf{b}}^{(1)}$, $\widetilde{\mathbf{y}}^{(1,1)} = \mathbf{y}^{(1)} - \mathbf{X}\widehat{\mathbf{b}}^{(1)}$, $\widetilde{\mathbf{y}}^{(2,0)} = \mathbf{y}^{(0)} - \mathbf{X}\widehat{\mathbf{b}}^{(2)}$, $\widetilde{\mathbf{y}}^{(2,1)} = \mathbf{y}^{(1)} - \mathbf{X}\widehat{\mathbf{b}}^{(2)}$.

9 Return $\widehat{\tau} := \frac{2}{n}(\sum_{i \in Z^+ \cap \overline{S}} \widetilde{y}_i^{(1,1)} + \sum_{i \in Z^+ \cap S} \widetilde{y}_i^{(2,1)} - \sum_{i \in Z^- \cap \overline{S}} \widetilde{y}_i^{(1,0)} - \sum_{i \in Z^- \cap S} \widetilde{y}_i^{(2,0)})$.
---

**ATE methods.** Our ATE estimation method is compared against five other approaches. (i) **HT Uniform:** This employs the Hurvitz Thompson estimator, using a uniform assignment of units to treatment and control groups. (ii) **GSW:** This design follows the methodology in [18]. (iii) **Classic Regression Adjustment:** As described in [24], this estimator is inherently biased. (iv) **Leverage Score-Based:** In this method, we learn two distinct vectors for $\mathbf{y}^{(1)}$ and $\mathbf{y}^{(0)}$ through leverage score sampling. The difference between these vectors predicts the treatment effect. Though this method has its advantages, it is biased and was previously employed by Addanki et al. [1]. (v) **4 Vectors:** Obtained by specializing Mou et al. [28] to linear regressors, this approach adopts the same cross-adjustment mechanism to mitigate bias as in this paper. However, instead of random vector regression, it determines separate vectors for $\mathbf{y}^{(1)}$ and $\mathbf{y}^{(0)}$ for each group, leading to four vectors obtained by linear regressions.

**ITE methods.** We discuss two distinct approaches for ITE methods. (i) **Baseline:** The ideal scenario or "oracle" baseline involves utilizing a linear function, trained on vector $\mathbf{t}$. Naturally, this is not feasible in practical applications. (ii) **leverage score based:** This is based on learning two different vectors by ridge leverage score sampling and using the difference of the two linear functions for estimating $\mathbf{t}$. This is an adaptation of the algorithm in [1] to the full observation setting — we make this adaptation because the exact algorithm of [1] cannot be applied to the full-observation setting due to sampling with replacement. We do not compare to [28] because their guarantees are only for an infinite population.

**Synthetic Datasets.** For synthetic datasets, we will construct $\mathbf{y}^{(0)}, \mathbf{y}^{(1)}$ such that $\boldsymbol{\mu}$ (for the ATE case) or $\mathbf{t}$ (for the ITE case) is a linear function of the covariate matrix.

1. **ATE Dataset.** There are 50 covariates (i.e., features). Each entry of $\mathbf{X} \in \mathbb{R}^{n \times d}$ is a uniform random number in $[0, 0.01]$. Each entry of $\mathbf{b} \in \mathbb{R}^{50}$ is a uniform random number in $[0, 1]$. The individual treatment effect vector $\boldsymbol{\mu}$ is $\mathbf{Xb} + \mathbf{r}$, where each entry of $\mathbf{r} \in \mathbb{R}^n$ is randomly picked from a mean zero Gaussian distribution with standard deviation of 0.2. Each entry of $\mathbf{y}^{(0)}$ is picked uniformly at random from $[0, 5]$. Then $\mathbf{y}^{(1)} = -\mathbf{y}^{(0)} + \boldsymbol{\mu}$ and $\mathbf{t} = \mathbf{y}^{(1)} - \mathbf{y}^{(0)}$. The results for this dataset with different number of samples is shown in the right plot of Figure 1. Our experiments illustrate that on the synthetic dataset, our approach outperforms all other approaches except the GSW design. However, we note that our approach is computationally more efficient than GSW design (see Table 6) and gives a much simpler design that can be used in different settings.

2. **ITE Dataset.** The covariate matrix $\mathbf{X}$ and the vectors $\mathbf{b}, \mathbf{r}, \mathbf{y}^{(0)}$ are picked similar to the ATE dataset. Then individual treatment effect vector $\mathbf{t}$ is $\mathbf{Xb} + \mathbf{r}$ and $\mathbf{y}^{(1)} = \mathbf{y}^{(0)} + \mathbf{t}$. The results for this dataset with different number of samples is shown in the left plot of Figure 1. ITE error for our method is consistently smaller than leverage score based method.

**Real-World Datasets** We analyze the following three distinct real-world datasets. A comprehensive breakdown, including variance and other measurements for each dataset, is available in Table 5 in the appendix.

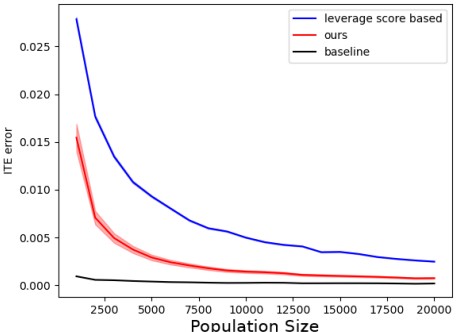 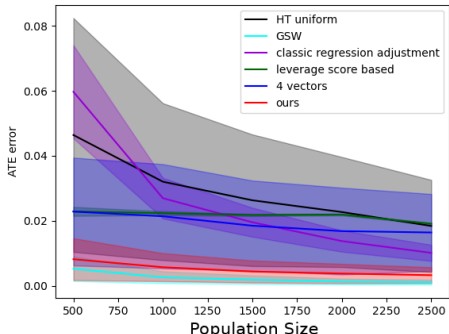

Figure 1: Results for synthetic ITE and ATE datasets are shown on the left and right, respectively. For different population sizes (i.e., $n$), estimation is performed for 1000 trials. Then the average of $\frac{\|\mathbf{t}-\widehat{\mathbf{t}}\|_2}{\sqrt{n}\cdot\|\mathbf{t}\|_2}$ over these trials is shown with solid lines for ITE and the shades around these lines denote the standard deviation. For ATE, the solid lines represent the average of $|\widehat{\tau}-\tau|/|\tau|$ over 1000 trials.

1. **Boston Dataset [17]:** This is a dataset of housing prices in the Boston area, consisting of 506 samples and 13 features. Since it has only one label, we set $\mathbf{y}^{(1)} = \mathbf{y}^{(0)}$, i.e., ATE is zero.
2. **IHDP Dataset [16, 11]:** Derived from the characteristics of children and their mothers, this dataset comprises 747 samples and 25 features. ATE for is -4.016.
3. **Twins Dataset [3]:** This dataset is constructed based on the characteristics and mortality rates of twin births in the US. We specifically selected samples that have complete feature values, resulting in a dataset of 32,120 samples with 50 features. The ATE of this dataset is 0.0064.

Table 1: Results of ATE estimation. For each result, the first number is the average of $|\tau - \widehat{\tau}|$ over 1000 trials and the second number is the standard deviation of this quantity.

| Dataset | Uniform HT | GSW | Classic Reg Adj | Lev Score | 4 vecs | Ours |
|---|---|---|---|---|---|---|
| Boston | 1.736 $\pm1.339$ | 0.663 $\pm0.510$ | 0.333 $\pm0.255$ | 0.658 $\pm0.504$ | 1.677 $\pm1.256$ | 0.628 $\pm0.459$ |
| IHDP | 0.272 $\pm0.206$ | 0.042 $\pm0.031$ | 0.012 $\pm0.009$ | 0.536 $\pm0.050$ | 0.264 $\pm0.203$ | 0.040 $\pm0.030$ |
| Twins | 1.351e$-$3 $\pm1.025$e$-$3 | 1.231e$-$3 $\pm0.937$e$-$3 | 1.201e$-$3 $\pm0.899$e$-$3 | 1.226e$-$3 $\pm0.911$e$-$3 | 1.369e$-$3 $\pm1.015$e$-$3 | 1.218e$-$3 $\pm0.936$e$-$3 |

Table 2: Results of ITE estimation. For each result, the first number is the average of $\frac{1}{\sqrt{n}}\|\mathbf{t}-\widehat{\mathbf{t}}\|_2$ over 1000 trials and the second number is the standard deviation of this quantity. The relative error columns report $\frac{1}{\sqrt{n}}\|\mathbf{t}-\widehat{\mathbf{t}}\|_2/\|\mathbf{t}\|_2$.

| Dataset | Baseline | Leverage Score | Ours | Leverage Score Relative | Ours Relative |
|---|---|---|---|---|---|
| Boston | 0.0 | $0.827 \pm 0.469$ | $7.654 \pm 1.678$ | - | - |
| IHDP | 0.548 | $2.449 \pm 0.071$ | $1.765 \pm 0.223$ | $0.021 \pm 0.637$e$-$3 | $0.015 \pm 1.9$e$-$3 |
| Twins | 0.155 | $0.156 \pm 2.66$e$-$5 | $0.156 \pm 2.21$e$-$4 | $5.571$e$-$3 $\pm 9.5$e$-$7 | $5.571$e$-$3 $\pm 7.9$e$-$7 |

# 6 Conclusion

In this paper, we considered the problems of mean estimation, ATE and ITE estimations in the presence of covariates. We considered the finite population setting and provided non-asymptotic variance bounds for several novel variants of the classical regression adjustment method-based estimators. Our guarantees are model-free, even if the covariates are arbitrary (are not informative of the estimand and may be chosen adversarially), the variance bounds are still meaningful and almost match the variance bounds of widely used estimators not based on covariates.

Our algorithms are simple and efficient, and we believe they readily extend to many related settings like arbitrary assignment probabilities and online treatment assignments. We also believe the results can be stated for the kernel setting. These extensions will be part of future work.

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

# A  Warm-up: Mean Estimation

In this section, we consider the problem of estimating the mean of a single vector $\mathbf{y}$ by observing only a few entries (i.e., $\bar{y} := \frac{1}{n}\sum_{i=1}^{n} y_i$). The most classic approach is to simply uniformly sample $m$ units from the population and report their average. This results in a variance of at most $\frac{1}{mn}\left\|\mathbf{y} - \overline{\mathbf{y}}\right\|_2^2$, where $\overline{\mathbf{y}}$ is a vector of the same size as $\mathbf{y}$ with all of its entries equal to $\bar{y}$.

The main theme of regression adjustment is to replace the dependence of variance on $\left\|\mathbf{y} - \overline{\mathbf{y}}\right\|_2^2$ with a dependence on the error of the best linear fit, i.e., $\min_{\mathbf{b}\in\mathbb{R}^d}\left\|\mathbf{y} - \overline{\mathbf{y}} - (\mathbf{X} - \overline{\mathbf{X}})\mathbf{b}\right\|_2^2$, where $\overline{\mathbf{X}}$ is a vector of the same size as $\mathbf{X}$ with all of its rows equal to $\overline{\mathbf{x}} = \frac{1}{n}\sum_{i=1}^{n}\mathbf{x}_i$. We wish to do so *without making any modeling assumptions* — a common modeling assumption in statistics is that the response vector is a linear function of the covariates plus a Gaussian noise. However, our goal is to achieve significant variance reduction if $\mathbf{y} - \overline{\mathbf{y}}$ can be well approximated by a vector in the span of $\mathbf{X} - \overline{\mathbf{X}}$ without any such modeling assumptions. In the worst case, when $\mathbf{y} - \overline{\mathbf{y}}$ is orthogonal to the column span of $(\mathbf{X} - \overline{\mathbf{X}})$, $\min_{\mathbf{b}\in\mathbb{R}^d}\left\|\mathbf{y} - \overline{\mathbf{y}} - (\mathbf{X} - \overline{\mathbf{X}})\mathbf{b}\right\|_2^2 = \left\|\mathbf{y} - \overline{\mathbf{y}}\right\|_2^2$ which recovers the result of the classic approach.

For the mean estimation problem, if an oracle provided $\mathbf{b}^* = \arg\min_{\mathbf{b}\in\mathbb{R}^d}\left\|\mathbf{y} - \overline{\mathbf{y}} - (\mathbf{X} - \overline{\mathbf{X}})\mathbf{b}\right\|_2^2$, the estimator $\widehat{y} = \frac{1}{|S|}\sum_{i\in S} y_i - (\mathbf{x}_i - \overline{\mathbf{x}})^\top \mathbf{b}^*$ is unbiased with its variance bounded by $\frac{1}{mn}\min_{\mathbf{b}\in\mathbb{R}^d}\left\|\mathbf{y} - \overline{\mathbf{y}} - (\mathbf{X} - \overline{\mathbf{X}})\mathbf{b}\right\|_2^2$, where $S$ is a uniformly sampled subset of the population of size $m$. This is a classic result (see Chapter 7 of [9]) that also follows from the following Lemma.

**Lemma 9** (9 - Chapter 2). *Let $\mathbf{y} \in \mathbb{R}^n$ and $\widehat{y}$ be the average of a uniformly sampled set of entries of $\mathbf{y}$ of size $m$. Then*

$$\mathbb{E}\left[(\widehat{y} - \bar{y})^2\right] = \frac{\left\|\mathbf{y} - \overline{\mathbf{y}}\right\|_2^2}{m(n-1)}\left(1 - \frac{m}{n}\right),$$

*where $\overline{\mathbf{y}}$ is a vector of size equal to $\mathbf{y}$ for which all the entries are equal to $\bar{y}$.*

One can easily see that by applying Lemma 9 to $\tilde{\mathbf{y}}$, where $\tilde{\mathbf{y}}_i = \mathbf{y}_i - (\mathbf{x}_i - \overline{\mathbf{x}})^\top \mathbf{b}$, a variance of $\frac{\left\|\mathbf{y} - \overline{\mathbf{y}} - (\mathbf{X} - \overline{\mathbf{X}})\mathbf{b}\right\|_2^2}{m(n-1)}\left(1 - \frac{m}{n}\right)$ can be obtained.

**Key challenge.**  The main challenge for the mean estimation problem is to compute $\mathbf{b}^*$ (or a vector close to it) without observing all the entries of vector $\mathbf{y}$. Note that we do not have access to any entry of $\mathbf{y} - \overline{\mathbf{y}}$ because we cannot compute $\bar{y}$ without observing all the entries of $\mathbf{y}$.

We resolve the issue of not being able to compute $\bar{y}$ by performing the regression on a modified matrix. We add a column of all ones to the matrix $\mathbf{X} - \overline{\mathbf{X}}$ and perform a regression on $\mathbf{y}$ using the modified matrix — see the proof of Theorem 10. More specifically, we prove the following.

$$\min_{\mathbf{b}\in\mathbb{R}^d}\left\|(\mathbf{X} - \overline{\mathbf{X}})\mathbf{b} - (\mathbf{y} - \overline{\mathbf{y}})\right\|_2^2 = \min_{\mathbf{b}\in\mathbb{R}^{d+1}}\left\|\begin{bmatrix}(\mathbf{X} - \overline{\mathbf{X}}) & \mathbf{1}\end{bmatrix}\mathbf{b} - \mathbf{y}\right\|_2^2$$

To solve the regression problem on the right-hand side without observing all the entries of $\mathbf{y}$, we use *leverage score sampling* [25, 10]. Leverage score sampling is a numerical linear algebraic tool that allows for solving a linear regression problem approximately by subsampling about $d$ units from the population and solving a regression problem over these units, where $d$ is the number of features [8] — see Section 3 for details. Combining these techniques gives the following results.

**Theorem 10** (Leverage score sampling based regression adjustment). *Let $\epsilon, \delta \in (0, 1]$, and $m \in \mathbb{N}$ and $\mathbf{X} \in \mathbb{R}^{n\times d}$ be a covariate matrix. Then, Algorithm 2 computes an unbiased estimator of the mean of an outcome vector $\mathbf{y}$ (on a population of size $n$) with $O(d\log(d/\delta)/\epsilon^2 + m)$ samples. Moreover, there is an event $\mathcal{E}$ with $\Pr(\mathcal{E}) \geq 1 - \delta$ (over the randomness of the algorithm) such that the variance of the estimator conditioned to $\mathcal{E}$ is bounded by*

$$(1 + \epsilon)\frac{1}{m(n-1)}\left(1 - \frac{m}{n}\right)\min_{\mathbf{b}}\left\|(\mathbf{X} - \overline{\mathbf{X}})\mathbf{b} - (\mathbf{y} - \overline{\mathbf{y}})\right\|_2^2. \tag{2}$$

Note that since the dependence of number of samples on $\delta$ is logarithmic, we can pick $\delta$ polynomially small, e.g., $\delta = 1/n^2$. The variance upper bound achieved by our method is compared to other methods in Table 3.

Table 3: Variance of different methods for estimating the mean of a population with few samples. $\zeta$ denotes the maximum norm over rows of $\mathbf{X}$, $0 < \phi < 1$ is an input parameter of GS walk design, and $\overline{\mathbf{y}}$ is a vector where all entries are equal to the mean of $\mathbf{y}$.

| Method | Variance | Reference |
|---|---|---|
| Uniform sampling $O(m)$ samples | $\frac{1}{m(n-1)}\left(1 - \frac{m}{n}\right)\|\mathbf{y} - \overline{\mathbf{y}}\|_2^2$ | [9] |
| GS walk design (by adjusting probabilities) $O(m)$ samples | $\frac{1}{m^2}\min_{\mathbf{b}\in\mathbb{R}^d}\left[\frac{1}{\phi}\left\|(\mathbf{X} - \overline{\mathbf{X}})\mathbf{b} - (\mathbf{y} - \overline{\mathbf{y}})\right\|_2^2 + \frac{1}{1-\phi}\zeta^2\|\mathbf{b}\|_2^2\right]$ | [18] |
| Two-phase $O(m\log(m/\delta)/\epsilon^2)$ samples | $(1+\epsilon)\frac{1}{m^2}\min_{\mathbf{b}\in\mathbb{R}^d}\left[\frac{m}{n}\|(\mathbf{X} - \overline{\mathbf{X}})\mathbf{b} - (\mathbf{y} - \overline{\mathbf{y}})\|_2^2 + \zeta^2\|\mathbf{b}\|_2^2\right]$ | This paper |
| Two-phase $O(m + d\log(d/\delta)/\epsilon^2)$ samples | $(1+\epsilon)\frac{1}{m(n-1)}\left(1 - \frac{m}{n}\right)\min_{\mathbf{b}\in\mathbb{R}^d}\left\|(\mathbf{X} - \overline{\mathbf{X}})\mathbf{b} - (\mathbf{y} - \overline{\mathbf{y}})\right\|_2^2$ | This paper |

**Regression adjustment.** Our estimators are based on a classic technique in statistics called regression adjustment [9]. Given a vector $\widehat{\mathbf{b}}$ (either given by an oracle or learned from the data), the classical technique uses the sample covariates to "adjust" the estimate of the sample mean $\frac{1}{|S'|}\sum_{i\in S'} y_i$ and outputs the following estimate

$$\frac{1}{|S'|}\sum_{i\in S'} y_i - (\mathbf{x}_i - \overline{\mathbf{x}})\widehat{\mathbf{b}}.$$

If $S'$ is a random sample independent of $\widehat{\mathbf{b}}$, the adjusted estimator is unbiased. The adjusted estimate will have a lower variance than the sample mean estimator if $\|\mathbf{y} - \overline{\mathbf{y}} - (\mathbf{X} - \overline{\mathbf{X}})\widehat{\mathbf{b}}\|_2^2 < \|\mathbf{y} - \overline{\mathbf{y}}\|_2^2$.

**Lemma 11.** *Let $\mathbf{y} \in \mathbb{R}^n$ and $\mathbf{X} \in \mathbb{R}^{n\times d}$. Let $\widehat{\mathbf{b}}$ be a fixed/preassigned vector. Then the variance of $\widehat{y} := \frac{1}{|S'|}\sum_{i\in S'} y_i - (\mathbf{x}_i - \overline{\mathbf{x}})^\top\widehat{\mathbf{b}}$, where $S'$ is a uniformly sampled subset of $[n]$ of size $m$, is*

$$\frac{1}{m(n-1)}\left(1 - \frac{m}{n}\right)\left\|(\mathbf{y} - \overline{\mathbf{y}}) - (\mathbf{X} - \overline{\mathbf{X}})\widehat{\mathbf{b}}\right\|_2^2.$$

*Proof.* Since $\widehat{\mathbf{b}}$ is fixed, by linearity of expectation

$$\mathbb{E}\left[\widehat{y}\right] = \mathbb{E}\left[\frac{1}{|S'|}\sum_{i\in S'} y_i\right] + \mathbb{E}\left[\frac{1}{|S'|}\sum_{i\in S'}(\overline{\mathbf{x}} - \mathbf{x}_i)^\top\widehat{\mathbf{b}}\right] = \overline{y} + (\overline{\mathbf{x}} - \frac{1}{n}\sum_{i\in S'}\mathbf{x}_i)^\top\widehat{\mathbf{b}} = \overline{y}.$$

Therefore by Lemma 9 (applied to the quantities $y_i + (\overline{\mathbf{x}} - \mathbf{x}_i)^\top\widehat{\mathbf{b}}$),

$$\mathbb{E}\left[(\widehat{y} - \overline{y})^2\right] = \frac{1}{m}\left(1 - \frac{m}{n}\right)\sum_{i=1}^n\frac{\left(y_i - \overline{y} - (\mathbf{x}_i - \overline{\mathbf{x}})^\top\widehat{\mathbf{b}}\right)^2}{n-1},$$

and the result follows. $\qquad\square$

Therefore our goal is to find $\widehat{\mathbf{b}}$ that minimizes $\|\mathbf{y} - \overline{\mathbf{y}} - (\mathbf{X} - \overline{\mathbf{X}})\widehat{\mathbf{b}}\|_2^2$. Our approach is to use leverage score sampling (Theorem 7) to find a vector $\widehat{\mathbf{b}}$ that approximately minimizes this error. A main technique in our approach is that the samples picked for leverage score sampling are independent of the samples drawn for estimation. More specifically, our estimator is based on two sets of samples. We use one set of samples $S$ to estimate $\widehat{\mathbf{b}}$ and we use the another set $S'$ of $m$ uniform samples (that is independent of $S$) in the following estimator $\widehat{y} = \frac{1}{|S'|}\sum_{i\in S'} y_i + (\overline{\mathbf{x}} - \mathbf{x}_i)^\top\widehat{\mathbf{b}}$. By Lemma 11, this is unbiased and achieves a variance of

$$\frac{1}{m(n-1)}\left(1 - \frac{m}{n}\right)\left\|(\mathbf{X} - \overline{\mathbf{X}})\widehat{\mathbf{b}} - (\mathbf{y} - \overline{\mathbf{y}})\right\|_2^2.$$

Although it is straightforward to carry this independence for the mean estimation problem, as we will see, we require a more involved algorithm to provide it for the treatment effect estimation problems.

**Algorithm 2:** (Ridge) leverage score sampling based regression adjustment

1 **Input:** $\mathbf{X} \in \mathbb{R}^{n \times d}$, $\mathbf{y} \in \mathbb{R}^n$, $m \in \mathbb{N}$, $0 < \epsilon < 1$.
2 For Theorem 10, set $\lambda = 0$, and for Theorem 12, set $\lambda = \zeta^2 \cdot n/m$      // $\zeta = \max_i \|\mathbf{X}_{i:}\|_2$
3 Compute the leverage scores $\ell_i^{(\lambda)}$ of $\widetilde{\mathbf{X}} = \begin{bmatrix} (\mathbf{X} - \overline{\mathbf{X}}) & \mathbf{1} \end{bmatrix}$.
4 Set $p_i = \min\{1, 3 \cdot \ell_i^{(\lambda)} \cdot \log(d/\delta)/\epsilon^2\}$ for $i \in [n]$.
5 Let $\mathbf{S} \in \mathbb{R}^{n \times n}$ be a random diagonal matrix with $\mathbf{S}_{ii} = 1/\sqrt{p_i}$ with probability $p_i$, and $\mathbf{S}_{ii} = 0$, otherwise.
6 Let $\widetilde{\mathbf{b}} = \arg\min_{\mathbf{b} \in \mathbb{R}^{d+1}} \left\| \mathbf{S}\widetilde{\mathbf{X}}\mathbf{b} - \mathbf{S}\mathbf{y} \right\|_2^2 + \lambda \|\mathbf{b}_{1:d}\|_2^2$, and $\widehat{\mathbf{b}} \in \mathbb{R}^d$ be a vector consisting of the first $d$ entries of $\widetilde{\mathbf{b}}$.
7 Let $S'$ be a uniformly sampled subset of size $m$ of $[n]$ and $\widehat{\tau} = \frac{1}{m} \sum_{i \in S'} y_i + (\overline{\mathbf{x}} - \mathbf{x}_i)^\top \widehat{\mathbf{b}}$
8 Return $\widehat{\tau}$.

**Theorem 10** (Leverage score sampling based regression adjustment). *Let $\epsilon, \delta \in (0, 1]$, and $m \in \mathbb{N}$ and $\mathbf{X} \in \mathbb{R}^{n \times d}$ be a covariate matrix. Then, Algorithm 2 computes an unbiased estimator of the mean of an outcome vector $\mathbf{y}$ (on a population of size $n$) with $O(d\log(d/\delta)/\epsilon^2 + m)$ samples. Moreover, there is an event $\mathcal{E}$ with $\Pr(\mathcal{E}) \geq 1 - \delta$ (over the randomness of the algorithm) such that the variance of the estimator conditioned to $\mathcal{E}$ is bounded by*

$$(1 + \epsilon)\frac{1}{m(n-1)}\left(1 - \frac{m}{n}\right)\min_{\mathbf{b}}\left\|(\mathbf{X} - \overline{\mathbf{X}})\mathbf{b} - (\mathbf{y} - \overline{\mathbf{y}})\right\|_2^2. \tag{2}$$

*Proof.* Let $\widetilde{\mathbf{X}} := \begin{bmatrix} (\mathbf{X} - \overline{\mathbf{X}}) & \mathbf{1} \end{bmatrix}$. We first show that

$$\min_{\mathbf{b} \in \mathbb{R}^d}\left\|(\mathbf{X} - \overline{\mathbf{X}})\mathbf{b} - (\mathbf{y} - \overline{\mathbf{y}})\right\|_2^2 = \min_{\mathbf{b} \in \mathbb{R}^{d+1}}\left\|\widetilde{\mathbf{X}}\mathbf{b} - \mathbf{y}\right\|_2^2. \tag{3}$$

Note that $\min_{\mathbf{b} \in \mathbb{R}^{d+1}}\left\|\widetilde{\mathbf{X}}\mathbf{b} - \mathbf{y}\right\|_2^2 = \left\|\pi_{\widetilde{\mathbf{X}}}\mathbf{y} - \mathbf{y}\right\|_2^2$. Since $\mathbf{1}$ is orthogonal to $\mathbf{X} - \overline{\mathbf{X}}$,

$$\pi_{\widetilde{\mathbf{X}}}\mathbf{y} = \pi_{(\mathbf{X}-\overline{\mathbf{X}})}\mathbf{y} + \pi_{\mathbf{1}}\mathbf{y}.$$

Moreover $\pi_{\mathbf{1}}\mathbf{y} = \overline{\mathbf{y}}$. Since $\mathbf{1}$ is orthogonal to $\mathbf{X} - \overline{\mathbf{X}}$, $\overline{\mathbf{y}} = \overline{y} \cdot \mathbf{1}$ is orthogonal to $\mathbf{X} - \overline{\mathbf{X}}$, and $\pi_{(\mathbf{X}-\overline{\mathbf{X}})}\overline{\mathbf{y}} = 0$. Therefore

$$\pi_{\widetilde{\mathbf{X}}}\mathbf{y} = \pi_{(\mathbf{X}-\overline{\mathbf{X}})}(\mathbf{y} - \overline{\mathbf{y}}) + \overline{\mathbf{y}}.$$

Subtracting both sides by $\mathbf{y}$ and taking the norm, we have

$$\left\|\pi_{\widetilde{\mathbf{X}}}\mathbf{y} - \mathbf{y}\right\|_2^2 = \left\|\pi_{(\mathbf{X}-\overline{\mathbf{X}})}(\mathbf{y} - \overline{\mathbf{y}}) - (\mathbf{y} - \overline{\mathbf{y}})\right\|_2^2 = \min_{\mathbf{b} \in \mathbb{R}^d}\left\|(\mathbf{X} - \overline{\mathbf{X}})\mathbf{b} - (\mathbf{y} - \overline{\mathbf{y}})\right\|_2^2,$$

which implies our claim. Now note that by sampling $O(d\log(d/\delta)/\epsilon^2)$ entries of $\mathbf{y}$, by Theorem 7, with high probability, we can find $\widetilde{\mathbf{b}}$ such that

$$\left\|\widetilde{\mathbf{X}}\widetilde{\mathbf{b}} - \mathbf{y}\right\|_2^2 \leq (1 + \epsilon)\min_{\mathbf{b} \in \mathbb{R}^{d+1}}\left\|\widetilde{\mathbf{X}}\mathbf{b} - \mathbf{y}\right\|_2^2 = (1 + \epsilon)\min_{\mathbf{b} \in \mathbb{R}^d}\left\|(\mathbf{X} - \overline{\mathbf{X}})\mathbf{b} - (\mathbf{y} - \overline{\mathbf{y}})\right\|_2^2. \tag{4}$$

Let $\widehat{\mathbf{b}} \in \mathbb{R}^d$ be a vector obtained by taking the first $d$ entries of $\widetilde{\mathbf{b}}$. Then, we have

$$\widetilde{\mathbf{X}}\widetilde{\mathbf{b}} - \mathbf{y} = (\mathbf{X} - \overline{\mathbf{X}})\widehat{\mathbf{b}} - \mathbf{y} + \widetilde{b}_{d+1} \cdot \mathbf{1}.$$

Now consider the following regression problem:

$$\min_{t \in \mathbb{R}}\left\|\mathbf{1} \cdot t - (\mathbf{y} - (\mathbf{X} - \overline{\mathbf{X}})\widehat{\mathbf{b}})\right\|_2^2.$$

The optimal solution is $\frac{1}{n}\mathbf{1}^\top(\mathbf{y} - (\mathbf{X} - \overline{\mathbf{X}})\widehat{\mathbf{b}}) = \frac{1}{n}\mathbf{1}^\top\mathbf{y} = \overline{y}$. Therefore

$$\left\|\widetilde{\mathbf{X}}\widetilde{\mathbf{b}} - \mathbf{y}\right\|_2^2 = \left\|(\mathbf{X} - \overline{\mathbf{X}})\widehat{\mathbf{b}} - \mathbf{y} + \widetilde{b}_{d+1} \cdot \mathbf{1}\right\|_2^2 \geq \left\|(\mathbf{X} - \overline{\mathbf{X}})\widehat{\mathbf{b}} - \mathbf{y} + \overline{\mathbf{y}}\right\|_2^2. \tag{5}$$

Thus by (4) and (5),

$$\left\|(\mathbf{X}-\overline{\mathbf{X}})\widehat{\mathbf{b}}-(\mathbf{y}-\overline{\mathbf{y}})\right\|_2^2 \le (1+\epsilon)\min_{\mathbf{b}\in\mathbb{R}^d}\left\|(\mathbf{X}-\overline{\mathbf{X}})\mathbf{b}-(\mathbf{y}-\overline{\mathbf{y}})\right\|_2^2,$$

and the result follows by Lemma 11. □

Note that the number of samples in Theorem 10 depends on the number of covariates (also called features). This dependence can be alleviated by using ridge leverage score sampling at the expense of a higher variance bound. If we want to sample $m < d\log(d/\delta)/\epsilon^2$, we can use ridge leverage score sampling, which gives a bound on the ridge regression error instead of the regression error. Using $\lambda$-ridge leverage scores, for $\lambda = \zeta^2 \cdot \frac{n}{m}$, where $\zeta$ is the maximum $\ell_2$ norm over rows of $\mathbf{X}$, gives the following result. Before providing this result, note that the block leverage score sampling approach of [12] shows that to solve

$$\min_{\mathbf{b}\in\mathbb{R}^d}\left[\|\mathbf{X}\mathbf{b}-\mathbf{y}\|_2^2 + \lambda\|\mathbf{b}_{1:d-1}\|_2^2\right],$$

we can use the same ridge leverage sampling as for $\min_{\mathbf{b}\in\mathbb{R}^d}\left[\|\mathbf{X}\mathbf{b}-\mathbf{y}\|_2^2 + \lambda\|\mathbf{b}\|_2^2\right]$, and the number of samples only needs to be increased by a constant factor. We require this for our ridge leverage score sampling-based mean estimation result.

**Theorem 12** (Ridge leverage score sampling based regression adjustment). *Let $0 < \epsilon < 1$, $0 < \delta < 0.5$, $m \in \mathbb{N}$, and $\mathbf{X}$ be a covariate matrix. Then Algorithm 2 computes an unbiased estimator of the mean of an outcome vector $\mathbf{y}$ (on a population of size $n$) by using $O(m\log(m/\delta)/\epsilon^2)$ samples. Moreover, there is an event $\mathcal{E}$ with $\Pr(\mathcal{E}) \ge 1-\delta$ (over the randomness of the algorithm) such that the variance of the estimator conditioned on $\mathcal{E}$ is bounded by*

$$(1+\epsilon)\frac{1}{m^2}\min_{\mathbf{b}\in\mathbb{R}^d}\left[\left\|(\mathbf{X}-\overline{\mathbf{X}})\mathbf{b}-(\mathbf{y}-\overline{\mathbf{y}})\right\|_2^2 + \zeta^2\|\mathbf{b}\|_2^2\right]. \tag{6}$$

*Proof.* Let $\widetilde{\mathbf{X}} := \begin{bmatrix}(\mathbf{X}-\overline{\mathbf{X}}) & \mathbf{1}\end{bmatrix}$, and

$$\mathbf{Z} = \begin{bmatrix}(\mathbf{X}-\overline{\mathbf{X}}) \\ \sqrt{\lambda}\mathbf{I}_d\end{bmatrix}, \quad \widetilde{\mathbf{Z}} = \begin{bmatrix}(\mathbf{X}-\overline{\mathbf{X}}) & \mathbf{1} \\ \sqrt{\lambda}\mathbf{I}_d & \mathbf{0}\end{bmatrix}.$$

Matrix $\widetilde{\mathbf{Z}}$ corresponds to the regression problem

$$\min_{\mathbf{b}\in\mathbb{R}^{d+1}}\left[\left\|\widetilde{\mathbf{X}}\mathbf{b}-\mathbf{y}\right\|_2^2 + \lambda\|\mathbf{b}_{1:d}\|_2^2\right].$$

In other words, defining the vector $\widetilde{\mathbf{y}} \in \mathbb{R}^{n+d}$ as a vector that is equal to $\mathbf{y}$ on the first $n$ entries and is zero on other entries, we have

$$\left\|\widetilde{\mathbf{Z}}\mathbf{b}-\widetilde{\mathbf{y}}\right\|_2^2 = \left\|\widetilde{\mathbf{X}}\mathbf{b}-\mathbf{y}\right\|_2^2 + \lambda\|\mathbf{b}_{1:d}\|_2^2,$$

for any vector $\mathbf{b} \in \mathbb{R}^{d+1}$. Now note that the solution to $\arg\min_{\mathbf{b}\in\mathbb{R}^{d+1}}\left\|\widetilde{\mathbf{Z}}\mathbf{b}-\widetilde{\mathbf{y}}\right\|_2^2$ is

$$(\widetilde{\mathbf{Z}}^\top\widetilde{\mathbf{Z}})^{-1}\widetilde{\mathbf{Z}}^\top\widetilde{\mathbf{y}} = (\widetilde{\mathbf{Z}}^\top\widetilde{\mathbf{Z}})^{-1}\widetilde{\mathbf{X}}^\top\mathbf{y},$$

by definition of $\widetilde{\mathbf{y}}$. Moreover by definition,

$$\widetilde{\mathbf{Z}}^\top\widetilde{\mathbf{Z}} = \widetilde{\mathbf{X}}^\top\widetilde{\mathbf{X}} + \lambda\begin{bmatrix}\mathbf{I}_d & \mathbf{0} \\ \mathbf{0}^\top & 0\end{bmatrix}.$$

Therefore since $\mathbf{1}^\top(\mathbf{X}-\overline{\mathbf{X}}) = \mathbf{0}^\top$, we have

$$\widetilde{\mathbf{Z}}^\top\widetilde{\mathbf{Z}} = \begin{bmatrix}(\mathbf{X}-\overline{\mathbf{X}})^\top(\mathbf{X}-\overline{\mathbf{X}}) + \lambda\mathbf{I}_d & \mathbf{0} \\ \mathbf{0}^\top & n\end{bmatrix}.$$

Since this is a block diagonal matrix

$$(\widetilde{\mathbf{Z}}^\top\widetilde{\mathbf{Z}})^{-1} = \begin{bmatrix}\left((\mathbf{X}-\overline{\mathbf{X}})^\top(\mathbf{X}-\overline{\mathbf{X}}) + \lambda\mathbf{I}_d\right)^{-1} & \mathbf{0} \\ \mathbf{0}^\top & n^{-1}\end{bmatrix}$$

Therefore
$$(\widetilde{\mathbf{Z}}^\top \widetilde{\mathbf{Z}})^{-1}\widetilde{\mathbf{Z}}^\top \widetilde{\mathbf{y}} = \begin{bmatrix} \left((\mathbf{X}-\overline{\mathbf{X}})^\top (\mathbf{X}-\overline{\mathbf{X}}) + \lambda \mathbf{I}_d\right)^{-1}(\mathbf{X}-\overline{\mathbf{X}})^\top \mathbf{y} \\ \overline{y} \end{bmatrix}.$$

Thus
$$\left\|\pi_{\widetilde{\mathbf{Z}}}\widetilde{\mathbf{y}} - \widetilde{\mathbf{y}}\right\|_2^2 = \left\|\begin{bmatrix}(\mathbf{X}-\overline{\mathbf{X}}) \\ \sqrt{\lambda}\mathbf{I}_d\end{bmatrix}\left((\mathbf{X}-\overline{\mathbf{X}})^\top(\mathbf{X}-\overline{\mathbf{X}}) + \lambda\mathbf{I}_d\right)^{-1}(\mathbf{X}-\overline{\mathbf{X}})^\top \mathbf{y} + \begin{bmatrix}\overline{\mathbf{y}} - \mathbf{y} \\ \mathbf{0}_d\end{bmatrix}\right\|_2^2.$$

Note that $\left((\mathbf{X}-\overline{\mathbf{X}})^\top(\mathbf{X}-\overline{\mathbf{X}}) + \lambda\mathbf{I}_d\right)^{-1}(\mathbf{X}-\overline{\mathbf{X}})^\top(\mathbf{y}-\overline{\mathbf{y}})$ is the solution to
$$\arg\min_{\mathbf{b}\in\mathbb{R}^d}\left[\left\|(\mathbf{X}-\overline{\mathbf{X}})\mathbf{b} - (\mathbf{y}-\overline{\mathbf{y}})\right\|_2^2 + \lambda\left\|\mathbf{b}\right\|_2^2\right],$$

and because $\mathbf{1}$ is in the null space of $\mathbf{X}-\overline{\mathbf{X}}$,
$$\left((\mathbf{X}-\overline{\mathbf{X}})^\top(\mathbf{X}-\overline{\mathbf{X}}) + \lambda\mathbf{I}_d\right)^{-1}(\mathbf{X}-\overline{\mathbf{X}})^\top(\mathbf{y}-\overline{\mathbf{y}}) = \left((\mathbf{X}-\overline{\mathbf{X}})^\top(\mathbf{X}-\overline{\mathbf{X}}) + \lambda\mathbf{I}_d\right)^{-1}(\mathbf{X}-\overline{\mathbf{X}})^\top\mathbf{y}.$$

Therefore
$$\left\|(\pi_{\widetilde{\mathbf{Z}}} - \mathbf{I})\widetilde{\mathbf{y}}\right\|_2^2 = \left\|(\pi_{\mathbf{Z}} - \mathbf{I})\begin{bmatrix}\mathbf{y} - \overline{\mathbf{y}} \\ \mathbf{0}_d\end{bmatrix}\right\|_2^2,$$

and
$$\min_{\mathbf{b}\in\mathbb{R}^d}\left[\left\|(\mathbf{X}-\overline{\mathbf{X}})\mathbf{b} - (\mathbf{y}-\overline{\mathbf{y}})\right\|_2^2 + \lambda\left\|\mathbf{b}\right\|_2^2\right] = \min_{\mathbf{b}\in\mathbb{R}^{d+1}}\left[\left\|\widetilde{\mathbf{X}}\mathbf{b} - \mathbf{y}\right\|_2^2 + \lambda\left\|\mathbf{b}_{1:d}\right\|_2^2\right]. \qquad (7)$$

Now note that by ridge leverage score sampling, we find a vector $\widetilde{\mathbf{b}}\in\mathbb{R}^{d+1}$ such that, with high probability,
$$\left\|\widetilde{\mathbf{X}}\widetilde{\mathbf{b}} - \mathbf{y}\right\|_2^2 + \lambda\left\|\widetilde{\mathbf{b}}_{1:d}\right\|_2^2 \le (1+\epsilon)\min_{\mathbf{b}\in\mathbb{R}^{d+1}}\left[\left\|\widetilde{\mathbf{X}}\mathbf{b} - \mathbf{y}\right\|_2^2 + \lambda\left\|\mathbf{b}_{1:d}\right\|_2^2\right].$$

Therefore
$$\left\|\widetilde{\mathbf{X}}\widetilde{\mathbf{b}} - \mathbf{y}\right\|_2^2 + \lambda\left\|\widetilde{\mathbf{b}}_{1:d}\right\|_2^2 \le (1+\epsilon)\min_{\mathbf{b}\in\mathbb{R}^d}\left[\left\|(\mathbf{X}-\overline{\mathbf{X}})\mathbf{b} - (\mathbf{y}-\overline{\mathbf{y}})\right\|_2^2 + \lambda\left\|\mathbf{b}\right\|_2^2\right]. \qquad (8)$$

Now let $\widehat{\mathbf{b}}\in\mathbb{R}^d$ be the vector that is equal to the first $d$ entries of $\widetilde{\mathbf{b}}$. Then we have
$$\left\|\widetilde{\mathbf{X}}\widetilde{\mathbf{b}} - \mathbf{y}\right\|_2^2 + \lambda\left\|\widetilde{\mathbf{b}}_{1:d}\right\|_2^2 = \left\|(\mathbf{X}-\overline{\mathbf{X}})\widehat{\mathbf{b}} + \widetilde{\mathbf{b}}_{d+1}\cdot\mathbf{1} - \mathbf{y}\right\|_2^2 + \lambda\left\|\widehat{\mathbf{b}}\right\|_2^2.$$

Now consider a regression problem of the following form.
$$\min_{t\in\mathbb{R}}\left\|\mathbf{1}\cdot t - (\mathbf{y} - (\mathbf{X}-\overline{\mathbf{X}})\widehat{\mathbf{b}})\right\|_2^2.$$

The optimal solution to this problem is $\frac{1}{n}\mathbf{1}^\top\left((\mathbf{y} - (\mathbf{X}-\overline{\mathbf{X}})\widehat{\mathbf{b}})\right) = \frac{1}{n}\mathbf{1}^\top\mathbf{y} = \overline{y}$. Therefore
$$\left\|\widetilde{\mathbf{X}}\widetilde{\mathbf{b}} - \mathbf{y}\right\|_2^2 = \left\|(\mathbf{X}-\overline{\mathbf{X}})\widehat{\mathbf{b}} - \mathbf{y} + \widetilde{b}_{d+1}\cdot\mathbf{1}\right\|_2^2 \ge \left\|(\mathbf{X}-\overline{\mathbf{X}})\widehat{\mathbf{b}} - \mathbf{y} + \overline{\mathbf{y}}\right\|_2^2. \qquad (9)$$

Thus by (8) and (9), and because $\lambda\left\|\widehat{\mathbf{b}}\right\|_2^2 \ge 0$,
$$\left\|(\mathbf{X}-\overline{\mathbf{X}})\widehat{\mathbf{b}} - (\mathbf{y}-\overline{\mathbf{y}})\right\|_2^2 \le (1+\epsilon)\min_{\mathbf{b}\in\mathbb{R}^d}\left[\left\|(\mathbf{X}-\overline{\mathbf{X}})\mathbf{b} - (\mathbf{y}-\overline{\mathbf{y}})\right\|_2^2 + \lambda\left\|\mathbf{b}\right\|_2^2\right].$$

Therefore by Lemma 11, the variance of the regression-adjusted estimator with $m$ samples is less than,
$$\frac{(1+\epsilon)}{mn}\min_{\mathbf{b}\in\mathbb{R}^d}\left[\left\|(\mathbf{X}-\overline{\mathbf{X}})\mathbf{b} - (\mathbf{y}-\overline{\mathbf{y}})\right\|_2^2 + \lambda\left\|\mathbf{b}\right\|_2^2\right]$$

Finally by definition $\frac{\lambda}{mn} = \frac{\zeta^2 n/m}{nm} = \frac{\zeta^2}{m^2}$, and the result follows. $\qquad\square$

We now explain previous approaches to mean estimation. The classical approach to regression adjustment [9] is to learn a vector $\widehat{\mathbf{b}}$ from the same set of samples that are used for estimation.

$$\widehat{\mathbf{b}} = \arg\min_{\mathbf{b}} \|\mathbf{y}_{S'} - \overline{\mathbf{y}}_{S'} - (\mathbf{X}_{S'} - \overline{\mathbf{X}}_{S'})\mathbf{b}\|_2^2.$$

The Gram-Schmidt walk design of Harshaw et al. [18] has originally been developed for the average treatment effect estimation and achieves the following bound using the Horvitz-Thompson estimator — see Algorithm 5.

**Theorem 13** (Gram-Schmidt walk design [18]). *Let $0 < \phi < 1$. Given $\mathbf{y}^{(1)}, \mathbf{y}^{(0)} \in \mathbb{R}^n$ and $\mathbf{X} \in \mathbb{R}^{n \times d}$, the Horvitz-Thompson estimator under the Gram-Schmidt walk design is unbiased and has a variance of*

$$\mathbb{E}\left[(\widehat{\tau} - \tau)^2\right] \leq \frac{1}{n^2} \min_{\mathbf{b} \in \mathbb{R}^n} \left[\frac{1}{\phi} \|\mathbf{X}\mathbf{b} - \boldsymbol{\mu}\|_2^2 + \frac{\zeta^2}{1 - \phi} \|\mathbf{b}\|_2^2\right],$$

*where $\boldsymbol{\mu} = \mathbf{y}^{(1)} + \mathbf{y}^{(0)}$.*

A special setting of ATE estimation problem reduces to mean estimation. Suppose $\mathbf{y}^{(1)} = \mathbf{y}$ and $\mathbf{y}^{(0)} = \mathbf{0}$. Then the average treatment effect for this problem equals the mean of $\mathbf{y}$. This approach partitions the samples into two groups. The first outcome is observed for the first group, and the second outcome is observed for the second group. The estimate is then produced by the Horvitz-Thompson estimator. Note that in our special case, only the samples in the first group amount to observation since $\mathbf{y}^{(0)}$ is the fixed vector of all zeros. This yields the following guarantee for mean estimation

**Theorem 14** (Gram-Schmidt walk based design [18]). *The Horvitz-Thompson estimator applied to the output of Algorithm 5 gives an unbiased estimator of the mean of $\mathbf{y}$ with a variance of less than or equal to*

$$\frac{1}{m^2} \min_{\mathbf{b} \in \mathbb{R}^d} \left[\|(\mathbf{X} - \overline{\mathbf{X}})\mathbf{b} - (\mathbf{y} - \overline{\mathbf{y}})\|_2^2 + \zeta^2 \|\mathbf{b}\|_2^2\right] \tag{10}$$

*with $O(m)$ samples.*

In comparison, the bound of Theorem 12 is better by a factor of $m/n$ on the $\|\mathbf{X}\mathbf{b} - \mathbf{y}\|_2^2$ term.

## B  Individual Treatment Effect Estimation

In this section, we give an ITE estimation algorithm with partial observation (Algorithm 3). Our approach is to learn a vector $\mathbf{b}$ that nearly optimally fits $\mathbf{t} := \mathbf{y}^{(1)} - \mathbf{y}^{(0)}$ linearly onto $\mathbf{X}$, i.e., $\|\mathbf{X}\mathbf{b} - \mathbf{t}\|_2$ is small. Since we do not have access to $\mathbf{t}/2$, we perform regression on a random vector for which the expectation is $\mathbf{t}$. To this end, we construct the random vector by setting $\mathbf{y}_i$ independently to either $\mathbf{y}_i^{(1)}$ or $-\mathbf{y}_i^{(0)}$ with equal probability. If we observe all samples, and use Theorem 8 on vectors $\mathbf{y}^{(1)}$ and $-\mathbf{y}^{(0)}$, this approach gives the following expected error bound.

$$\mathbb{E}\left[\|2\mathbf{X}\mathbf{b}^* - \mathbf{t}\|_2^2\right] \leq d \left\|\mathbf{y}^{(1)} + \mathbf{y}^{(0)}\right\|_\infty^2 + \min_{\mathbf{b}} \|\mathbf{X}\mathbf{b} - \mathbf{t}\|_2^2.$$

We can achieve a similar bound using leverage score sampling by thresholding the probability of sampling rows from below while we guarantee that the number of samples does not increase too much with high probability. The following is the main technical contribution of this part.

**Theorem 1** (ITE Estimation). *For any $\epsilon, \delta, \alpha \in (0, 1]$, there exists a randomized algorithm (Algorithm 3) that observes a potential outcome for $O(d \log(d/\delta)/\epsilon^2 + \alpha \cdot n)$ units and outputs a vector $\widehat{\mathbf{t}}$. Moreover there is an event $\mathcal{E}$ such that $\Pr(\mathcal{E}) \geq 1 - \delta$ (over the randomness of the algorithm) and*

$$\mathbb{E}\left[\|\widehat{\mathbf{t}} - \mathbf{t}\|_2^2 | \mathcal{E}\right] \leq (1 + \epsilon) \cdot \min_{\mathbf{b} \in \mathbb{R}^d} \|\mathbf{X}\mathbf{b} - \mathbf{t}\|_2^2 + \frac{d}{\alpha} \cdot (1 + \epsilon) \cdot \left\|\mathbf{y}^{(1)} + \mathbf{y}^{(0)}\right\|_\infty^2.$$

*Proof.* Note that $2\mathbf{y} = \mathbf{t} + \mathbf{z} \odot \boldsymbol{\mu}$. Then

$$2\mathbf{X}\widehat{\mathbf{b}} = 2\mathbf{X}(\mathbf{X}^\top \mathbf{S}\mathbf{S}\mathbf{X})^{-1}\mathbf{X}^\top \mathbf{S}\mathbf{S}\mathbf{y}$$
$$= \mathbf{X}(\mathbf{X}^\top \mathbf{S}\mathbf{S}\mathbf{X})^{-1}\mathbf{X}^\top \mathbf{S}\mathbf{S}\mathbf{t} + \mathbf{X}(\mathbf{X}^\top \mathbf{S}\mathbf{S}\mathbf{X})^{-1}\mathbf{X}^\top \mathbf{S}\mathbf{S}(\mathbf{z} \odot \boldsymbol{\mu}).$$

**Algorithm 3:** ITE estimation with random vector regression and leverage score sampling

1 **Input:** $\mathbf{X} \in \mathbb{R}^{n \times d}, \mathbf{y}^{(1)}, \mathbf{y}^{(2)} \in \mathbb{R}^n, 0 < \epsilon < 1, 0 < \alpha \le 1$.
2 Compute the leverage scores $\ell_i$ of $\mathbf{X}$.
3 Set $p_i = \min\{1, \max\{\alpha, 3 \cdot \ell_i \cdot \log(d/\delta)/\epsilon^2\}\}$ for $i \in [n]$.
4 Let $\mathbf{S} \in \mathbb{R}^{n \times n}$ be a random diagonal matrix with $\mathbf{S}_{ii} = 1/\sqrt{p_i}$ with probability $p_i$, and $\mathbf{S}_{ii} = 0$, otherwise.
5 Let $\widehat{\mathbf{b}} = \arg\min_{\mathbf{b} \in \mathbb{R}^{d+1}} \|\mathbf{SXb} - \mathbf{Sy}\|_2^2$.
6 Return $\widehat{\mathbf{t}} := 2\mathbf{X}\widehat{\mathbf{b}}$.

Moreover

$$(\mathbf{X}^\top \mathbf{SSX})^{-1} \mathbf{X}^\top \mathbf{SSt} = \arg\min_{\mathbf{b} \in \mathbb{R}^d} \|\mathbf{SXb} - \mathbf{St}\|_2^2 .$$

Therefore by Theorem 7, and since $\|\pi_\mathbf{X} \mathbf{t} - \mathbf{t}\|_2^2 = \min_\mathbf{b} \|\mathbf{Xb} - \mathbf{t}\|_2^2$,

$$\left\| \mathbf{X}(\mathbf{X}^\top \mathbf{SSX})^{-1} \mathbf{X}^\top \mathbf{SSt} - \mathbf{t} \right\|_2^2 \le (1 + \epsilon) \|\pi_\mathbf{X} \mathbf{t} - \mathbf{t}\|_2^2 .$$

Now note that

$$\mathbb{E}\left[\left\| \mathbf{X}(\mathbf{X}^\top \mathbf{SSX})^{-1} \mathbf{X}^\top \mathbf{SS}(\mathbf{z} \odot \boldsymbol{\mu}) \right\|_2^2\right] \le (1 + \epsilon)\mathbb{E}\left[(\mathbf{z} \odot \boldsymbol{\mu})^\top \mathbf{SSX}(\mathbf{X}^\top \mathbf{SSX})^{-1} \mathbf{X}^\top \mathbf{SS}(\mathbf{z} \odot \boldsymbol{\mu})\right]$$

$$= (1 + \epsilon) \sum_{i:\mathbf{S}_{ii} \ne 0} \frac{\ell_i(\mathbf{SX}) \cdot \boldsymbol{\mu}_i^2}{p_i}$$

$$\le \frac{1}{\alpha} \cdot (1 + \epsilon) \|\boldsymbol{\mu}\|_\infty^2 \sum_{i:\mathbf{S}_{ii} \ne 0} \ell_i(\mathbf{SX})$$

$$\le \frac{d}{\alpha} \cdot (1 + \epsilon) \|\boldsymbol{\mu}\|_\infty^2 ,$$

where the last inequality follows from the fact that the sum of leverage scores is less than $d$, and the penultimate inequality follows from the definition of $\mathbf{p}_i$. Finally, the result follows by noting that,

$$\mathbb{E}\left[(\mathbf{X}(\mathbf{X}^\top \mathbf{SSX})^{-1} \mathbf{X}^\top \mathbf{SSt} - \mathbf{t})^\top \mathbf{X}(\mathbf{X}^\top \mathbf{SSX})^{-1} \mathbf{X}^\top \mathbf{SS}(\mathbf{z} \odot \boldsymbol{\mu})\right] = 0.$$

$\square$

## C  ATE with Full Observation

Our approach for ATE with full observation is similar to ITE in the sense that we perform regression on a random vector to learn a solution vector $\widehat{\mathbf{b}}$. Then, for estimation, we apply regression adjustment based on $\widehat{\mathbf{b}}$ to adjust the outcomes. We first present a naive version of this approach that is biased and achieves a factor proportional to $d^2/n^2$ in one of the terms bounding the variance. We improve this to $d/n^2$ by partitioning the units into two groups and using the learned vector for each group to adjust the estimation in the other group. With this approach, our estimator is unbiased.

Before discussing our results, we discuss the other approaches in the literature. The Gram-Schmidt walk design of [18] that uses the Horvitz-Thompson estimator gives the following variance for the ATE estimation.

$$\frac{1}{n^2} \min_{\mathbf{b} \in \mathbb{R}^n} \left[\frac{1}{\phi} \|\mathbf{Xb} - \boldsymbol{\mu}\|_2^2 + \frac{\zeta^2}{1 - \phi} \|\mathbf{b}\|_2^2\right],$$

where $0 < \phi < 1$ is an input parameter. The classic regression adjustment with interaction term gives an *asymptotic* variance proportional to $\frac{1}{m}(\sigma_1^2 + 2\sigma_0^2 + \sigma_{1,0})$, where $\sigma_1^2$ is the variance of $\widetilde{y}_i^{(1)} = y_i^{(1)} - \overline{y}^{(1)} + (\mathbf{X}_i - \overline{\mathbf{x}})\mathbf{b}^{(1)}$ [24], where $\mathbf{b}^{(1)}$ is the best linear fit for $\mathbf{y}^{(1)} - \overline{\mathbf{y}}^{(1)}$ using $\mathbf{X} - \overline{\mathbf{X}}$ when $n \to \infty$ [24]. Similarly, $\sigma_1^2$ is the variance of $\widetilde{y}_i^{(0)} = y_i^{(0)} - \overline{y}^{(0)} + (\mathbf{X}_i - \overline{\mathbf{x}})\mathbf{b}^{(0)}$, and $\sigma_{1,0}$ is the covariance of $\widetilde{y}_i^{(1)} \cdot \widetilde{y}_i^{(0)}$. Note that the above mentioned bound is achieved by learning two separate vectors for regression over $y_i^{(1)} - \overline{\mathbf{y}}^{(1)}$ and $y_i^{(0)} - \overline{\mathbf{y}}^{(0)}$. Moreover, note that because these bounds are asymptotic, we need to assume a model of data generation.

For our naive approach, we first need the following technical lemma.

**Lemma 15.** *Let* $\mathbf{X} \in \mathbb{R}^{n \times d}$, $\mathbf{T} \in \mathbb{R}^{n \times n}$ *be a diagonal matrix and* $\mathbf{z} \in \{-1, +1\}^n$ *be a random vector where each entry is picked independently and uniformly at random. Then*

$$\mathbb{E}\left[\mathbf{z}^\top \mathbf{T} \pi_{\mathbf{X}} \mathbf{z} \mathbf{z}^\top \pi_{\mathbf{X}} \mathbf{T} \mathbf{z}\right] \leq (d^2 + 2d) \cdot \|\mathbf{t}\|_\infty^2,$$

*where* $\mathbf{t} \in \mathbb{R}^n$ *is the vector corresponding to the diagonal matrix* $\mathbf{T}$.

*Proof.* Let $\mathbf{M} \in \mathbb{R}^{n,n}$ and. Then

$$\mathbb{E}\left[\mathbf{z}^\top \mathbf{M} \mathbf{z} \mathbf{z}^\top \mathbf{M}^\top \mathbf{z}\right] = \mathbb{E}\left[(\sum_{i=1}^{n} \sum_{j=1}^{n} z_i z_j \mathbf{M}_{ij})^2\right]$$

$$= \mathbb{E}\left[\sum_{i=1}^{n} \sum_{j=1}^{n} \sum_{k=1}^{n} \sum_{l=1}^{n} z_i z_j \mathbf{z}_k \mathbf{z}_l \mathbf{M}_{ij} \mathbf{M}_{kl}\right]$$

$$= \sum_{i=1}^{n} \sum_{j=1}^{n} \sum_{k=1}^{n} \sum_{l=1}^{n} \mathbb{E}\left[z_i z_j \mathbf{z}_k \mathbf{z}_l \mathbf{M}_{ij} \mathbf{M}_{kl}\right].$$

Note that for any term such that there exists one of the $i, j, k, l$ that is not equal to any of the other three, the expectation is equal to zero. Therefore we need to look at terms where for each $i, j, k, l$, at least one of the other three is equal to it. Therefore we have four cases:

1. $i = j = k = l$. This gives an expectation of $\mathbf{M}_{ii}^2$. So total from these terms is $\sum_{i=1}^{n} \mathbf{M}_{ii}^2$.

2. $i = j$ and $k = l$ but $i \neq k$. This gives an expectation of $\mathbf{M}_{ii} \mathbf{M}_{kk}$. So total from these terms is $2\sum_{i=1}^{n-1} \sum_{k=i+1}^{n} \mathbf{M}_{ii} \mathbf{M}_{kk}$.

3. $i = k$, $j = l$ and $i \neq j$. This gives an expectation of $\mathbf{M}_{ij}^2$. So total from these terms is $2\sum_{i=1}^{n-1} \sum_{j=i+1}^{n} \mathbf{M}_{ij}^2$.

4. $i = l$, $j = k$, $i \neq j$. This gives an expectation of $\mathbf{M}_{ij} \mathbf{M}_{ji}$. So total from these terms is $2\sum_{i=1}^{n-1} \sum_{j=i+1}^{n} \mathbf{M}_{ij} \mathbf{M}_{ji}$.

Recall that we denote the leverage scores of matrix $\mathbf{X}$ with $\ell_i$. Therefore taking the above over our matrix $\mathbf{M} = \mathbf{T} \pi_{\mathbf{X}}$, since $\mathbf{M}_{ij} = t_i (\pi_{\mathbf{X}})_{ij}$, we have

$$\mathbb{E}\left[\mathbf{z}^\top \mathbf{M} \mathbf{z} \mathbf{z}^\top \mathbf{M}^\top \mathbf{z}\right] = \sum_{i=1}^{n} \sum_{j=1}^{n} t_i t_j (\pi_{\mathbf{X}})_{ii} (\pi_{\mathbf{X}})_{jj}$$

$$+ 2\sum_{i=1}^{n-1} \sum_{j=i+1}^{n} t_i^2 (\pi_{\mathbf{X}})_{ij}^2$$

$$+ 2\sum_{i=1}^{n-1} \sum_{j=i+1}^{n} t_i t_j (\pi_{\mathbf{X}})_{ij}^2$$

$$\leq \|\mathbf{t}\|_\infty^2 \cdot \left((\sum_{i=1}^{n} \ell_i)^2 + 2\|\pi_{\mathbf{X}}\|_F^2\right)$$

$$\leq \|\mathbf{t}\|_\infty^2 \cdot \left(d^2 + 2\mathrm{tr}(\pi_{\mathbf{X}})\right)$$

$$\leq \|\mathbf{t}\|_\infty^2 \cdot \left(d^2 + 2d\right),$$

where the last two inequalities follow from the fact that the sum of leverage scores (which are the diagonal entries of the projection matrix $\pi_{\mathbf{X}}$) is at most the rank of the matrix. $\square$

We are now equipped to analyze the ATE estimation obtained by purely using random vector regression and regression adjustment.

**Theorem 16** (ATE with full observation). *Let $\mathbf{y}^{(0)}, \mathbf{y}^{(1)} \in \mathbb{R}^n$ and $\mathbf{y} \in \mathbb{R}^n$ be a random vector such that for each $i \in [n]$, $y_i$ is independently and with equal probability either equal to $y_i^{(0)}$ or $y_i^{(1)}$. Moreover let $Z^+$ be the set of entries $i$ such that $y_i$ is equal to $y_i^{(1)}$ and $Z^- = [n] \setminus Z^+$. Let $\widehat{\mathbf{b}} = \arg\min_{\mathbf{b}} \|\mathbf{X}\mathbf{b} - \mathbf{y}\|_2$. Let $\widetilde{\mathbf{y}}^{(0)} = \mathbf{y}^{(0)} - \mathbf{X}\widehat{\mathbf{b}}$ and $\widetilde{\mathbf{y}}^{(1)} = \mathbf{y}^{(1)} - \mathbf{X}\widehat{\mathbf{b}}$. Let $\widehat{\tau}$ be the following Horvitz-Thompson estimator, i.e.,*

$$\widehat{\tau} = \frac{2}{n}\left(\sum_{i \in Z^+} \widetilde{y}_i^{(1)} - \sum_{i \in Z^-} \widetilde{y}_i^{(0)}\right).$$

*Then $|\mathbb{E}\left[\widehat{\tau} - \tau\right]| \leq \frac{d}{n}\|\mathbf{y}^{(1)} - \mathbf{y}^{(0)}\|_\infty$, and*

$$\mathbb{E}\left[(\widehat{\tau} - \tau)^2\right] \leq \frac{d^2 + 2d}{n^2} \cdot \left\|\mathbf{y}^{(1)} - \mathbf{y}^{(0)}\right\|_\infty^2 + \frac{1}{n^2} \min_{\mathbf{b}} \|\mathbf{X}\mathbf{b} - \boldsymbol{\mu}\|_2^2,$$

*where $\boldsymbol{\mu} = \mathbf{y}^{(1)} + \mathbf{y}^{(0)}$.*

*Proof.* Let $\mathbf{z} \in \{-1, +1\}^n$, where $z_i = +1$ if $i \in Z^+$, and $z_i = -1$, otherwise. Moreover let $\widetilde{\boldsymbol{\mu}} := \widetilde{\mathbf{y}}^{(0)} + \widetilde{\mathbf{y}}^{(1)} = \mathbf{y}^{(0)} + \mathbf{y}^{(1)} - 2\mathbf{X}\widehat{\mathbf{b}}$. First, note that $2\mathbf{y} = \boldsymbol{\mu} + \mathbf{z} \odot \mathbf{t}$. Therefore

$$\widetilde{\boldsymbol{\mu}} = \boldsymbol{\mu} - 2\mathbf{X}\widehat{\mathbf{b}} = \boldsymbol{\mu} - 2\pi_{\mathbf{X}}\mathbf{y} = \boldsymbol{\mu} - \pi_{\mathbf{X}}(\boldsymbol{\mu} + \mathbf{z} \odot \mathbf{t}) = (\boldsymbol{\mu} - \pi_{\mathbf{X}}\boldsymbol{\mu}) - \pi_{\mathbf{X}}(\mathbf{z} \odot \mathbf{t}).$$

Moreover

$$
\begin{aligned}
\widehat{\tau} - \tau &= \frac{1}{n}\left(\sum_{i \in Z^+} (2\widetilde{y}_i^{(1)} - y_i^{(1)} + y_i^{(0)}) - \sum_{i \in Z^-}(2\widetilde{y}_i^{(0)} + y_i^{(1)} - y_i^{(0)})\right) \qquad (11) \\
&= \frac{1}{n}\left(\sum_{i \in Z^+}(y_i^{(1)} + y_i^{(0)} - 2\mathbf{X}\widehat{\mathbf{b}}) - \sum_{i \in Z^-}(y_i^{(1)} + y_i^{(0)} - 2\mathbf{x}_i^\top \widehat{\mathbf{b}})\right) \\
&= \frac{1}{n}\widetilde{\boldsymbol{\mu}}^\top \mathbf{z}.
\end{aligned}
$$

Therefore

$$\mathbb{E}\left[\widehat{\tau} - \tau\right] = \frac{1}{n}\mathbb{E}\left[(\boldsymbol{\mu} - \pi_{\mathbf{X}}\boldsymbol{\mu})^\top \mathbf{z} - (\mathbf{z} \odot \mathbf{t})^\top \pi_{\mathbf{X}}\mathbf{z}\right]. \qquad (12)$$

Thus by the linearity of expectation and since $z_i$ is equal to $+1$ and $-1$ with an equal probability, $\mathbb{E}\left[(\boldsymbol{\mu} - \pi_{\mathbf{X}}\boldsymbol{\mu})^\top \mathbf{z}\right] = 0$. Let $\mathbf{T}$ be a diagonal matrix associated with $\mathbf{t}$. Then $\mathbf{z} \odot \mathbf{t} = \mathbf{T}\mathbf{z}$. Therefore

$$\mathbb{E}\left[\widehat{\tau} - \tau\right] = -\frac{1}{n}\mathbb{E}\left[\mathbf{z}^\top \mathbf{T}\pi_{\mathbf{X}}\mathbf{z}\right]$$

Therefore since $z_i$ and $z_j$ are independent for $i \neq j$, and $z_i^2 = 1$ with probability one.

$$\mathbb{E}\left[\widehat{\tau} - \tau\right] = -\frac{1}{n}\sum_{i=1}^n t_i (\pi_{\mathbf{X}})_{ii} = -\frac{1}{n}\sum_{i=1}^n t_i \ell_i$$

Therefore by the triangle inequality

$$|\mathbb{E}\left[\widehat{\tau} - \tau\right]| \leq \frac{1}{n}\sum_{i=1}^n |t_i \ell_i| \leq \frac{\|\mathbf{t}\|_\infty}{n}\sum_{i=1}^n |\ell_i|.$$

Then the first part of the theorem follows by the fact that the sum of leverage scores is less than $d$, and noting that by definition, $\mathbf{t} = \mathbf{y}^{(1)} - \mathbf{y}^{(0)}$.

We now prove the second part of the theorem. First, note that since $z_i$ and $z_j$ are independent for $i \neq j$ and $z_i = \pm 1$ with equal probability,

$$\mathbb{E}\left[(\boldsymbol{\mu} - \pi_{\mathbf{X}}\boldsymbol{\mu})^\top \mathbf{z}\mathbf{z}^\top \mathbf{T}\pi_{\mathbf{X}}\mathbf{z}\right] = \mathbb{E}\left[(\boldsymbol{\mu} - \pi_{\mathbf{X}}\boldsymbol{\mu})^\top \mathbf{T}\pi_{\mathbf{X}}\mathbf{z}\right] = 0.$$

Therefore by (11) and Lemma 15,

$$\mathbb{E}\left[(\widehat{\tau} - \tau)^2\right] = \frac{1}{n^2}\mathbb{E}\left[\mathbf{z}^\top(\boldsymbol{\mu} - \pi_\mathbf{X}\boldsymbol{\mu})(\boldsymbol{\mu} - \pi_\mathbf{X}\boldsymbol{\mu})^\top\mathbf{z}\right]$$

$$+ \frac{1}{n^2}\mathbb{E}\left[\mathbf{z}^\top\mathbf{T}\pi_\mathbf{X}\mathbf{z}\mathbf{z}^\top\pi_\mathbf{X}\mathbf{T}\mathbf{z}\right]$$

$$\leq \frac{1}{n^2}\left(\|\boldsymbol{\mu} - \pi_\mathbf{X}\boldsymbol{\mu}\|_2^2 + (d^2 + 2d)\cdot\|\mathbf{t}\|_\infty^2\right).$$

$\square$

The next step is two use a subset of samples for regression and the rest of the samples for estimating the ATE.

Since we want to have a guarantee on the regression error, we need to make sure that the sampling (i.e., partitioning) to two groups is done according to leverage scores. However, if the leverage score of a row is greater than half, then it cannot have a probability larger than its leverage score for being assigned to each group. To reduce this probability, we use the ridge leverage score. The following lemma states that by picking a large enough ridge parameter, we can achieve a small enough ridge leverage score for all rows (therefore, small enough sampling probabilities). This is at the expense of a regularization term of the form $\lambda \cdot \|\mathbf{b}\|_2^2$ in the regression error.

**Theorem 6.** *Let $\zeta := \max_{i\in[n]}\|\mathbf{x}_i\|_2$. Then for $\lambda \geq c \cdot \zeta^2$, for $c \geq 1$, $\ell_i^{(\lambda)}(\mathbf{X}) \leq \frac{1}{c}$ for all $i \in [n]$.*

*Proof.* Recall that $\zeta := \max_{i\in[n]}\|\mathbf{x}_i\|_2$. Therefore, $\|\mathbf{x}_i\|^2 \leq \zeta^2$ for all $i \in [n]$. We have

$$\ell_i^{(\lambda)}(\mathbf{X}) = \mathbf{x}_i^\top(\mathbf{X}^\top\mathbf{X} + \lambda\mathbf{I})^{-1}\mathbf{x}_i \leq \mathbf{x}_i^\top(\lambda\mathbf{I})^{-1}\mathbf{x}_i = \|\mathbf{x}_i\|^2/\lambda \leq 1/c.$$

$\square$

Equipped with this, we present the main result of this section.

**Theorem 2** (ATE Estimation – Full Observation). *For any $\epsilon, \delta \in (0, 1]$, there exists a randomized algorithm (Algorithm 1) that computes an unbiased estimate $\widehat{\tau}$ of the ATE $\tau$, and for which there is an event $\mathcal{E}$ with $\Pr(\mathcal{E}) \geq 1 - \delta$ (over the randomness of the algorithm) such that*

$$\mathbb{E}\left[(\widehat{\tau} - \tau)^2|\mathcal{E}\right] \leq \frac{8(1+\epsilon)}{n^2}\min_{\mathbf{b}\in\mathbb{R}^d}\left(\|\mathbf{X}\mathbf{b} - \boldsymbol{\mu}\|_2^2 + 100\log(n/\delta)\cdot\zeta^2\cdot\|\mathbf{b}\|_2^2\right) + \frac{32d}{n^2}\cdot\|\mathbf{y}^{(1)} - \mathbf{y}^{(0)}\|_\infty^2,$$

*where $\zeta := \max_{i\in[n]}\|\mathbf{x}_i\|_2$ and $\boldsymbol{\mu} := \mathbf{y}^{(0)} + \mathbf{y}^{(1)}$ is the total outcome vector.*

*Proof.* For $S \subseteq [n]$ and $\mathbf{z} \in \{-1, +1\}^n$ (where $Z^+ = \{i : z_i = +1\}$ and $Z^- = \{i : z_i = -1\}$), we define

$$\tau_S := \frac{1}{|S|}\cdot\sum_{i\in S}(y_i^{(1)} - y_i^{(0)}), \text{ and } \widehat{\tau}_S := \frac{2}{|S|}\cdot\left(\sum_{i\in Z^+\cap S}\widetilde{y}_i^{(2,1)} - \sum_{i\in Z^-\cap S}\widetilde{y}_i^{(2,0)}\right).$$

Similarly, define $\widehat{\tau}_{\overline{S}}$. Then we have,

$$\mathbb{E}_\mathbf{z}\left[\mathbb{E}_S\left[(\widehat{\tau} - \tau)^2\right]\right] = \mathbb{E}_\mathbf{z}\left[\mathbb{E}_S\left[\left(\frac{|S|\cdot\widehat{\tau}_S + |\overline{S}|\cdot\widehat{\tau}_{\overline{S}}}{n} - \frac{|S|\cdot\tau_S + |\overline{S}|\cdot\tau_{\overline{S}}}{n}\right)^2\right]\right]$$

$$= \mathbb{E}_S\left[\frac{|\overline{S}|^2}{n^2}\cdot\mathbb{E}_\mathbf{z}\left[(\widehat{\tau}_{\overline{S}} - \tau_{\overline{S}})^2\right]\right]$$

$$+ \mathbb{E}_S\left[\frac{|S|^2}{n^2}\cdot\mathbb{E}_\mathbf{z}\left[(\widehat{\tau}_S - \tau_S)^2\right]\right]$$

$$+ \mathbb{E}_S\left[2\cdot\frac{|S|\cdot|\overline{S}|}{n^2}\cdot\mathbb{E}_\mathbf{z}\left[(\widehat{\tau}_{\overline{S}} - \tau_{\overline{S}})\cdot(\widehat{\tau}_S - \tau_S)\right]\right].$$

We now bound all of the terms above. Let

$$\widetilde{\boldsymbol{\mu}}^{(1)} := \widetilde{\mathbf{y}}^{(1,0)} + \widetilde{\mathbf{y}}^{(1,1)} = \mathbf{y}^{(0)} + \mathbf{y}^{(1)} - 2\mathbf{X}\widehat{\mathbf{b}}^{(1)}$$

$$\widetilde{\boldsymbol{\mu}}^{(2)} := \widetilde{\mathbf{y}}^{(2,0)} + \widetilde{\mathbf{y}}^{(2,1)} = \mathbf{y}^{(0)} + \mathbf{y}^{(1)} - 2\mathbf{X}\widehat{\mathbf{b}}^{(2)}$$

Note that

$$\widehat{\tau}_{\overline{S}} - \tau_{\overline{S}} = \frac{1}{|\overline{S}|} \cdot \left( \sum_{i \in Z^+ \cap \overline{S}} (2\widetilde{y}_i^{(1,1)} - y_i^{(1)} + y_i^{(0)}) - \sum_{i \in Z^- \cap \overline{S}} (2\widetilde{y}_i^{(1,0)} + y_i^{(1)} - y_i^{(0)}) \right)$$

$$= \frac{1}{|\overline{S}|} \cdot \left( \sum_{i \in Z^+ \cap \overline{S}} (y_i^{(1)} + y_i^{(0)} - 2\mathbf{x}_i^\top \widehat{\mathbf{b}}^{(1)}) - \sum_{i \in Z^- \cap \overline{S}} (y_i^{(1)} + y_i^{(0)} - 2\mathbf{x}_i^\top \widehat{\mathbf{b}}^{(1)}) \right)$$

$$= \frac{1}{|\overline{S}|} \cdot (\widetilde{\boldsymbol{\mu}}_{\overline{S}}^{(1)})^\top \mathbf{z}_{\overline{S}}.$$

Similarly, we have

$$\widehat{\tau}_S - \tau_S = \frac{1}{|S|} \cdot (\widetilde{\boldsymbol{\mu}}_S^{(2)})^\top \mathbf{z}_S$$

Therefore

$$\frac{|S| \cdot |\overline{S}|}{n^2} \cdot \mathbb{E}_{\mathbf{z}} \left[ (\widehat{\tau}_{\overline{S}} - \tau_{\overline{S}}) \cdot (\widehat{\tau}_S - \tau_S) \right] = \frac{1}{n^2} \cdot \mathbb{E}_{\mathbf{z}} \left[ \mathbf{z}_{\overline{S}}^\top \widetilde{\boldsymbol{\mu}}_{\overline{S}}^{(1)} (\widetilde{\boldsymbol{\mu}}_S^{(2)})^\top \mathbf{z}_S \right]$$

$$\leq \frac{1}{n^2} \cdot \left( \mathbb{E}_{\mathbf{z}} \left[ \mathbf{z}_{\overline{S}}^\top \widetilde{\boldsymbol{\mu}}_{\overline{S}}^{(1)} (\widetilde{\boldsymbol{\mu}}_{\overline{S}}^{(1)})^\top \mathbf{z}_{\overline{S}} \right] \cdot \mathbb{E}_{\mathbf{z}} \left[ \mathbf{z}_S^\top \widetilde{\boldsymbol{\mu}}_S^{(2)} (\widetilde{\boldsymbol{\mu}}_S^{(2)})^\top \mathbf{z}_S \right] \right)^{1/2},$$

where the inequality follows from Cauchy-Schwarz inequality. Moreover

$$\frac{|\overline{S}|^2}{n^2} \cdot \mathbb{E}_{\mathbf{z}} \left[ (\widehat{\tau}_{\overline{S}} - \tau_{\overline{S}})^2 \right] = \frac{1}{n^2} \cdot \mathbb{E}_{\mathbf{z}} \left[ \mathbf{z}_{\overline{S}}^\top \widetilde{\boldsymbol{\mu}}_{\overline{S}}^{(1)} (\widetilde{\boldsymbol{\mu}}_{\overline{S}}^{(1)})^\top \mathbf{z}_{\overline{S}} \right], \text{ and}$$

$$\frac{|S|^2}{n^2} \cdot \mathbb{E}_{\mathbf{z}} \left[ (\widehat{\tau}_S - \tau_S)^2 \right] = \frac{1}{n^2} \cdot \mathbb{E}_{\mathbf{z}} \left[ \mathbf{z}_S^\top \widetilde{\boldsymbol{\mu}}_S^{(2)} (\widetilde{\boldsymbol{\mu}}_S^{(2)})^\top \mathbf{z}_S \right]$$

Therefore we only need to bound $\mathbb{E}_{\mathbf{z}} \left[ \mathbf{z}_S^\top \widetilde{\boldsymbol{\mu}}_S^{(2)} (\widetilde{\boldsymbol{\mu}}_S^{(2)})^\top \mathbf{z}_S \right]$ and $\mathbb{E}_{\mathbf{z}} \left[ \mathbf{z}_{\overline{S}}^\top \widetilde{\boldsymbol{\mu}}_{\overline{S}}^{(1)} (\widetilde{\boldsymbol{\mu}}_{\overline{S}}^{(1)})^\top \mathbf{z}_{\overline{S}} \right]$. Since $S$ and $\overline{S}$ are disjoint, $\widetilde{\boldsymbol{\mu}}_{\overline{S}}^{(1)}$ is independent of $\mathbf{z}_{\overline{S}}$. Moreover, the entries of $\mathbf{z}$ are independent from each other. Therefore, since $2\mathbf{y} = \boldsymbol{\mu} + \mathbf{z} \odot \mathbf{t}$,

$$\mathbb{E}_{\mathbf{z}} \left[ \mathbf{z}_{\overline{S}}^\top \widetilde{\boldsymbol{\mu}}_{\overline{S}}^{(1)} (\widetilde{\boldsymbol{\mu}}_{\overline{S}}^{(1)})^\top \mathbf{z}_{\overline{S}} \right] = \mathbb{E}_{\mathbf{z}} \left[ \sum_{i \in \overline{S}} (\widetilde{\mu}_i^{(1)})^2 \right]$$

$$= \mathbb{E}_{\mathbf{z}} \left[ \left\| (\boldsymbol{\mu} - \mathbf{X}(\mathbf{X}^\top \mathbf{SSX} + \lambda \cdot \mathbf{I})^{-1} \mathbf{X}^\top \mathbf{SS}(\boldsymbol{\mu} + \mathbf{z} \odot \mathbf{t}))_{\overline{S}} \right\|_2^2 \right],$$

where $\lambda = 100\zeta^2 \log(n)$. We have

$$\mathbb{E}_{\mathbf{z}} \left[ \left\| (\boldsymbol{\mu} - \mathbf{X}(\mathbf{X}^\top \mathbf{SSX} + \lambda \cdot \mathbf{I})^{-1} \mathbf{X}^\top \mathbf{SS}(\boldsymbol{\mu} + \mathbf{z} \odot \mathbf{t}))_{\overline{S}} \right\|_2^2 \right]$$

$$\leq 2\mathbb{E}_{\mathbf{z}} \left[ \left\| (\boldsymbol{\mu} - \mathbf{X}(\mathbf{X}^\top \mathbf{SSX} + \lambda \cdot \mathbf{I})^{-1} \mathbf{X}^\top \mathbf{SS}\boldsymbol{\mu})_{\overline{S}} \right\|_2^2 \right]$$

$$+ 2\mathbb{E}_{\mathbf{z}} \left[ \left\| (\mathbf{X}(\mathbf{X}^\top \mathbf{SSX} + \lambda \cdot \mathbf{I})^{-1} \mathbf{X}^\top \mathbf{SSTz})_{\overline{S}} \right\|_2^2 \right].$$

Note that due to guarantees of leverage score sampling,

$$\left\| (\boldsymbol{\mu} - \mathbf{X}(\mathbf{X}^\top \mathbf{SSX} + \lambda \cdot \mathbf{I})^{-1} \mathbf{X}^\top \mathbf{SS}\boldsymbol{\mu})_{\overline{S}} \right\|_2^2 \leq \left\| \boldsymbol{\mu} - \mathbf{X}(\mathbf{X}^\top \mathbf{SSX} + \lambda \cdot \mathbf{I})^{-1} \mathbf{X}^\top \mathbf{SS}\boldsymbol{\mu} \right\|_2^2$$

$$\leq (1 + \epsilon) \min_{\mathbf{b}} \left( \left\| \mathbf{Xb} - \boldsymbol{\mu} \right\| + \lambda \left\| \mathbf{b} \right\|_2^2 \right)$$

Moreover

$$\mathbb{E}_{\mathbf{z}} \left[ \left\| (\mathbf{X}(\mathbf{X}^\top \mathbf{SSX} + \lambda \cdot \mathbf{I})^{-1} \mathbf{X}^\top \mathbf{SSTz})_{\overline{S}} \right\|_2^2 \right] \leq \mathbb{E}_{\mathbf{z}} \left[ \left\| \mathbf{X}(\mathbf{X}^\top \mathbf{SSX} + \lambda \cdot \mathbf{I})^{-1} \mathbf{X}^\top \mathbf{SSTz} \right\|_2^2 \right].$$

Since

$$\mathbf{X}^\top \mathbf{X} \preceq (1 + \epsilon) \mathbf{X}^\top \mathbf{SSX} + \lambda \cdot \mathbf{I},$$

we have

$$\mathbb{E}_{\mathbf{z}}\left[\left\|\mathbf{X}(\mathbf{X}^\top\mathbf{S}\mathbf{S}\mathbf{X}+\lambda\cdot\mathbf{I})^{-1}\mathbf{X}^\top\mathbf{S}\mathbf{S}\mathbf{T}\mathbf{z}\right\|_2^2\right]\leq\mathbb{E}_{\mathbf{z}}\left[\mathbf{z}^\top\mathbf{T}\mathbf{S}\mathbf{S}\mathbf{X}(\mathbf{X}^\top\mathbf{S}\mathbf{S}\mathbf{X}+\lambda\cdot\mathbf{I})^{-1}\mathbf{X}^\top\mathbf{S}\mathbf{S}\mathbf{T}\mathbf{z}\right]$$

$$=\sum_{i\in S}t_i^2\mathbf{S}_{ii}^2(\mathbf{S}\mathbf{X}(\mathbf{X}^\top\mathbf{S}\mathbf{S}\mathbf{X}+\lambda\cdot\mathbf{I})^{-1}\mathbf{X}^\top\mathbf{S})_{ii}^2$$

$$\leq 4\left\|\mathbf{t}\right\|_\infty^2\cdot d.$$

Therefore

$$\mathbb{E}_{\mathbf{z}}\left[\mathbf{z}_{\overline{S}}^\top\widetilde{\boldsymbol{\mu}}_{\overline{S}}^{(1)}(\widetilde{\boldsymbol{\mu}}_{\overline{S}}^{(1)})^\top\mathbf{z}_{\overline{S}}\right]\leq 8\cdot\left\|\mathbf{t}\right\|_\infty^2\cdot d+2\cdot(1+\epsilon)\min_{\mathbf{b}}\left(\left\|\mathbf{X}\mathbf{b}-\boldsymbol{\mu}\right\|+\lambda\left\|\mathbf{b}\right\|_2^2\right)$$

With a similar argument, one can show that

$$\mathbb{E}_{\mathbf{z}}\left[\mathbf{z}_S^\top\widetilde{\boldsymbol{\mu}}_S^{(2)}(\widetilde{\boldsymbol{\mu}}_S^{(2)})^\top\mathbf{z}_S\right]\leq 8\cdot\left\|\mathbf{t}\right\|_\infty^2\cdot d+2\cdot(1+\epsilon)\min_{\mathbf{b}}\left(\left\|\mathbf{X}\mathbf{b}-\boldsymbol{\mu}\right\|+\lambda\left\|\mathbf{b}\right\|_2^2\right).$$

Thus

$$\mathbb{E}_{\mathbf{z}}\left[\mathbb{E}_S\left[(\widehat{\tau}-\tau)^2\right]\right]\leq\frac{32d}{n^2}\cdot\left\|\mathbf{y}^{(1)}-\mathbf{y}^{(0)}\right\|_\infty^2+\frac{8(1+\epsilon)}{n^2}\min_{\mathbf{b}}\left(\left\|\mathbf{X}\mathbf{b}-\boldsymbol{\mu}\right\|_2^2+100\log(n)\zeta^2\cdot\left\|\mathbf{b}\right\|_2^2\right).$$

$$\square$$

We can get a high probability bound for the above result using the Hanson-Wright inequality [31]. Adjusting the vectors $\mathbf{y}^{(1)}$ and $\mathbf{y}^{(0)}$ using $\mathbf{X}\mathbf{b}^*$ as $\widetilde{\mathbf{y}}^{(0)}=\mathbf{y}^{(0)}-\mathbf{X}\widehat{\mathbf{b}}$ and $\widetilde{\mathbf{y}}^{(1)}=\mathbf{y}^{(1)}-\mathbf{X}\widehat{\mathbf{b}}$, and using the Horvitz-Thompson estimator on the adjusted outcomes $\widetilde{\mathbf{y}}^{(0)}$ and $\widetilde{\mathbf{y}}^{(1)}$, we achieve the following result.

### C.1 Unconditional Bound for ATE

The only reason the $1-\delta$ probability appears in our bound for the variance of ATE estimation (Theorem 2) is that leverage score sampling only gives a high probability guarantee for the regression error. Moreover, our variance only depends on the regression error, and the estimator is always unbiased, even if the leverage score sampling does not give the $(1+\epsilon)$ approximation guarantee on the regression. This "bad event" happens with probability at most $\delta$. Therefore, it is enough to bound the regression error when the bad event happens. Here we present such an approach. However, to keep our argument simple, we just work with a single solution vector. To actually apply this to our ATE estimation approach, one needs to consider both solution vectors for groups $S$ and $\overline{S}$ and consider the bad event for both. Such an argument follows similar to the following and provides the same result (up to constant factors). This approach essentially can make all of our results unconditional (i.e., remove the $1-\delta$ probability) at the expense of increasing the variance/error bounds by a constant factor and adding an additive term proportional to $\delta$.

If $\mathbf{X}^\top\mathbf{S}^\top\mathbf{S}\mathbf{X}$ is not a spectral approximation of $\mathbf{X}^\top\mathbf{X}$, i.e., $(1-\epsilon)\mathbf{X}^\top\mathbf{X}\preceq\mathbf{X}^\top\mathbf{S}^\top\mathbf{S}\mathbf{X}\preceq(1+\epsilon)\mathbf{X}^\top\mathbf{X}$ does not hold (this can be checked since we have access to matrix $\mathbf{X}$ and only happens with probability at most $\delta$), we just set $\mathbf{b}=0$. In this case, $\|\mathbf{X}\mathbf{b}-\boldsymbol{\mu}\|_2^2+\lambda\|\mathbf{b}\|_2^2=\|\boldsymbol{\mu}\|_2^2$. Otherwise (i.e., $(1-\epsilon)\mathbf{X}^\top\mathbf{X}\preceq\mathbf{X}^\top\mathbf{S}^\top\mathbf{S}\mathbf{X}\preceq(1+\epsilon)\mathbf{X}^\top\mathbf{X}$) let $\widehat{\mathbf{X}}\in\mathbb{R}^{(n+d)\times d}$ be the matrix $\mathbf{X}$ concatenated with factor $\lambda$ of the identity matrix and $\widehat{\boldsymbol{\mu}}\in\mathbb{R}^{n+d}$ be the vector $\boldsymbol{\mu}$ concatenated with a zero vector and

$$\mathbf{b}^*:=\arg\min_{\mathbf{b}}\|\widehat{\mathbf{X}}\mathbf{b}-\widehat{\boldsymbol{\mu}}\|_2^2=\arg\min_{\mathbf{b}}\|\mathbf{X}\mathbf{b}-\boldsymbol{\mu}\|_2^2+\lambda\|\mathbf{b}\|_2^2.$$

Let $\widetilde{\mathbf{X}}\in\mathbb{R}^{(n+d)\times d}$ be the matrix $\mathbf{S}\mathbf{X}$ concatenated with the factor $\lambda$ of the identity matrix and $\widetilde{\boldsymbol{\mu}}\in\mathbb{R}^{(n+d)}$ be the vector $\mathbf{S}\boldsymbol{\mu}$ concatenated with a zero vector and

$$\widetilde{\mathbf{b}}:=\arg\min_{\mathbf{b}}\|\widetilde{\mathbf{X}}\mathbf{b}-\widetilde{\boldsymbol{\mu}}\|_2^2=\arg\min_{\mathbf{b}}\|\mathbf{S}\mathbf{X}\mathbf{b}-\mathbf{S}\boldsymbol{\mu}\|_2^2+\lambda\|\mathbf{b}\|_2^2.$$

Then since $\widetilde{\mathbf{X}}$ is a spectral approximation of $\widehat{\mathbf{X}}$ (i.e., $(1-\epsilon)\widehat{\mathbf{X}}^\top\widehat{\mathbf{X}}\preceq\widetilde{\mathbf{X}}^\top\widetilde{\mathbf{X}}\preceq(1+\epsilon)\widehat{\mathbf{X}}^\top\widehat{\mathbf{X}}$),

$$\|\widehat{\mathbf{X}}\widetilde{\mathbf{b}}-\widehat{\boldsymbol{\mu}}\|_2\leq\|\widehat{\mathbf{X}}\mathbf{b}^*-\widehat{\boldsymbol{\mu}}\|_2+\|\widehat{\mathbf{X}}\widetilde{\mathbf{b}}-\widehat{\mathbf{X}}\mathbf{b}^*\|_2\leq\|\widehat{\mathbf{X}}\mathbf{b}^*-\widehat{\boldsymbol{\mu}}\|_2+\frac{1}{1-\epsilon}\|\widetilde{\mathbf{X}}\widetilde{\mathbf{b}}-\widetilde{\mathbf{X}}\mathbf{b}^*\|_2,\quad(13)$$

where the first inequality follows by triangle inequality. Moreover, by triangle inequality and optimality of $\widetilde{\mathbf{b}}$

$$\|\widetilde{\mathbf{X}}\widetilde{\mathbf{b}} - \widetilde{\mathbf{X}}\mathbf{b}^*\|_2 \le \|\widetilde{\mathbf{X}}\widetilde{\mathbf{b}} - \widetilde{\boldsymbol{\mu}}\|_2 + \|\widetilde{\mathbf{X}}\mathbf{b}^* - \widetilde{\boldsymbol{\mu}}\|_2 \le 2 \cdot \|\widetilde{\mathbf{X}}\mathbf{b}^* - \widetilde{\boldsymbol{\mu}}\|_2. \tag{14}$$

Now let $\mathcal{E}_1$ be the event that $\mathbf{X}^\top \mathbf{S}^\top \mathbf{S} \mathbf{X}$ is not a spectral approximation of $\mathbf{X}^\top \mathbf{X}$. Let $\mathcal{E}_2$ be the event that $\mathbf{X}^\top \mathbf{S}^\top \mathbf{S} \mathbf{X}$ is a spectral approximation of $\mathbf{X}^\top \mathbf{X}$, but $\|\mathbf{S}\mathbf{X}\mathbf{b} - \mathbf{S}\boldsymbol{\mu}\|_2^2 + \lambda\|\mathbf{b}\|_2^2 > (1+\epsilon) \cdot (\|\mathbf{X}\mathbf{b} - \boldsymbol{\mu}\|_2^2 + \lambda\|\mathbf{b}\|_2^2)$. Finally let $\mathcal{E}_3$ be the event that $\mathbf{X}^\top \mathbf{S}^\top \mathbf{S} \mathbf{X}$ is a spectral approximation of $\mathbf{X}^\top \mathbf{X}$, but $\|\mathbf{S}\mathbf{X}\mathbf{b} - \mathbf{S}\boldsymbol{\mu}\|_2^2 + \lambda\|\mathbf{b}\|_2^2 \le (1+\epsilon) \cdot (\|\mathbf{X}\mathbf{b} - \boldsymbol{\mu}\|_2^2 + \lambda\|\mathbf{b}\|_2^2)$. Then, by the law of total expectation,

$$\mathbb{E}[(\widehat{\tau} - \tau)^2] = \mathbb{E}[(\widehat{\tau} - \tau)^2 | \mathcal{E}_1] \cdot \mathbb{P}[\mathcal{E}_1] + \mathbb{E}[(\widehat{\tau} - \tau)^2 | \mathcal{E}_2] \cdot \mathbb{P}[\mathcal{E}_2] + \mathbb{E}[(\widehat{\tau} - \tau)^2 | \mathcal{E}_3] \cdot \mathbb{P}[\mathcal{E}_3]$$

Note that $\mathbb{P}[\mathcal{E}_1], \mathbb{P}[\mathcal{E}_2] \le \delta$ and therefore by the above argument and Theorem 2,

$$\mathbb{E}[(\widehat{\tau} - \tau)^2] \le \delta \cdot \|\boldsymbol{\mu}\|_2^2 + \delta \cdot \mathbb{E}[(\widehat{\tau} - \tau)^2 | \mathcal{E}_2] \tag{15}$$
$$+ (1-\delta) \left( \frac{8(1+\epsilon)}{n^2} \min_{\mathbf{b} \in \mathbb{R}^d} \left( \|\mathbf{X}\mathbf{b} - \boldsymbol{\mu}\|_2^2 + 100 \log(n/\delta) \cdot \zeta^2 \cdot \|\mathbf{b}\|_2^2 \right) + \frac{32d}{n^2} \cdot \|\mathbf{y}^{(1)} - \mathbf{y}^{(0)}\|_\infty^2 \right)$$

Now note that

$$\mathbb{E}[(\widehat{\tau} - \tau)^2 | \mathcal{E}_2]$$
$$\le \mathbb{E}[\frac{8(1+\epsilon)}{n^2} \cdot \left( \left\| \mathbf{X}\widetilde{\mathbf{b}} - \boldsymbol{\mu} \right\|_2^2 + 100 \log(n/\delta) \cdot \zeta^2 \cdot \left\| \widetilde{\mathbf{b}} \right\|_2^2 \right) + \frac{32d}{n^2} \cdot \|\mathbf{y}^{(1)} - \mathbf{y}^{(0)}\|_\infty^2 | \mathcal{E}_2]$$
$$\le \frac{1}{\mathbb{P}[\mathcal{E}_2]} \cdot \mathbb{E}[\frac{8(1+\epsilon)}{n^2} \cdot \left( \left\| \mathbf{X}\widetilde{\mathbf{b}} - \boldsymbol{\mu} \right\|_2^2 + 100 \log(n/\delta) \cdot \zeta^2 \cdot \left\| \widetilde{\mathbf{b}} \right\|_2^2 \right) + \frac{32d}{n^2} \cdot \|\mathbf{y}^{(1)} - \mathbf{y}^{(0)}\|_\infty^2]$$
$$\le \frac{1}{\mathbb{P}[\mathcal{E}_2]} \cdot \mathbb{E}[\frac{8(1+\epsilon)}{n^2} \cdot \left( \|\mathbf{X}\mathbf{b}^* - \boldsymbol{\mu}\|_2^2 + 100 \log(\frac{n}{\delta}) \cdot \zeta^2 \cdot \|\mathbf{b}^*\|_2^2 \right) + \frac{32d}{n^2} \cdot \|\mathbf{y}^{(1)} - \mathbf{y}^{(0)}\|_\infty^2]$$
$$+ \frac{1}{\mathbb{P}[\mathcal{E}_2]} \cdot \mathbb{E}[\frac{8(1+\epsilon)}{n^2} \cdot \frac{2}{1-\epsilon} \cdot \left( \|\mathbf{S}\mathbf{X}\mathbf{b}^* - \mathbf{S}\boldsymbol{\mu}\|_2^2 + 100 \log(\frac{n}{\delta}) \cdot \zeta^2 \cdot \|\mathbf{b}^*\|_2^2 \right)],$$

where the last inequality follows from (13) and (14). Now since leverage score sampling preserves the norms in expectation, we have $\mathbb{E}\left[ \|\widetilde{\mathbf{X}}\mathbf{b}^* - \widetilde{\boldsymbol{\mu}}\|_2 \right] = \|\widehat{\mathbf{X}}\mathbf{b}^* - \widehat{\boldsymbol{\mu}}\|_2$. Therefore

$$\mathbb{E}[(\widehat{\tau} - \tau)^2 | \mathcal{E}_2] \cdot \mathbb{P}[\mathcal{E}_2]$$
$$\le \frac{8(1+\epsilon)}{n^2} \cdot (1 + \frac{2}{1-\epsilon}) \cdot \left( \|\mathbf{S}\mathbf{X}\mathbf{b}^* - \mathbf{S}\boldsymbol{\mu}\|_2^2 + 100 \log(\frac{n}{\delta}) \cdot \zeta^2 \cdot \|\mathbf{b}^*\|_2^2 \right)$$
$$+ \frac{32d}{n^2} \cdot \|\mathbf{y}^{(1)} - \mathbf{y}^{(0)}\|_\infty^2,$$

Combining this with (15), we have

$$\mathbb{E}[(\widehat{\tau} - \tau)^2]$$
$$\le \left( \frac{64d}{n^2} \cdot \|\mathbf{y}^{(1)} - \mathbf{y}^{(0)}\|_\infty^2 + \frac{8(2 + \frac{2}{1-\epsilon})}{n^2} \min_b \left( \|\mathbf{X}\mathbf{b} - \boldsymbol{\mu}\|_2^2 + 100 \log(n/\delta) \cdot \zeta^2 \cdot \|\mathbf{b}\|_2^2 \right) \right)$$
$$+ \delta \cdot \|\boldsymbol{\mu}\|_2^2.$$

# D ATE with Partial Observation

A simple approach for estimating ATE using partial observations is to first uniformly sample a subset of size $m$ of the population and then apply any of the methods for ATE with full observation to this subset. If we observe the outcomes of the samples in the subset independently and with equal probability and apply the Horvitz-Thompson estimator, we achieve an squared error of less than $\frac{\|\boldsymbol{\mu}\|_2^2}{mn} + \frac{\|\mathbf{t} - \overline{\mathbf{t}}\|_2^2}{m} \left( 1 - \frac{m}{n} \right)$. If we use the Gram-Schmidt walk design and the Horvitz-Thompson estimator on the uniformly sampled subset, then we achieve an unbiased estimator with a variance of less than

$$\mathbb{E}\left[ (\widehat{\tau} - \tau)^2 \right] \le \frac{1}{m^2} \min_{b \in \mathbb{R}^d} \left[ \frac{m}{n} \cdot \frac{1}{\phi} \|\mathbf{X}\mathbf{b} - \boldsymbol{\mu}\|_2^2 + \frac{1}{1-\phi} \zeta^2 \|\mathbf{b}\|_2^2 \right] + \frac{\|\mathbf{t} - \overline{\mathbf{t}}\|_2^2}{m} \left( 1 - \frac{m}{n} \right).$$

Our approach is to adjust the Horvitz-Thompson estimator itself. We call this partial observation regression adjusted Horvitz-Thompson (RAHT) estimator. The first result we prove in this section is independent of the algorithm we use for estimation over the uniformly selected subset of the population. We introduce the following novel estimator by applying regression adjustment to HT estimator.

**Definition 17** (Partial Observation Regression-Adjusted Horvitz-Thompson (RAHT) Estimator).
*Given a matrix of covariates $X \in \mathbb{R}^{n \times d}$, a fixed vector $\widehat{\mathbf{b}}$, and an integer $1 \leq m \leq n$, we introduce the* partial observation regression adjusted Horvitz-Thompson estimator *as the following. Estimation is based on a uniformly sampled set $S$ of size $m$ that we partition to two sets $Z^+$ and $Z^-$ by a random process (e.g., Gram-Schmidt walk). Then the ATE estimate is the following.*

$$\widehat{\tau} = \frac{1}{m}\left[\sum_{i \in Z^+} \frac{y_i^{(1)}}{\mathbb{P}\left[i \in Z^+\right]} - \sum_{i \in Z^-} \frac{y_i^{(0)}}{\mathbb{P}\left[i \in Z^-\right]}\right] - \frac{1}{m}\sum_{i \in S} \mathbf{x}_i^\top \widehat{\mathbf{b}}. \tag{16}$$

The following result indicates that the partial observation regression-adjusted Horvitz-Thompson estimator is a powerful tool for estimating ATE with partial observation.

**Lemma 18.** *Let $\mathbf{y}^{(1)}, \mathbf{y}^{(0)} \in \mathbb{R}^n$, $\widehat{\mathbf{b}} \in \mathbb{R}^d$ be a fixed/preassigned vector and $\mathbf{X} \in \mathbb{R}^{n \times d}$. Given an unbiased estimator $\widehat{\tau}_S$ for $\tau_S := \frac{1}{|S|}\sum_{i \in S} y_i^{(1)} - y_i^{(1)}$, and a uniformly sampled set $S$ of size $m$, the estimator $\widehat{\tau} := \widehat{\tau}_S - \frac{1}{m}\sum_{i \in S}(\mathbf{x}_i - \overline{\mathbf{x}})^\top \widehat{\mathbf{b}}$ is unbiased and its variance is*

$$\mathbb{E}\left[(\widehat{\tau} - \tau)^2\right] \leq \mathbb{E}_{S \sim \mathcal{D}}\left[\mathbb{E}\left[(\widehat{\tau}_S - \tau_S)^2\right]\right] + \frac{1}{mn}\left\|(\mathbf{X} - \overline{\mathbf{X}})^\top \widehat{\mathbf{b}} - (\mathbf{t} - \overline{\mathbf{t}})\right\|_2^2, \tag{17}$$

*where the second expectation on the right-hand side is over the randomness of $\widehat{\tau}_S$, and $\mathcal{D}$ denotes the distribution of choosing a uniformly random set of size $m$.*

*Proof.* We have $\mathbb{E}\left[(\widehat{\tau} - \tau)^2\right] = \mathbb{E}_{S \sim \mathcal{D}}\left[\mathbb{E}\left[(\widehat{\tau}_S - (\frac{1}{m}\sum_{i \in S}(\mathbf{x}_i - \overline{\mathbf{x}})^\top \widehat{\mathbf{b}}) - \tau)^2\right]\right]$. Moreover

$$(\widehat{\tau}_S - (\frac{1}{m}\sum_{i \in S}(\mathbf{x}_i - \overline{\mathbf{x}})^\top \widehat{\mathbf{b}}) - \tau)^2 = (\widehat{\tau}_S - \tau_S + \tau_S - (\frac{1}{m}\sum_{i \in S}(\mathbf{x}_i - \overline{\mathbf{x}})^\top \widehat{\mathbf{b}}) - \tau)^2$$

$$= (\widehat{\tau}_S - \tau_S)^2 + (\tau_S - (\frac{1}{m}\sum_{i \in S}(\mathbf{x}_i - \overline{\mathbf{x}})^\top \widehat{\mathbf{b}}) - \tau)^2 + 2(\widehat{\tau}_S - \tau_S)(\tau_S - (\frac{1}{m}\sum_{i \in S}(\mathbf{x}_i - \overline{\mathbf{x}})^\top \widehat{\mathbf{b}}) - \tau).$$

Since $\widehat{\tau}_S$ is an unbiased estimator of $\tau_S$,

$$\mathbb{E}_{S \sim \mathcal{D}}\left[\mathbb{E}\left[2(\widehat{\tau}_S - \tau_S)(\tau_S - (\frac{1}{m}\sum_{i \in S}(\mathbf{x}_i - \overline{\mathbf{x}})^\top \widehat{\mathbf{b}}) - \tau)\right]\right]$$

$$= \mathbb{E}_{S \sim \mathcal{D}}\left[\mathbb{E}\left[2(\widehat{\tau}_S - \tau_S)\right](\tau_S - (\frac{1}{m}\sum_{i \in S}(\mathbf{x}_i - \overline{\mathbf{x}})^\top \widehat{\mathbf{b}}) - \tau)\right] = 0.$$

Moreover

$$\mathbb{E}_{S \sim \mathcal{D}}\left[\mathbb{E}\left[(\tau_S - (\frac{1}{m}\sum_{i \in S}(\mathbf{x}_i - \overline{\mathbf{x}})^\top \widehat{\mathbf{b}}) - \tau)^2\right]\right] = \mathbb{E}_{S \sim \mathcal{D}}\left[(\tau_S - (\frac{1}{m}\sum_{i \in S}(\mathbf{x}_i - \overline{\mathbf{x}})^\top \widehat{\mathbf{b}}) - \tau)^2\right].$$

Since $\mathbb{E}_{S \sim \mathcal{D}}\left[\tau_S - (\frac{1}{m}\sum_{i \in S}(\mathbf{x}_i - \overline{\mathbf{x}})^\top \widehat{\mathbf{b}})\right] = \tau$, by Lemma 9,

$$\mathbb{E}_{S \sim \mathcal{D}}\left[(\tau_S - (\frac{1}{m}\sum_{i \in S}(\mathbf{x}_i - \overline{\mathbf{x}})^\top \widehat{\mathbf{b}}) - \tau)^2\right] = \frac{1 - m/n}{m(n-1)}\left\|(\overline{\mathbf{X}} - \mathbf{X})\widehat{\mathbf{b}} + (\mathbf{y}^{(1)} - \mathbf{y}^{(0)}) - \overline{\mathbf{t}}\right\|_2^2$$

$$\leq \frac{1}{mn}\left\|(\overline{\mathbf{X}} - \mathbf{X})\widehat{\mathbf{b}} - (\overline{\mathbf{t}} - \mathbf{t})\right\|_2^2.$$

$\square$

We next prove a varaince bound if the Gram-Schmidt walk design is applied for estimation over the uniformly selected subset.

**Theorem 19.** *Let $\widehat{\tau}_S$ be an estimate obtained by the Horvitz-Thompson estimator on the Gram-Schmidt walk design. Let $\widehat{\mathbf{b}}$ be a fixed/preassigned vector and $0 < \phi < 1$. Then, the partial observation regression-adjusted Horvitz-Thompson estimator with $m$ samples is an unbiased estimator with variance of*

$$\mathbb{E}\left[(\widehat{\tau} - \tau)^2\right] \leq \frac{1}{mn} \frac{1}{\phi} \|\mathbf{X}\mathbf{b}^* - \boldsymbol{\mu}\|_2^2 + \frac{1}{m^2} \frac{\zeta^2}{(1-\phi)} \|\mathbf{b}^*\|_2^2 + \frac{1}{mn} \left\|(\mathbf{X} - \overline{\mathbf{X}})^\top \widehat{\mathbf{b}} - (\mathbf{t} - \overline{\mathbf{t}})\right\|_2^2,$$

*where $\mathbf{b}^* = \arg\min_{\mathbf{b} \in \mathbb{R}^d} \left[\frac{1}{\phi} \|\mathbf{X}\mathbf{b} - \boldsymbol{\mu}\|_2^2 + \frac{\zeta^2}{(1-\phi)} \|\mathbf{b}\|_2^2\right]$.*

*Proof.* By Theorem 13 and Lemma 18,

$$\mathbb{E}\left[(\widehat{\tau} - \tau)^2\right] \leq \mathbb{E}_{S\sim\mathcal{D}}\left[\frac{1}{m^2} \min_{\mathbf{b}\in\mathbb{R}^n}\left[\frac{1}{\phi}\|\mathbf{X}_{S:}\mathbf{b} - \boldsymbol{\mu}_S\|_2^2 + \frac{\zeta^2}{1-\phi}\|\mathbf{b}\|_2^2\right]\right]$$
$$+ \frac{1}{mn}\left\|(\mathbf{X}-\overline{\mathbf{X}})^\top\widehat{\mathbf{b}} - (\mathbf{t}-\overline{\mathbf{t}})\right\|_2^2$$
$$\leq \frac{1}{m^2}\mathbb{E}_{S\sim\mathcal{D}}\left[\frac{1}{\phi}\|\mathbf{X}_{S:}\mathbf{b}^* - \boldsymbol{\mu}_S\|_2^2 + \frac{\zeta^2}{1-\phi}\|\mathbf{b}^*\|_2^2\right] + \frac{1}{mn}\left\|(\mathbf{X}-\overline{\mathbf{X}})^\top\widehat{\mathbf{b}} - (\mathbf{t}-\overline{\mathbf{t}})\right\|_2^2$$
$$= \frac{1}{m^2}\mathbb{E}_{S\sim\mathcal{D}}\left[\frac{1}{\phi}\|\mathbf{X}_{S:}\mathbf{b}^* - \boldsymbol{\mu}_S\|_2^2\right] + \frac{1}{m^2}\frac{\zeta^2}{1-\phi}\|\mathbf{b}^*\|_2^2$$
$$+ \frac{1}{mn}\left\|(\mathbf{X}-\overline{\mathbf{X}})^\top\widehat{\mathbf{b}} - (\mathbf{t}-\overline{\mathbf{t}})\right\|_2^2$$

Now note that

$$\|\mathbf{X}_{S:}\mathbf{b}^* - \boldsymbol{\mu}_S\|_2^2 = \sum_{i\in S}(\mathbf{x}_i^\top\mathbf{b}^* - \mu_i)^2.$$

The probability of any $i \in [n]$ appearing in a uniformly sampled set of size $m$ is

$$\frac{\binom{n-1}{m-1}}{\binom{n}{m}} = \frac{m}{n}.$$

Therefore

$$\mathbb{E}_{S\sim\mathcal{D}}\left[\|\mathbf{X}_{S:}\mathbf{b}^* - \boldsymbol{\mu}_S\|_2^2\right] = \frac{m}{n}\sum_{i\in[n]}(\mathbf{x}_i^\top\mathbf{b}^* - \mu_i)^2 = \frac{m}{n}\|\mathbf{X}\mathbf{b}^* - \boldsymbol{\mu}\|_2^2,$$

and the result follows by combining the above. $\qquad\square$

In Lemma 18 and Theorem 19, we have assumed that the vector $\widehat{\mathbf{b}}$ (which is used for adjusting the Horvitz-Thompson estimator) is given. For our final result, we combine our approach for regression on random vectors (with partial observation according to leverage score sampling) with partial observation RAHT to obtain a complete algorithm for estimation.

**Theorem 3** (ATE Estimation – Partial Observation). *For any $\epsilon, \delta \in (0,1]$, $\phi \in (0,1)$, and $m \in [n]$, there exists a randomized algorithm (Algorithm 4) that observes a potential outcome for $O(d\log(d/\delta)/\epsilon^2 + m)$ units and outputs an unbiased estimate $\widehat{\tau}$ of the ATE $\tau$, for which there is an event $\mathcal{E}$ with $\Pr(\mathcal{E}) \geq 1 - \delta$ (over the randomness of the algorithm), such that*

$$\mathbb{E}\left[(\widehat{\tau} - \tau)^2 | \mathcal{E}\right] \leq \frac{1}{mn}\frac{1}{\phi}\|\mathbf{X}\mathbf{b}^* - \boldsymbol{\mu}\|_2^2 + \frac{1}{m^2}\frac{\zeta^2}{(1-\phi)}\|\mathbf{b}^*\|_2^2 + \frac{100d\cdot\log(d/\delta)}{n^2\epsilon^2}\|\boldsymbol{\mu}\|_\infty^2$$
$$+ (1+\epsilon)\cdot\left(\frac{1}{m}\|\boldsymbol{\mu}\|_\infty^2 + \frac{1}{mn}\|(\mathbf{X}-\overline{\mathbf{X}})\widehat{\mathbf{b}} - (\mathbf{t}-\overline{\mathbf{t}})\|_2^2 + \frac{\lambda}{mn}\|\widehat{\mathbf{b}}\|_2^2\right),$$

*where $\zeta := \max_{i\in[n]}\|\mathbf{x}_i\|_2$, $\boldsymbol{\mu} := \mathbf{y}^{(0)} + \mathbf{y}^{(1)}$, $\lambda = \frac{6\cdot\log(d/\delta)}{\epsilon^2}\cdot\zeta^2$, $\overline{\mathbf{t}} \in \mathbb{R}^n$ is a vector where all the entries are equal to $\overline{t} = \frac{1}{n}\sum_{i=1}^n t_i$, and*

$$\mathbf{b}^* = \arg\min_{b\in\mathbb{R}^d}\left[\frac{m}{n}\|\mathbf{X}\mathbf{b} - \boldsymbol{\mu}\|_2^2 + \zeta^2\|\mathbf{b}\|_2^2\right], \widehat{\mathbf{b}} = \arg\min_{\mathbf{b}\in\mathbb{R}^d}\left[\|(\mathbf{X}-\overline{\mathbf{X}})\mathbf{b} - (\mathbf{t}-\overline{\mathbf{t}})\|_2^2 + \lambda\|\mathbf{b}\|_2^2\right].$$

**Algorithm 4:** Regression Adjusted Horvitz-Thompson Estimation

**1** **Input:** $\mathbf{X} \in \mathbb{R}^{n \times d}, \mathbf{y}^{(1)}, \mathbf{y}^{(0)} \in \mathbb{R}^n, m \in \mathbb{N}, 0 < \epsilon < 1, 0 < \phi < 1.$

**2** Set $\alpha = d/n$ and $\lambda = \frac{6 \cdot \log(d/\delta)}{\epsilon^2} \cdot \zeta^2.$

**3** Compute the $\lambda$-ridge leverage scores $\ell_i^{(\lambda)}$ of $\widetilde{\mathbf{X}} = \begin{bmatrix} (\mathbf{X} - \overline{\mathbf{X}}) & \mathbf{1} \end{bmatrix}$.

**4** Set $p_i = \min\{1, \max\{\alpha, 3 \cdot \ell_i \cdot \log(d/\delta)/\epsilon^2\}\}$.

**5** For all $i \in [n]$ define the random variable $q_i$ that is 1 with probability $p_i$, and is zero, otherwise. Let $S$ be the set of all indices $i$ with $q_i = 1$.

**6** Let $\mathbf{S} \in \mathbb{R}^{n \times n}$ be a random diagonal matrix with $\mathbf{S}_{ii} = 1/\sqrt{p_i}$ if $q_i = 1$, and $\mathbf{S}_{ii} = 0$, otherwise.

**7** Let $\mathbf{y} \in \mathbb{R}^n$ be a random vector where $y_i$ is independently equal to either $y_i^{(1)}$ or $-y_i^{(0)}$ with equal probability.

**8** Let $\widetilde{\mathbf{v}} = \arg\min_{\mathbf{b} \in \mathbb{R}^{d+1}} \left\| \mathbf{S}\widetilde{\mathbf{X}}\mathbf{b} - \mathbf{S}\mathbf{y} \right\|_2^2 + \lambda \left\| \mathbf{b}_{1:d} \right\|_2^2$, and $\widehat{\mathbf{v}} \in \mathbb{R}^d$ be a vector consisting of the first $d$ entries of $\widetilde{\mathbf{v}}$

**9** Let $T$ be a uniformly sampled set of size $m$ from $[n] \setminus S$.

**10** Obtain $\widehat{\tau}_{\overline{S}}$ by Horvitz-Thompson estimator and Gram-Schmidt walk design on $T$ with parameter $\phi$.          `// See Theorem 13`

**11** Let $\widehat{\tau}_S = 2 \sum_{i \in S} y_i$.

**12** Return $\frac{|\overline{S}|}{n} \cdot (\widehat{\tau}_{\overline{S}} - \frac{1}{m} \sum_{i \in T} (\mathbf{x}_i - \overline{\mathbf{x}})^\top \widehat{\mathbf{v}}) + \frac{|S|}{n} \cdot \widehat{\tau}_S$.

---

*Proof.* We first prove that the estimator is unbiased. By the law of total expectation, we have

$$\mathbb{E}[\widehat{\tau} - \tau] = \sum_{S,\mathbf{y}} \mathbb{E}[\widehat{\tau} - \tau | S] \cdot \mathbb{P}[S],$$

where the summation is over all sets $S$ that are selected from leverage score sampling. Note that given $S$ and $\mathbf{y}$, the solution vector $\widetilde{\mathbf{v}}$ is fixed. We denote this fixed solution vector with $\widetilde{\mathbf{v}}^{S,\mathbf{y}}$. Let $\widehat{\mathbf{v}}^{S,\mathbf{y}} \in \mathbb{R}^d$ be the vector obtained by taking the first $d$ entries of $\widetilde{\mathbf{v}}^{S,\mathbf{y}}$. Let $\overline{\mathbf{x}}^{\overline{S}}$ be the average row of $\mathbf{X}_{\overline{S}}$. Then we have

$$\mathbb{E}[\widehat{\tau} - \tau | S] = \sum_{\mathbf{y}} \mathbb{E}\left[ \frac{|\overline{S}|}{n} \cdot \left( \widehat{\tau}_{\overline{S}} - \frac{1}{m} \sum_{i \in T} (\mathbf{x}_i - \overline{\mathbf{x}}^{\overline{S}})^\top \widetilde{\mathbf{v}}^{S,\mathbf{y}} \right) - \frac{|\overline{S}|}{n} \cdot \tau_{\overline{S}} | S, \mathbf{y} \right] \cdot \mathbb{P}[\mathbf{y}]$$
$$+ \mathbb{E}\left[ \frac{|S|}{n} \cdot \widehat{\tau}_S - \frac{|S|}{n} \cdot \tau_S | S \right],$$

where the summation is over the assignments of the random vector $\mathbf{y}$, $\overline{\mathbf{x}}^{\overline{S}}$ is the average row over the set of rows indexed by $\overline{S}$, $T$ is a uniformly random subset of $\overline{S}$ of size $m$, $\widehat{\tau}_{\overline{S}}$ is estimated by using GSW design and Horvitz-Thompson estimator on set $T$, and $\widehat{\tau}_S$ is estimated by the Bernoulli design and Horvitz-Thompson estimator (the observation is compatible with vector $\mathbf{y}$). Trivially since $\widehat{\tau}_S$ is an unbiased estimator of $\tau_S$, the second term is equal to zero. Moreover since $\widehat{\tau}_{\overline{S}}$ is an unbiased estimator of $\tau_{\overline{S}}$, over the expectation, they cancel out and therefore,

$$\mathbb{E}[\widehat{\tau} - \tau | S] = \sum_{\mathbf{y}} -\frac{|\overline{S}|}{n} \cdot \mathbb{E}\left[ \left( \frac{1}{m} \sum_{i \in T} (\mathbf{x}_i - \overline{\mathbf{x}}^{\overline{S}})^\top \widetilde{\mathbf{v}}^{S,\mathbf{y}} \right) | S, \mathbf{y} \right] \cdot \mathbb{P}[\mathbf{y}].$$

Now since $\widetilde{\mathbf{v}}^{S,\mathbf{y}}$ is fixed given $S$ and $\mathbf{y}$, we have

$$\mathbb{E}[\widehat{\tau} - \tau | S] = \sum_{\mathbf{y}} -\frac{|\overline{S}|}{mn} \cdot \mathbb{E}\left[ \sum_{i \in T} (\mathbf{x}_i - \overline{\mathbf{x}}^{\overline{S}})^\top | S, \mathbf{y} \right] \widetilde{\mathbf{v}}^{S,\mathbf{y}} \cdot \mathbb{P}[\mathbf{y}].$$

We have

$$\mathbb{E}\left[ \sum_{i \in T} (\mathbf{x}_i - \overline{\mathbf{x}}^{\overline{S}})^\top | S, \mathbf{y} \right] = \left( \sum_{i \in \overline{S}} \frac{\binom{|\overline{S}|-1}{m-1}}{\binom{|\overline{S}|}{m}} \mathbf{x}_i^\top \right) - \frac{m}{|\overline{S}|} \left( \sum_{i \in \overline{S}} \mathbf{x}_i^\top \right)^\top = 0.$$

Therefore $\mathbb{E}\left[\widehat{\tau} - \tau | S\right] = 0$ and $\mathbb{E}\left[\widehat{\tau} - \tau\right] = 0$. Thus, the estimator is unbiased.

We define the event $\mathcal{E}$ to be the event where the regression error is within a factor of $(1 + \epsilon)$ of error and the number of nonzero entries in the matrix $\mathbf{S}$ is less than $100d \cdot \log(d/\delta)/\epsilon^2$. Note that

$$\mathbb{E}\left[(\widehat{\tau} - \tau)^2 | \mathcal{E}\right] = \sum_{S, y} \mathbb{E}\left[(\widehat{\tau} - \tau)^2 | S, \mathbf{y}\right] \cdot \mathbb{P}\left[S, \mathbf{y} | \mathcal{E}\right]$$

Note that condition to $S$ and $\mathbf{y}$,

$$\widehat{\tau} = \frac{|\overline{S}|}{n} \cdot \left(\widehat{\tau}_{\overline{S}} - \frac{1}{m}\sum_{i \in T}(\mathbf{x}_i - \overline{\mathbf{x}}^{\overline{S}})^\top \widetilde{\mathbf{v}}^{S, \mathbf{y}}\right) + \frac{|S|}{n} \cdot \widehat{\tau}_S,$$

where $\widehat{\tau}_{\overline{S}}$ and $\widehat{\tau}_S$ are unbiased estimators for $\tau_{\overline{S}}$ and $\tau_S$, respectively. Then since $\tau = \frac{|\overline{S}|}{n} \cdot \tau_{\overline{S}} + \frac{|S|}{n} \cdot \tau_S$, this is an unbiased estimator. Now we have

$$\mathbb{E}\left[(\widehat{\tau} - \tau)^2 | S, \mathbf{y}\right]$$

$$= \mathbb{E}\left[(\frac{|\overline{S}|}{n} \cdot \left(\widehat{\tau}_{\overline{S}} - \frac{1}{m}\sum_{i \in T}(\mathbf{x}_i - \overline{\mathbf{x}}^{\overline{S}})^\top \widetilde{\mathbf{v}}^{S, \mathbf{y}}\right) + \frac{|S|}{n} \cdot \widehat{\tau}_S - \frac{|\overline{S}|}{n} \cdot \tau_{\overline{S}} - \frac{|S|}{n} \cdot \tau_S)^2 | S, \mathbf{y}\right]$$

$$= \mathbb{E}\left[(\frac{|\overline{S}|}{n} \cdot \left(\widehat{\tau}_{\overline{S}} - \frac{1}{m}\sum_{i \in T}(\mathbf{x}_i - \overline{\mathbf{x}}^{\overline{S}})^\top \widetilde{\mathbf{v}}^{S, \mathbf{y}}\right) - \frac{|\overline{S}|}{n} \cdot \tau_{\overline{S}})^2 | S, \mathbf{y}\right]$$

$$+ \mathbb{E}\left[(\frac{|S|}{n} \cdot \widehat{\tau}_S - \frac{|S|}{n} \cdot \tau_S)^2 | S, \mathbf{y}\right]$$

$$+ 2\mathbb{E}\left[(\frac{|\overline{S}|}{n} \cdot \left(\widehat{\tau}_{\overline{S}} - \frac{1}{m}\sum_{i \in T}(\mathbf{x}_i - \overline{\mathbf{x}}^{\overline{S}})^\top \widetilde{\mathbf{v}}^{S, \mathbf{y}}\right) - \frac{|\overline{S}|}{n} \cdot \tau_{\overline{S}})(\frac{|S|}{n} \cdot \widehat{\tau}_S - \frac{|S|}{n} \cdot \tau_S) | S, \mathbf{y}\right] \qquad (18)$$

Note that condition to $S, \mathbf{y}$, $(\frac{|S|}{n} \cdot \widehat{\tau}_S - \frac{|S|}{n} \cdot \tau_S)$ is a fixed number. Therefore

$$\mathbb{E}\left[(\frac{|\overline{S}|}{n} \cdot \left(\widehat{\tau}_{\overline{S}} - \frac{1}{m}\sum_{i \in T}(\mathbf{x}_i - \overline{\mathbf{x}}^{\overline{S}})^\top \widetilde{\mathbf{v}}^{S, \mathbf{y}}\right) - \frac{|\overline{S}|}{n} \cdot \tau_{\overline{S}})(\frac{|S|}{n} \cdot \widehat{\tau}_S - \frac{|S|}{n} \cdot \tau_S) | S, \mathbf{y}\right]$$

$$= \mathbb{E}\left[(\frac{|\overline{S}|}{n} \cdot \left(\widehat{\tau}_{\overline{S}} - \frac{1}{m}\sum_{i \in T}(\mathbf{x}_i - \overline{\mathbf{x}}^{\overline{S}})^\top \widetilde{\mathbf{v}}^{S, \mathbf{y}}\right) - \frac{|\overline{S}|}{n} \cdot \tau_{\overline{S}}) | S, \mathbf{y}\right] \cdot \mathbb{E}\left[(\frac{|S|}{n} \cdot \widehat{\tau}_S - \frac{|S|}{n} \cdot \tau_S) | S, \mathbf{y}\right].$$

Since $\mathbb{E}\left[\widehat{\tau}_{\overline{S}}\right] = \tau_{\overline{S}}$ and $\frac{1}{|\overline{S}|}\sum_{i \in \overline{S}} \mathbf{x}_i = \overline{\mathbf{x}}^{\overline{S}}$, the above quantity is equal to zero. Now note that since we use the Horvitz-Thompson estimator to estimate $\widehat{\tau}$, we have

$$\mathbb{E}\left[(\frac{|S|}{n} \cdot \widehat{\tau}_S - \frac{|S|}{n} \cdot \tau_S)^2 | S, y\right] = \left(\frac{|S|}{n}\right)^2 \frac{1}{|S|^2} \|\boldsymbol{\mu}_S\|_2^2 \le \frac{100d \cdot \log(d/\delta)}{n^2 \cdot \epsilon^2} \|\boldsymbol{\mu}\|_\infty^2. \qquad (19)$$

Now note that $\widehat{\mathbf{b}}$ is fixed given $S, \mathbf{y}$. Let

$$\widetilde{\mathbf{b}}^{\overline{S}} = \arg\min_{\mathbf{b} \in \mathbb{R}^d} \left[\frac{1}{\phi} \|\mathbf{X}_{\overline{S}}\mathbf{b} - \boldsymbol{\mu}_{\overline{S}}\|_2^2 + \frac{\zeta^2}{(1 - \phi)} \|\mathbf{b}\|_2^2\right].$$

Then by Theorem 19,

$$\mathbb{E}\left[(\frac{|\overline{S}|}{n} \cdot \left(\widehat{\tau}_{\overline{S}} - \frac{1}{m}\sum_{i \in T}(\mathbf{x}_i - \overline{\mathbf{x}}^{\overline{S}})^\top \widetilde{\mathbf{v}}^{S, \mathbf{y}}\right) - \frac{|\overline{S}|}{n} \cdot \tau_{\overline{S}})^2 | S, \mathbf{y}\right]$$

$$\le \left(\frac{|\overline{S}|}{n}\right)^2 \left(\frac{1}{m \cdot |\overline{S}|} \frac{1}{\phi} \left\|\mathbf{X}_{\overline{S}}\widetilde{\mathbf{b}}^{\overline{S}} - \boldsymbol{\mu}_{\overline{S}}\right\|_2^2 + \frac{1}{m^2} \frac{\zeta^2}{(1 - \phi)} \left\|\widetilde{\mathbf{b}}^{\overline{S}}\right\|_2^2 + \frac{1}{m \cdot |\overline{S}|} \left\|(\overline{\mathbf{X}}^{\overline{S}} - \mathbf{X}_{\overline{S}})\widetilde{\mathbf{v}}^{S, \mathbf{y}} - (\overline{\mathbf{t}}^{\overline{S}} - \mathbf{t}_{\overline{S}})\right\|_2^2\right)$$

$$\le \frac{1}{mn} \frac{1}{\phi} \left\|\mathbf{X}_{\overline{S}}\widetilde{\mathbf{b}}^{\overline{S}} - \boldsymbol{\mu}_{\overline{S}}\right\|_2^2 + \frac{1}{m^2} \frac{\zeta^2}{(1 - \phi)} \left\|\widetilde{\mathbf{b}}^{\overline{S}}\right\|_2^2 + \frac{1}{mn} \left\|(\overline{\mathbf{X}}^{\overline{S}} - \mathbf{X}_{\overline{S}})\widetilde{\mathbf{v}}^{S, \mathbf{y}} - (\overline{\mathbf{t}}^{\overline{S}} - \mathbf{t}_{\overline{S}})\right\|_2^2$$

$$\le \frac{1}{mn} \frac{1}{\phi} \|\mathbf{X}_{\overline{S}}\mathbf{b}^* - \boldsymbol{\mu}_{\overline{S}}\|_2^2 + \frac{1}{m^2} \frac{\zeta^2}{(1 - \phi)} \|\mathbf{b}^*\|_2^2 + \frac{1}{mn} \left\|(\overline{\mathbf{X}}^{\overline{S}} - \mathbf{X}_{\overline{S}})\widetilde{\mathbf{v}}^{S, \mathbf{y}} - (\overline{\mathbf{t}}^{\overline{S}} - \mathbf{t}_{\overline{S}})\right\|_2^2$$

$$\le \frac{1}{mn} \frac{1}{\phi} \|\mathbf{X}\mathbf{b}^* - \boldsymbol{\mu}\|_2^2 + \frac{1}{m^2} \frac{\zeta^2}{(1 - \phi)} \|\mathbf{b}^*\|_2^2 + \frac{1}{mn} \left\|(\overline{\mathbf{X}}^{\overline{S}} - \mathbf{X}_{\overline{S}})\widetilde{\mathbf{v}}^{S, \mathbf{y}} - (\overline{\mathbf{t}}^{\overline{S}} - \mathbf{t}_{\overline{S}})\right\|_2^2 \qquad (20)$$

We now use the guarantees of leverage score sampling to bound the following

$$\left\| (\overline{\mathbf{X}}^{\overline{S}} - \mathbf{X}_{\overline{S}})\widetilde{\mathbf{v}}^{S,\mathbf{y}} - (\overline{\mathbf{t}}^{\overline{S}} - \mathbf{t}_{\overline{S}}) \right\|_2^2 .$$

The matrix $\overline{\mathbf{X}}^{\overline{S}}$ denotes a matrix with a size equal to $\mathbf{X}_{\overline{S}}$ such that all of its rows are equal to the average row of $\mathbf{X}_{\overline{S}}$. Similarly, $\overline{\mathbf{t}}^{\overline{S}}$ is a vector with a size equal to $\overline{\mathbf{t}}_{\overline{S}}$ such that all of its entries are equal to the average entry of $\overline{\mathbf{t}}_{\overline{S}}$. Similar to the proof of Theorem 12, let $\widetilde{\mathbf{X}}^{\overline{S}} := \left[ (\mathbf{X}_{\overline{S}} - \overline{\mathbf{X}}^{\overline{S}}) \quad \mathbf{1} \right]$, and

$$\mathbf{Z}^{\overline{S}} = \begin{bmatrix} (\mathbf{X}_{\overline{S}} - \overline{\mathbf{X}}^{\overline{S}}) \\ \sqrt{\lambda}\mathbf{I}_d \end{bmatrix}, \quad \widetilde{\mathbf{Z}}^{\overline{S}} = \begin{bmatrix} (\mathbf{X}_{\overline{S}} - \overline{\mathbf{X}}^{\overline{S}}) & \mathbf{1} \\ \sqrt{\lambda}\mathbf{I}_d & \mathbf{0} \end{bmatrix} .$$

Moreover, let $\widetilde{\mathbf{y}}^{(1,\overline{S})}, \widetilde{\mathbf{y}}^{(0,\overline{S})} \in \mathbb{R}^{|S|+d}$ be vectors that are equal to $\mathbf{y}_{\overline{S}}^{(1)}, \mathbf{y}_{\overline{S}}^{(0)}$, respectively, on their first $|\overline{S}|$ entries, and the rest of their entries are zero. Also let $\widetilde{\mathbf{t}}^{\overline{S}} = \widetilde{\mathbf{y}}^{(1,\overline{S})} - \widetilde{\mathbf{y}}^{(0,\overline{S})}$. Then by Theorem 1, the vector $\widetilde{\mathbf{v}}$ in Algorithm 4 satisfies

$$\mathbb{E}\left[ \left\| 2\widetilde{\mathbf{Z}}^{\overline{S}}\widetilde{\mathbf{v}} - \widetilde{\mathbf{t}}^{\overline{S}} \right\|_2^2 \Big| S \right] \leq \frac{d}{\alpha} \cdot (1+\epsilon) \left\| \boldsymbol{\mu}_{\overline{S}} \right\|_\infty^2 + (1+\epsilon) \min_{\mathbf{b} \in \mathbb{R}^{d+1}} \left\| \widetilde{\mathbf{Z}}^{\overline{S}}\mathbf{b} - \widetilde{\mathbf{t}}^{\overline{S}} \right\|_2^2$$

$$\leq \frac{d}{\alpha} \cdot (1+\epsilon) \left\| \boldsymbol{\mu} \right\|_\infty^2 + (1+\epsilon) \min_{\mathbf{b} \in \mathbb{R}^{d+1}} \left\| \widetilde{\mathbf{Z}}^{\overline{S}}\mathbf{b} - \widetilde{\mathbf{t}}^{\overline{S}} \right\|_2^2$$

Note that

$$\left\| \widetilde{\mathbf{Z}}^{\overline{S}}\mathbf{b} - \widetilde{\mathbf{t}}^{\overline{S}} \right\|_2^2 = \left\| \widetilde{\mathbf{X}}^{\overline{S}}\mathbf{b} - \mathbf{t}_{\overline{S}} \right\|_2^2 + \lambda \left\| \mathbf{b}_{1:d} \right\|_2^2 ,$$

and similar to the proof of Theorem 12, one can see that

$$\min_{\mathbf{b} \in \mathbb{R}^d} \left[ \left\| (\mathbf{X}_{\overline{S}} - \overline{\mathbf{X}}^{\overline{S}})\mathbf{b} - (\mathbf{t}_{\overline{S}} - \overline{\mathbf{t}}^{\overline{S}}) \right\|_2^2 + \lambda \left\| \mathbf{b} \right\|_2^2 \right] = \min_{\mathbf{b} \in \mathbb{R}^{d+1}} \left[ \left\| \widetilde{\mathbf{X}}^{\overline{S}}\mathbf{b} - \mathbf{t}_{\overline{S}} \right\|_2^2 + \lambda \left\| \mathbf{b}_{1:d} \right\|_2^2 \right]$$

Moreover

$$\left\| 2\widetilde{\mathbf{Z}}^{\overline{S}}\widetilde{\mathbf{v}} - \widetilde{\mathbf{t}}^{\overline{S}} \right\|_2^2 = \left\| (\mathbf{X}_{\overline{S}} - \overline{\mathbf{X}}^{\overline{S}})\widehat{\mathbf{v}} + \widetilde{\mathbf{v}}_{d+1} \cdot \mathbf{1} - \mathbf{t}_{\overline{S}} \right\|_2^2 + \lambda \left\| \widehat{\mathbf{v}} \right\|_2^2 .$$

Again similar to the proof of Theorem 12, one can show that

$$\left\| (\mathbf{X}_{\overline{S}} - \overline{\mathbf{X}}^{\overline{S}})\widehat{\mathbf{v}} - (\mathbf{t}_{\overline{S}} - \overline{\mathbf{t}}^{\overline{S}}) \right\|_2^2 \leq \left\| (\mathbf{X}_{\overline{S}} - \overline{\mathbf{X}}^{\overline{S}})\widehat{\mathbf{v}} + \widetilde{\mathbf{v}}_{d+1} \cdot \mathbf{1} - \mathbf{t}_{\overline{S}} \right\|_2^2$$

Combining the above and noting that $\lambda \cdot \left\| \widehat{\mathbf{v}} \right\|_2^2 \geq 0$, we have

$$\mathbb{E}\left[ \left\| (\mathbf{X}_{\overline{S}} - \overline{\mathbf{X}}^{\overline{S}})\widehat{\mathbf{v}} - (\mathbf{t}_{\overline{S}} - \overline{\mathbf{t}}^{\overline{S}}) \right\|_2^2 \right]$$

$$\leq \frac{d}{\alpha} \cdot (1+\epsilon) \left\| \boldsymbol{\mu} \right\|_\infty^2 + (1+\epsilon) \min_{\mathbf{b} \in \mathbb{R}^d} \left[ \left\| (\mathbf{X}_{\overline{S}} - \overline{\mathbf{X}}^{\overline{S}})\mathbf{b} - (\mathbf{t}_{\overline{S}} - \overline{\mathbf{t}}^{\overline{S}}) \right\|_2^2 + \lambda \left\| \mathbf{b} \right\|_2^2 \right]$$

$$\leq \frac{d}{\alpha} \cdot (1+\epsilon) \left\| \boldsymbol{\mu} \right\|_\infty^2 + (1+\epsilon) \left[ \left\| (\mathbf{X}_{\overline{S}} - \overline{\mathbf{X}}^{\overline{S}})\widehat{\mathbf{b}} - (\mathbf{t}_{\overline{S}} - \overline{\mathbf{t}}^{\overline{S}}) \right\|_2^2 + \lambda \left\| \widehat{\mathbf{b}} \right\|_2^2 \right]$$

Now since $(\mathbf{X}_{\overline{S}} - \overline{\mathbf{X}}^{\overline{S}})\widehat{\mathbf{b}} - (\mathbf{t}_{\overline{S}} - \overline{\mathbf{t}}^{\overline{S}})$ is orthogonal to the vector of all ones, we have

$$\left\| (\mathbf{X}_{\overline{S}} - \overline{\mathbf{X}}^{\overline{S}})\widehat{\mathbf{b}} - (\mathbf{t}_{\overline{S}} - \overline{\mathbf{t}}^{\overline{S}}) \right\|_2^2 \leq \left\| (\mathbf{X}_{\overline{S}} - \overline{\mathbf{X}}^{\overline{S}})\widehat{\mathbf{b}} - (\mathbf{t}_{\overline{S}} - \overline{\mathbf{t}}^{\overline{S}}) \right\|_2^2 + \left\| (\overline{\mathbf{X}}^{\overline{S}} - \overline{\mathbf{X}}_{\overline{S}})\widehat{\mathbf{b}} - (\overline{\mathbf{t}}^{\overline{S}} - \overline{\mathbf{t}}_{\overline{S}}) \right\|_2^2$$

$$= \left\| (\mathbf{X}_{\overline{S}} - \overline{\mathbf{X}}_{\overline{S}})\widehat{\mathbf{b}} - (\mathbf{t}_{\overline{S}} - \overline{\mathbf{t}}_{\overline{S}}) \right\|_2^2$$

$$\leq \left\| (\mathbf{X} - \overline{\mathbf{X}})\widehat{\mathbf{b}} - (\mathbf{t} - \overline{\mathbf{t}}) \right\|_2^2 .$$

Table 4: Variance of different methods for estimating ATE with few samples. $\boldsymbol{\mu} = \mathbf{y}^{(1)} + \mathbf{y}^{(0)}$, $\mathbf{t} = \mathbf{y}^{(1)} - \mathbf{y}^{(0)}$. $0 < \phi < 1$ is an input parameter. $0 < \epsilon < 1$ is an error parameter. $\zeta$ is the maximum $\ell_2$ norm over rows of $\mathbf{X}$. $\mathbf{b}^* = \arg\min_{b \in \mathbb{R}^d} \left[ \frac{1}{\phi} \|\mathbf{X}\mathbf{b} - \boldsymbol{\mu}\|_2^2 + \frac{1}{1-\phi}\zeta^2 \|\mathbf{b}\|_2^2 \right]$, and $\widehat{\mathbf{b}} = \arg\min_{\mathbf{b} \in \mathbb{R}^d} \left\| (\mathbf{X} - \overline{\mathbf{X}})\widehat{\mathbf{b}} - (\mathbf{t} - \overline{\mathbf{t}}) \right\|_2^2 + \lambda \left\| \widehat{\mathbf{b}} \right\|_2^2$, where $\lambda = \frac{6 \cdot \log(d/\delta)}{\epsilon^2} \cdot \zeta^2$.

| Method | Variance |
|---|---|
| Uniform Sampling + HT Estimator | $\dfrac{\|\boldsymbol{\mu}\|_2^2}{mn} + \dfrac{S_{\mathbf{t}}^2}{m}\left(1 - \dfrac{m}{n}\right)$ |
| GS Walk (trailed by uniform sampling) | $\dfrac{1}{m^2} \min_{b \in \mathbb{R}^d} \left[ \dfrac{m}{n} \cdot \dfrac{1}{\phi} \|\mathbf{X}\mathbf{b} - \boldsymbol{\mu}\|_2^2 + \dfrac{1}{1-\phi}\zeta^2 \|\mathbf{b}\|_2^2 \right] + \dfrac{S_{\mathbf{t}}^2}{m}\left(1 - \dfrac{m}{n}\right)$ |
| RAHT estimator + Regression on random vector + GS walk design | $\dfrac{1}{mn}\dfrac{1}{\phi}\|\mathbf{X}\mathbf{b}^* - \boldsymbol{\mu}\|_2^2 + \dfrac{1}{m^2}\dfrac{\zeta^2}{(1-\phi)}\|\mathbf{b}^*\|_2^2 + \dfrac{100d \cdot \log(d/\delta)}{n^2\epsilon^2}\|\boldsymbol{\mu}\|_\infty^2$ $+(1+\epsilon)\cdot\left(\dfrac{1}{m}\|\boldsymbol{\mu}\|_\infty^2 + \dfrac{1}{mn}\left\|(\mathbf{X} - \overline{\mathbf{X}})\widehat{\mathbf{b}} - (\mathbf{t} - \overline{\mathbf{t}})\right\|_2^2 + \dfrac{\lambda}{mn}\left\|\widehat{\mathbf{b}}\right\|_2^2\right)$ |

Therefore

$$\mathbb{E}\left[ \left\| (\mathbf{X}_{\overline{S}} - \overline{\mathbf{X}}^{\overline{S}})\widehat{\mathbf{v}} - (\mathbf{t}_{\overline{S}} - \overline{\mathbf{t}}^{\overline{S}}) \right\|_2^2 \right] \tag{21}$$

$$\leq \frac{d}{\alpha} \cdot (1+\epsilon) \|\boldsymbol{\mu}\|_\infty^2 + (1+\epsilon)\left[ \left\| (\mathbf{X} - \overline{\mathbf{X}})\widehat{\mathbf{b}} - (\mathbf{t} - \overline{\mathbf{t}}) \right\|_2^2 + \lambda \left\| \widehat{\mathbf{b}} \right\|_2^2 \right].$$

the results follows by picking $\alpha = d/n$ and combining (18), (19), (20), and (21).

$\square$

We compared the variance bounds obtained by our approach for ATE estimation with partial observation with other methods in Table 4.

# E  Detailed Description of Datasets and Extra Experiments

The costliest operation in our algorithms is to compute (ridge) leverage scores. This can be performed in input-sparsity time (i.e., $\tilde{O}(\text{nnz} + d^3)$, where nnz denotes the number of nonzero entries in the covariate matrix $\mathbf{X}$ [10]). Therefore, our algorithms are much more efficient than to the GSW design, which has a running time of $O(n^2 d)$ [18]. This significant difference in the running time is also illustrated in our experiments, as shown in Table 6.

Some details of the datasets we used in our experiments are presented in Table 5. Moreover, the comparison of our ATE approach with two other methods (Lin's [24] and difference in means) is presented in Table 7 and Figure 2.

Table 5: Detailed description of datasets.

| Dataset | Boston | IHDP | Twins | Synthetic ITE | Synthetic ATE |
|---|---|---|---|---|---|
| # samples | 506 | 747 | 32120 | 10000 | 10000 |
| # features | 13 | 25 | 50 | 50 | 50 |
| ATE | 0.0 | -4.016 | 0.0064 | 6.762 | 56.911 |
| $\|\boldsymbol{\mu}\|_\infty$ | 100 | 17.776 | 2.0 | 25.972 | 150.081 |
| $\frac{1}{\sqrt{n}}\|\boldsymbol{\mu}\|_2$ | 48.668 | 9.045 | 0.303 | 12.740 | 71.713 |
| $\frac{1}{\sqrt{n}}\|\mathbf{y}^{(1)}\|_2$ | 24.334 | 6.464 | 0.179 | 10.186 | 69.579 |
| $\frac{1}{\sqrt{n}}\|\mathbf{y}^{(0)}\|_2$ | 24.334 | 2.749 | 0.160 | 2.883 | 2.891 |
| $\frac{1}{\sqrt{n}}\min\|\mathbf{Xb} - \boldsymbol{\mu}\|_2$ | 9.831 | 0.846 | 0.276 | 3.865 | 0.200 |
| $\frac{1}{\sqrt{n}}\min\|\mathbf{Xb} - \mathbf{y}^{(1)}\|_2$ | 4.915 | 0.585 | 0.167 | 1.940 | 1.923 |
| $\frac{1}{\sqrt{n}}\min\|\mathbf{Xb} - \mathbf{y}^{(0)}\|_2$ | 4.915 | 0.407 | 0.149 | 1.930 | 1.912 |
| $\frac{1}{\sqrt{n}}\min\|\mathbf{Xb} - (\mathbf{y}^{(1)} - \mathbf{y}^{(0)})\|_2$ | 0.0 | 0.548 | 0.155 | 0.199 | 3.830 |

Table 6: Average time (in milliseconds) for estimating ATE over 1000 trials for different methods.

| Dataset | Uniform HT | GSW | Classic Reg Adj | Lev Score | 4 vecs | Ours |
|---|---|---|---|---|---|---|
| Boston | 0.1 | 9.5 | 0.4 | 1.1 | 2.1 | 1.4 |
| IHDP | 0.1 | 19.8 | 0.6 | 1.6 | 2.5 | 2.0 |
| Twins | 2.8 | 27719.0 | 13.7 | 43.9 | 54.8 | 53.2 |

Table 7: Results of ATE estimation. For each result, the first number is the average of $|\tau - \widehat{\tau}|$ over 1000 trials and the second number is the standard deviation of this quantity.

| Dataset | Uniform HT | GSW | Classic Reg Adj | Lin's [24] | difference in means | Lev Score | 4 vecs | Ours |
|---|---|---|---|---|---|---|---|---|
| Boston | 1.736 ±1.339 | 0.663 ±0.510 | 0.333 ±0.255 | 0.349 ±0.262 | 0.655 ±0.496 | 0.658 ±0.504 | 1.677 ±1.256 | 0.628 ±0.459 |
| IHDP | 0.272 ±0.206 | 0.042 ±0.031 | 0.012 ±0.009 | 0.013 ±0.010 | 0.049 ±0.036 | 0.536 ±0.050 | 0.264 ±0.203 | 0.040 ±0.030 |
| Twins | 1.351e−3 ±1.025e−3 | 1.231e−3 ±0.937e−3 | 1.201e−3 ±0.899e−3 | 86839.0 ±1091706.9 | 1.301e−3 ±0.981e−3 | 1.226e−3 ±0.911e−3 | 1.369e−3 ±1.015e−3 | 1.218e−3 ±0.936e−3 |

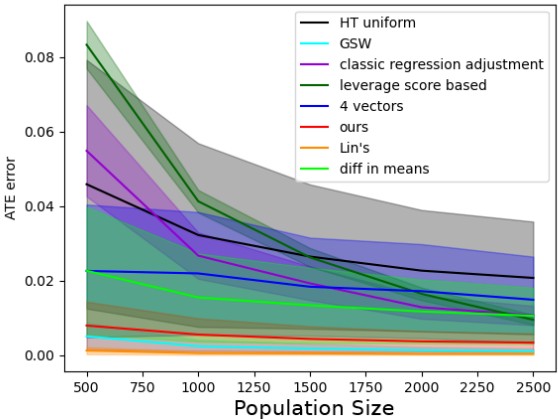

Figure 2: Results for synthetic ATE dataset. For each number of samples, estimation is performed for 1000 trials. Then the average of $|\hat{\tau} - \tau|/|\tau|$ over these trials is shown with solid lines and the shades around these lines denote the standard deviation.

# F    Gram-Schmidt Walk Design

Harshaw et al [18] introduced Algorithm 5 for balanced experimental design.

---

**Algorithm 5:** Non-Uniform Gram-Schmidt Walk

---

**1** **Input:** Matrix $\mathbf{X} \in \mathbb{R}^{n \times d}$, $0 < \phi < 1$, vector $\mathbf{p} \in \mathbb{R}^n$.

**2** Initialize the vector of fractional assignments $\mathbf{z}^{(1)} \leftarrow (2p_1 - 1, \ldots, 2p_n - 1)$.

**3** Initialize an index $j \leftarrow 1$.

**4** Select an initial pivot unit $k$ uniformly at random from $[n]$.

**5** Set $\zeta \leftarrow \max_{i \in [n]} \|\mathbf{X}_{i:}\|_2$.

**6** Let $\mathbf{B} \in \mathbb{R}^{n \times (n+d)}$ be a matrix such that $\mathbf{B}_{i:} := \left[ \begin{matrix} \sqrt{\phi} \cdot \mathbf{e}_i \\ \zeta^{-1}\sqrt{1 - \phi} \cdot \mathbf{X}_{i:} \end{matrix} \right]$, where $\mathbf{e}_i$ is the $i$'th

basis vector of dimension $n$.

**7** **while** $\mathbf{z}^{(j)} \notin \{-1, +1\}^n$ **do**

**8** $\quad$ Create the set $S \leftarrow \{i \in [n] : |z_i^{(j)}| < 1\}$.

**9** $\quad$ If $k \notin S$, select a new pivot $k$ from $S$ uniformly at random.

**10** $\quad$ Compute a step direction as $\mathbf{u}^{(j)} \leftarrow \arg\min_{\mathbf{u}}\{\|\mathbf{B}\mathbf{u}\|_2 : u_i = 0 \text{ for all } i \notin S, u_k = 1\}$.

**11** $\quad$ Let $\Delta = \{\delta \in \mathbb{R} : \mathbf{z}^{(j)} + \delta \cdot \mathbf{u}^{(j)} \in [-1, 1]^n\}$.

**12** $\quad$ Set $\delta^+ \leftarrow |\max \Delta|$ and $\delta^- \leftarrow |\min \Delta|$.

**13** $\quad$ Pick a random step size $\delta_j$ that is equal to $\delta^+$ with probability $\delta^-/(\delta^+ + \delta^-)$, and is equal to $-\delta^-$ with probability $\delta^+/(\delta^+ + \delta^-)$.

**14** $\quad$ Update fractional assignmenyt: $\mathbf{z}^{(j+1)} \leftarrow \mathbf{z}^{(j)} + \delta_j \mathbf{u}^{(j)}$.

**15** $\quad$ Increment the index $j \leftarrow j + 1$.

**16** Return the assigment vector $\mathbf{z}^{(j)} \in \{-1, +1\}^n$.

---

