# OpenReview forum: "Finite Population Regression Adjustment and Non-asymptotic Guarantees for Treatment Effect Estimation"
_NeurIPS.cc/2023/Conference — NeurIPS 2023 poster_

### Official Review · Reviewer_UdXn · 2023-07-04

**Soundness:** 3 good
**Presentation:** 3 good
**Contribution:** 3 good
**Rating:** 7
**Confidence:** 5

**Summary:**

In this paper, authors present regression adjusted estimators for estimating the average treatment effect under the Bernoulli design.
In particular, they show that by using the leverage scores and a ridge regression adjustment, favorable finite sample bounds on the (conditional) variance may be obtained.
This contributes nicely to a long-standing debate about the use of regression adjustment in randomized control trials.
Finally, simulations are performed which coroborate theoretical results.

**Strengths:**

The main strength of this paper is a finite sample analysis of regression adjusted estimators under the Bernoulli design.
There has been much debate about whether regression adjusted estimators are appropriate for estimates of the average treatment effect, [12, 20].
However, that entire literature has been studied under asymptotic analyses, which provides little insight into finite sample performance.

(Harshaw et al, 2019) have shown that under a sophisticated experimental design, favorable finite sample bounds on the variance may be obtained.
Moreover, in large samples, (using a well-chosen parameter), the Gram--Schmidt Walk design achieves the same asymptotic variance as Lin's regression adjustment.
This puts regression adjustment on the defensive and raises a natural question: can regression adjusted estimators achieve similar finite sample results?

This paper does a really remarkable job of answering this question in the affirmative, up to some minor details.
It will be of interest not only to the NeurIPS community, but also to the statistical causal inference community more broadly.

**Weaknesses:**

The paper has a few minor issues, but I believe they can all be addressed by the authors in a satisfactory way without too much work.

 ## Weakness 1 -- Conditional Analysis

The major weakness is a certain ambiguity that arises in a few of the theorems.
For example, Theorem 3 does not analyze the estimator in Algorithm 1 unconditionally.
Rather, it analyses it conditioned on a high probability event (i.e. "with probability $1 - \delta$, Algorithm 1 computes an unbiased estimator of the average treatment effect with variance...")
Although this is natural in computer science, this is very unnatural in this area of causal inference.
The reason is that we place larger value on the unconditional statistical inference; the validity / width of confidence intervals depends on the bias and variance, not a conditional bias or variance.
Moreover, Theorem 3 suggests that in order to get small variance, we should be setting $\delta$ very large; however, this will incur a large bias, which is not discussed in the Theorem nor in the paper.

In order for the causal inference community to fully appreciate these results (and they will!) the authors should derive bounds on the unconditional bias / variance.
This will have the additional benefit of giving experimenters advice on how to choose $\delta$ to minimize the MSE.
This critique applies to Theorem 4 as well.

## Weakness 2 -- Constant $C$

As it stands, the constant in Theorem 7 is presently unspecified.
This means that the Algorithm 1 cannot be run!
Authors should give a precise constant here.

## Weakness 3 -- ITE Results are Weak

Theorem 2 shows that the collection of ITEs can be estimated up to a constant error.
This is unsurprising because ITE estimation is considered impossible due to the fundamental problem of causal inference.
I find this estimator and these results not very exciting because it's not clear what a practitioner would do with such a guarantee -- not a single ITE is really known.
If authors are pressed on space, I recommend moving this to the appendix; otherwise, it's fine to leave in the paper.

## Weakness 4 -- Comparisons to GSW

I think that the comparison to (Harshaw et al 2019) is great and highlights the relevance of this work.
However, some of the comparisons are not relevant or could be improved.
Here is a list of improvements:

- (Line 137): Authors write that the variance of GSW is proportional to "XYZ". However, that's missing the important aspect of (Harshaw et al 2019), which is that these terms can be traded-off! So, I think you have to address (if only very briefly) that GSW-Design allows for trade-off of these terms, but that for $\phi = 1/2$ it is proportional to "XYZ".
- (Lines 33, 39, 141, etc): Authors write that GSW Design does not have an online analogue (which is currently true), but I don't think that this is how I would compare your estimator. The practical benefit of your regression adjustment estimator is that the data collection process is just so much simpler -- the Bernoulli design is decentralized, asynchronous, you name it! So although (Harshaw et al 2019) raised an online design as an open problem, it's really only interesting for covariate balancing. Being able to use such a simple design is beneficial much beyond online assignment. I think focusing on the practical simplicity of the Bernoulli design over the practical complexities of the GSW Design is much stronger argument in favor of your estimator, not the fact that Bernoulli design can be implemented online.
- (Line 264): Authors write that GSW Design "is a much slower algorithm". This doesn't seem true. (Harshaw et al 2019) give an implementation of GSW-Design that runs in $O(n^2 d)$, which would seem to also be required by for the linear system solve required to compute the ridge regression and scores.

## Minor Misc Comments

The following are minor miscellaneous comments

- (Line 126): "work" seems like the wrong word, a typo.
- (Line 242): The notation $\mathbf{P}_{\mathcal{P}}$ is used but not defined. I think it's fine to just remove the subscript.
- (Line 203): I think it is confusing / inappropriate to call Algorithm 1 "leverage score sampling" because that seems to suggest that the assignment to treatment and control (i.e. "sampling") is based on leverage scores. Can you call this something different? Perhaps "cross leverage score adjustment" or something?
- (Section 4.2): authors don't specify the value of $\phi$ used for the Gram--Schmidt Walk Design.
- It's a shame that most of the algorithms are in the appendix. If possible, I would recommend bringing them to the main body for camera ready version, perhaps at the cost of putting ITE estimation results in the appendix.

**Questions:**

- Can you address my comment on conditional versus unconditional analysis?
- Can you address my comment on the constant $C$ that is used?
- How does the Regression Adjusted Horvitz---Thompson estimator relate to double robust estimators?

**Limitations:**

yes

---

> ### Author Rebuttal · Authors · 2023-08-10
>
> We thank the reviewer for their positive assessment of our work and our contributions to finite sample analysis of regression adjustment. We thank them also for their detailed suggestions, especially those on the presentation, which we will address in a revised version of the paper.
>
> > The major weakness is a certain ambiguity that arises in a few of the theorems. For example, Theorem 3 does not analyze the estimator in Algorithm 1 unconditionally...
>
> Thank you for pointing this out. We also believe that stating unconditional bounds would improve the paper. We will add the following discussion to the paper.
>
> The only reason the $1-\delta$ probability appears is that leverage score sampling only gives a high probability guarantee for the regression error. Moreover our variance only depends on the regression error and the estimator is always unbiased, even if the leverage score sampling does not give the $(1+\epsilon)$ approximation guarantee on the regression. This “bad event” happens with probability at most $\delta$. Therefore it is enough to bound the regression error when the bad event happens.
>
> If A^T S^T S A is not a spectral approximation of A^T A (this can be checked since we have access to matrix A and only happens with probability at most $\delta$), we just set $b=0$ and in this case, $||Xb-\mu||\_2^2 + \lambda ||b||\_2^2=||\mu||\_2^2$.
> Otherwise let $\hat{X}$ be the matrix $X$ concatenated with identity and $\hat{\mu}$ be the vector $\mu$ concatenated with a zero vector and $b^*:=\arg\min||\hat{X}b-\hat{\mu}||\_2^2=\arg\min ||Xb-\mu||\_2^2+\lambda||b||\_2^2$. Let $\tilde{X}$ be the matrix $SX$ concatenated with identity and $\tilde{\mu}$ be the vector $S\mu$ concatenated with a zero vector and $\tilde{b}:=\arg\min || \tilde{X} b - \tilde{\mu} ||\_2^2$. Then since $\tilde{X}$ is a spectral approximation of $\hat{X}$,
> $$ || \hat{X} \tilde{b} - \hat{\mu} ||\_2 \leq || \hat{X} b^* - \hat{\mu} ||\_2 + || \hat{X} \tilde{b} - \hat{X} b^* ||\_2 \leq || \hat{X} b^* - \hat{\mu} ||\_2 + \frac{1}{1-\epsilon}|| \tilde{X} \tilde{b} - \tilde{X} b^* ||\_2 $$.
> Moreover
> $$||\tilde{X}\tilde{b}-\tilde{X}b^*||\_2\leq ||\tilde{X}\tilde{b}-\tilde{\mu}||\_2+||\tilde{X} b^*-\tilde{\mu}||\_2\leq 2\cdot||\tilde{X}b^*-\tilde{\mu}||\_2$$
> Now since leverage score sampling preserves the norms in expectation, we have $\mathbb{E}\_S||\tilde{X}b^*-\tilde{\mu}||\_2=||\hat{X}b^*-\hat{\mu}||\_2$.
> Then by the law of total expectation,
> $$\mathbb{E}[(\widehat{\tau} - \tau)^2] \leq (1-\delta) \left( \frac{32d}{n^2} \cdot || y^{(1)} - y^{(0)}||\_{\infty}^2 + \frac{8 (1+\frac{2}{1-\epsilon})}{n^2} \min\_{b} \left(||X b - \mu||\_2^2 + 100 \log(n/\delta) \cdot \zeta^2  \cdot ||b||\_2^2 \right) \right) + \delta \cdot || \mu ||\_2^2.$$
>
> > As it stands, the constant in Theorem 7 is presently unspecified. This means that the Algorithm 1 cannot be run! Authors should give a precise constant here.
>
> The analysis from other papers (see [9]) shows that it is enough to pick $c$ about 10. We add a note about this to the paper.
>
> > Theorem 2 shows that the collection of ITEs can be estimated up to a constant error. This is unsurprising because ITE estimation is considered impossible due to the fundamental problem of causal inference. I find this estimator and these results not very exciting because it's not clear what a practitioner would do with such a guarantee -- not a single ITE is really known.
>
> Although we agree that single ITEs cannot be estimated using our approach (and because of the fundamental problem of causal inference), we note that such bounds can be used to estimate other summary statistics. For example, a variance bound for ATE can be derived from our ITE bounds using the Cauchy-Schwarz inequality. In addition, bounds on median ITE or quantiles can be derived from our approach. Exploring these directions more is an interesting avenue for future work.
>
> > Authors write that the variance of GSW is proportional to "XYZ". However, that's missing the important aspect of (Harshaw et al 2019), which is that these terms can be traded-off! So, I think you have to address (if only very briefly) that GSW-Design allows for trade-off of these terms, but that for $\phi=1/2$ it is proportional to "XYZ".
>
> Thanks for this suggestion. We will add a note about the trade-off of GSW design to the paper.
>
> > Authors write that GSW Design does not have an online analogue (which is currently true), but I don't think that this is how I would compare your estimator. The practical benefit of your regression adjustment estimator is that the data collection process is just so much simpler -- the Bernoulli design is decentralized, asynchronous, you name it! ...
>
> Thank you for recognizing the generality and applicability of our results. We appreciate the suggestions and will add this discussion to the paper.
>
> > Authors write that GSW Design "is a much slower algorithm"...
>
> Thanks – we will add a discussion about the running time of our algorithm. Roughly, leverage scores can be computed in input-sparsity time. More specifically in $O(\text{nnz}(X) + d^{\omega})$ where $\text{nnz}(X)$ is the number of nonzero entries of matrix $X$ and $\omega$ is the matrix multiplication exponent. In the worst case, when $X$ is a dense matrix, this is $O(nd + d^{\omega})$. Note that this is the bulk of our computation since the running time of other steps of our algorithms is also $O(nd)$. In comparison, the running time of GSW design is $O(n^2 d)$. Note that since typically $n \gg d$, our running time is much better than GSW.
>
> In addition, since our operations involve matrix-matrix or matrix-vector multiplications, this is naturally parallelized in computers. In comparison, GSW design requires a random walk with n steps, which makes it less parallelizable. Therefore in practice, one might see even more dramatic differences in terms of running time.

---

> > ### Comment · Reviewer_UdXn · 2023-08-11
> > **response to authors**
> >
> > I thank the author for their thoughtful response to my review.
> > I have a few more questions for them.
> >
> > 1. Given the above work which provides unconditional bounds on the mean squared error in terms of $\delta$, can authors give practical suggestions for the value of $\delta$ to pick that might make this unconditional bound small? This seems crucial to the relevance of the results.
> >
> > 2. Authors write "The analysis from other papers (see [9]) shows that it is enough to pick about 10. We add a note about this to the paper.". However, this is confusing. The way the paper is written, it seems that there exists a particular constant and that the results only go through for this precise constant. If so, it seems imperative to find *exactly* this constant, rather than understanding it "approximately". Am I misunderstanding the results or their presentation?

---

> > > ### Author Response · Authors · 2023-08-13
> > > **Clarification regarding parameters of leverage score sampling**
> > >
> > > Thank you for the second round of comments!
> > >
> > > > Given the above work which provides unconditional bounds on the mean squared error in terms of $\delta$, can authors give practical suggestions for the value of $\delta$ to pick that might make this unconditional bound small? This seems crucial to the relevance of the results.
> > >
> > > Note that the dependence of the first term on $1/\delta$ is only logarithmic, while the second term depends on $\delta$. Therefore it is reasonable to consider $\delta=1/\text{poly}(n)$. For example, if we take $\delta=1/n^2$, then we get the following bound,
> > >
> > > $$\mathbb{E}[(\widehat{\tau} - \tau)^2] \leq (1-\frac{1}{n^2}) \left( \frac{32d}{n^2} \cdot || y^{(1)} - y^{(0)}||\_{\infty}^2 + \frac{8 (1+\frac{2}{1-\epsilon})}{n^2} \min\_{b} \left(||X b - \mu||\_2^2 + 300 \log(n) \cdot \zeta^2  \cdot ||b||\_2^2 \right) \right) + \frac{1}{n^2} \cdot || \mu ||\_2^2.$$
> > >
> > > > Authors write "The analysis from other papers (see [9]) shows that it is enough to pick about 10. We add a note about this to the paper.". However, this is confusing. The way the paper is written, it seems that there exists a particular constant and that the results only go through for this precise constant. If so, it seems imperative to find exactly this constant, rather than understanding it "approximately". Am I misunderstanding the results or their presentation?
> > >
> > > The constant $C$ is an oversampling parameter to have the guarantees of leverage score sampling. Note that taking any larger constant is also sufficient to have the guarantees as well, i.e., we need a sufficient oversampling. Therefore we do not need its exact value. An upper bound for $C$ essentially arises from matrix concentration inequalities used to prove the guarantees of the leverage score sampling. Then taking any oversampling parameter larger than or equal to this upper bound is enough to have the guarantees. In particular, Lemma 4 of [9] is as the following.
> > >
> > > Lemma 4 (Spectral Approximation via Leverage Score Sampling). Given an error parameter $0 < \epsilon < 1$, let $u$ be a vector of leverage score overestimates, i.e., $\tau_i(A) ≤ u_i$ for all $i$. Let $\alpha$ be a sampling rate parameter and let $c$ be a fixed positive constant. For each row, we define a sampling probability $p_i = \min\\{1, \alpha \cdot u_i c \log d \\}$. Furthermore, let $\text{Sample}(u, \alpha)$ denote a function which returns a random diagonal matrix $S$ with independently chosen entries $S_{ii} = \frac{1}{\sqrt{p_i}}$ with probability $p_i$ and $0$ otherwise. If we set $\alpha = \epsilon^{−2}$ , $S = \text{Sample}(u, \epsilon^{−2} )$ has at most $\sum_i \min\\{1, \alpha \cdot u_i c \log d \\} = O(\alpha c ||u||\_1 \log d)$ nonzero entries and $\frac{1}{\sqrt{1+\epsilon}}SA$ is a $\frac{1+\epsilon}{1-\epsilon}$-spectral approximation for $A$ with probability at least $1 − d^{−c/3}$.
> > >
> > > To have a probability of success of at least $1-\delta$, we need to take $c=\frac{3\log(1/\delta)}{\log d}$. Then an upper bound for $C$ is the coefficient of $\frac{\log(1/\delta)}{\log d}$ which is $3$. Therefore we can take any oversampling parameter greater than or equal to $3$, and the result still holds. Similar results apply to ridge leverage score sampling. We will add a note about this to the paper and will make sure the statement of theorems presents this clearly.

---

> > > > ### Comment · Reviewer_UdXn · 2023-08-14
> > > > **response**
> > > >
> > > > Thank you to the authors for replying with thoughtful responses.
> > > >
> > > > I think the discussion above, both (1) choosing $\delta$ and (2) clarifying the role of $C$ is very helpful. I think that the inclusion of both of these things in the paper will be critical to it being adopted and understood by the broader causal inference community. In fact, (if I may) I recommend simply setting $C = 3$ throughout the paper as this will be easier to understand.
> > > >
> > > > I agree with other reviewers that the presentation is somewhat confusing and lacking -- I hope that these comments can help to address these issues in presentation. Overall, I think the contribution of the work overcomes the issues in the presentation, so I maintain my score of accept.

---

### Official Review · Reviewer_ZgSr · 2023-07-04

**Soundness:** 2 fair
**Presentation:** 2 fair
**Contribution:** 2 fair
**Rating:** 5
**Confidence:** 3

**Summary:**

This paper focuses on estimation of individual and average treatment effects (ITE and ATE) with regression adjustment, which is combined with the method of ridge leverage score sampling in order to obtain the desirable variance bounds. The leading case is algorithm 1, which estimates ATE with leverage score sampling and cross adjustment. In this algorithm, for each observational unit, either treated or control unit is observed with equal probability. Then observed outcomes are adjusted via cross regression adjustments for which the regression coefficients are computed using ridge leverage score samples. The paper provides a number of theoretical results regarding the variance bounds and reports experimental results using both synthetic and real-data datasets.

**Strengths:**

- The main strength of the paper is that it provides finite sample variance bounds for estimating the sample mean, individual treatment effects, and the average treatment effect with regression adjustment.
- The research question addressed in the paper is of general nature and is important in many applications.

**Weaknesses:**

- To my reading of the paper, the most important issue is that it is unclear whether the advances in this paper are substantial relative to Harshaw et al [16]. It seems that theoretically, the variance bounds are comparable between the proposed methods in this paper and the Gram-Schmidt walk (GSW) design method of Harshaw et al [16]. Furthermore, the right panel of Figure 1 shows that GSW outperforms the current paper.
- The paper criticizes the GSW design, saying that it is not suitable for online experimental design (line 33 and lines 139-142). However, the proposed method is not immediately suitable for online settings. For example, it is unclear how to implement ridge leverage score sampling in a fully online fashion because the leverage score changes as more observations are available.
- The one advantage of the proposed method over GSW might be computational speed in view of table 6 in the supplement. However, there is no theoretical analysis in terms of computational aspects.

**Questions:**

- Is there any chance to obtain lower bounds for variance? It would complement the existing results on the upper bounds.
- Would it be possible to estimate the variance bounds?
- Line 192, is the transpose missing for the last $X$?
- Steps 5 and 6 in algorithm 1 may contain typos. I think these steps are meant for leverage score sampling but the stated steps are unclear. Please check them.
- Line 264. It might be helpful to provide a summary of running times in the main text.
- There is very little discussion on the experimental results in section 4. Probably this is due to the page limit. It would be helpful if there is a short summary of numerical findings in the main text.

**Limitations:**

One important limitation that is not mentioned in the paper is how to conduct statistical inference on ATE. This is typically done using asymptotic arguments. For example, Harshaw et al [16] provides asymptotic normality and suggests a confidence interval for ATE based on asymptotic normality. Although this paper focuses on finite sample variance bounds, it might be fair to mention the lack of inference methods.

---

> ### Author Rebuttal · Authors · 2023-08-10
>
> We thank the reviewer very much for their review and their questions/suggestions, which we address below.
>
> > To my reading of the paper, the most important issue is that it is unclear whether the advances in this paper are substantial relative to Harshaw et al [16]. It seems that theoretically, the variance bounds are comparable between the proposed methods in this paper and the Gram-Schmidt walk (GSW) design method of Harshaw et al [16]. Furthermore, the right panel of Figure 1 shows that GSW outperforms the current paper.
>
> Note that although the performance of GSW is similar to our approach, it is much slower than our algorithm. For example, on the twins dataset with about 32 thousand samples, the GSW design is about 500 times slower than our algorithm.
>
> > The paper criticizes the GSW design, saying that it is not suitable for online experimental design (line 33 and lines 139-142). However, the proposed method is not immediately suitable for online settings. For example, it is unclear how to implement ridge leverage score sampling in a fully online fashion because the leverage score changes as more observations are available.
>
> We would like to emphasize that we did not intend to criticize the GSW design. It certainly is an important work in the intersection of statistics and linear algebra. Our goal was only to address the differences between our approach and GSW design. We will make sure this is reflected in our writing.
>
> Regarding the online setting, we like to note that since we are using ridge leverage score sampling, as we pick the regularization (ridge) parameter, we make sure that the probability of selection of each row for any of the potential outcomes is at most 0.5. More specifically, note that in leverage score sampling, we only require to use upper bounds on leverage scores (not the exact leverage score), as stated in Theorem 7. Then by the constants we have picked in our algorithms/theorems we make sure that the probability of picking a unit for outcome 0 or outcome 1 is less than 0.5. Then essentially, the algorithm only needs to toss a fair coin for each unit arriving in the online setting. The only requirement here is to have the largest singular value of the covariate matrix to be bounded during the online experiment (because of our constant). We note that this is a very mild assumption. We will add a discussion about this. A similar argument applies to our subsampling (partial observation) approach as well. In addition, note that leverage scores can be computed in an online manner. See the following paper.
> Cohen, Michael B., Cameron Musco, and Jakub Pachocki. "Online row sampling." Theory of Computing 16, no. 1 (2020): 1-25.
>
> > The one advantage of the proposed method over GSW might be computational speed in view of table 6 in the supplement. However, there is no theoretical analysis in terms of computational aspects.
>
> Thanks – we will add a discussion about the running time of our algorithm. Roughly, leverage scores can be computed in input-sparsity time. More specifically in $O(\text{nnz}(X) + d^{\omega})$ where $\text{nnz}(X)$ is the number of nonzero entries of matrix $X$ and $\omega$ is the matrix multiplication exponent. In the worst case, when $X$ is a dense matrix, this is $O(nd + d^{\omega})$. Note that this is the bulk of our computation since the running time of other steps of our algorithms is also $O(nd)$. In comparison, the running time of GSW design is $O(n^2 d)$. Note that since typically $n \gg d$, our running time is much better than GSW.
>
> In addition, since our operations involve matrix-matrix or matrix-vector multiplications, this is naturally parallelized in computers. In comparison, GSW design requires a random walk with n steps, which makes it less parallelizable. Therefore in practice, one might see even more dramatic differences in terms of running time.
>
> > Is there any chance to obtain lower bounds for variance? It would complement the existing results on the upper bounds.
>
> We will add a discussion about terms in the error bounds, but one general way of looking at our bounds is that regression adjustment replaces the population variance (which is equivalent to using a vector of all zero for adjustment) with the error of best linear fit.
>
> We note that the Bernoulli design is min-max optimal (see Lemma 4.1 of [16]), and the following works provide hardness bounds on the discrepancy part of the GSW design. We will add a discussion regarding these to the paper.
> Zhang, Peng. "Hardness Results for Minimizing the Covariance of Randomly Signed Sum of Vectors." arXiv preprint arXiv:2211.14658 (2022).
> Charikar, Moses, Alantha Newman, and Aleksandar Nikolov. "Tight hardness results for minimizing discrepancy." In Proceedings of the twenty-second annual ACM-SIAM symposium on Discrete Algorithms, pp. 1607-1614. Society for Industrial and Applied Mathematics, 2011.
>
> >Would it be possible to estimate the variance bounds?
>
> Since the variance bounds depend on the regression error of vectors to which we do not have access (for example, $y^{(1)}+y^{(0)}$ since for each $i$, we only can observe either $y^{(1)}_i$ or $y^{(0)}_i$.), it seems non-obvious how to obtain meaningful variance estimates. However, this is an interesting suggestion to study for future work.

---

> > ### Comment · Reviewer_ZgSr · 2023-08-18
> > **Further comments**
> >
> > - I very much appreciate the authors' rebuttal; especially, I like the argument regarding a discussion about the running time of their algorithm vs. GSW.
> >
> > - It seems that the authors did not reply to all of my points (e.g., there seems no discussion of the lack of inference methods). I presume that the authors did not respond to each of them because some of them are rather straightforward.
> >
> > - I believe that the paper could be improved a lot after one or two rounds of careful rewriting and so I raised my rating by one more point from 4 to 5.

---

> > > ### Author Response · Authors · 2023-08-21
> > > **Inference**
> > >
> > > Since we are using the Bernoulli design, we believe that the central limit theorem (CLT) results hold in this case, allowing for inference as well. Both Lin [20] (Lemma 6 in Supplementary Material) and Freedman [*] (Theorem 1) have provided CLT results for regression adjustment. We believe these can be adapted to our results, but we defer this task to subsequent works. However, we will include a discussion mentioning this as future work and note that Harshaw et al. [16] have also provided such results.
> > >
> > > [*] Freedman, David A. "On Regression Adjustments in Experiments with Several Treatments." The Annals of Applied Statistics (2008): 176-196.

---

### Official Review · Reviewer_HGb9 · 2023-07-06

**Soundness:** 2 fair
**Presentation:** 1 poor
**Contribution:** 3 good
**Rating:** 4
**Confidence:** 3

**Summary:**

This paper explores the design and analysis of randomized experiments for treatment effect estimation, which is an important problem in causal inference. The goal of treatment effect estimation is to estimate the effect of a specific treatment on individual subjects or the average effect in the population, using only one of the two potential outcomes (either treatment or control) per subject. The objective is to obtain a precise estimator of the treatment effect, which is unbiased and has a smaller variance. However, it is often also desirable to minimize the number of subjects exposed to the experiment due to practical or ethical considerations. To address this, recent research has focused on leveraging linear algebraic techniques to design optimal experiments and estimate treatment effects.

The authors identify three main limitations in previous approaches: (1) inefficiency resulting from using the entire population in the experiment, (2) sub-optimal variance bound of the resulting estimator, and (3) inflexibility in experimental design, particularly in online settings. To overcome these challenges, the authors propose combining subsampling techniques from numerical linear algebra with classical statistical methods of regression adjustment. This integrated approach aims to obtain an unbiased estimator with a variance comparable to the optimal value, while also being suitable for online experiment settings.

The paper is organized as follows. In Section 2, the authors demonstrate the effectiveness of their proposed approach by considering four types of problems: (1) mean estimation of a vector, (2) individual treatment effect (ITE) estimation, (3) average treatment effect estimation (ATE) with full observation of both potential outcomes, and (4) ATE with partial observation, where only one potential outcome per subject is available. The primary focus is on problems (2) and (4), while problems (1) and (3) serve as precursors to facilitate the exposition of ideas. In Section 3, the authors describe the key techniques employed in their approach, providing a brief outline of the proofs for some theorems. Subsequently, they present experimental results from synthetic datasets (Section 4.1) and real-world datasets (Section 4.2) to validate the effectiveness of their proposed method.

**Strengths:**

The paper demonstrates several strengths that contribute to its overall quality and significance. Firstly, the problem of treatment effect estimation addressed in the paper holds substantial importance within the field of causal inference. The research questions explored are highly relevant and carry practical implications, underscoring the significance of the study.

Secondly, the proposed approach of integrating subsampling techniques rooted in randomized numerical linear algebra with regression adjustment is a well-founded and suitable strategy for tackling the problem at hand. Additionally, the authors' systematic approach in Section 2 effectively illuminates the challenges associated with addressing the problem and provides a rationale for the proposed approach.

Furthermore, the technical tools employed in the paper are well-established and widely utilized in the field. The authors' adept development of methods and analytical arguments reflects a strong grasp of these tools, bolstering the credibility of their findings.

In summary, the strengths of the paper lie in the significance of the addressed problem, the logical and appropriate nature of the proposed approach, and the authors' proficient utilization of standard technical tools. These strengths collectively enhance the quality and validity of the research presented in the paper.

**Weaknesses:**

While  the paper demonstrates strengths, there are several areas where improvements can be made to enhance clarity and strengthen the overall presentation. The weaknesses can be broadly categorized into two main areas: clarity and organization, as well as the soundness of the arguments.


I.  Concerns regarding clarity and organization

Firstly, the paper lacks clarity, potentially resulting from a lack of organization. To improve clarity, the authors should clearly identify the specific problems they are focusing on and indicate which parts of the paper address these issues. This could be achieved by adding a dedicated subsection or table in the Introduction that summarizes the problems addressed, the solutions proposed, and references to specific sections and theorems.

Additionally, restructuring sections would contribute to better organization and understanding. Merging Sections 2.1 and 2.2 into a single subsection titled "ITE Estimation" and merging Sections 2.3 and 2.4 under the title "ATE Estimation" would streamline the presentation and emphasize the two treatment effect estimation problems. Furthermore, Section 3 would benefit from reorganization and clearer subsection headings to provide explicit explanations of the specific techniques being described. For example, Section 3.1, currently titled "Random vector regression," contains a combination of an auxiliary theorem (Theorem 8) used to prove Theorem 2 (specifically, the display equation in line 113) related to ITE estimation, as well as a proof sketch of Theorem 3 in Section 2.3 related to ATE estimation. The rationale for grouping these two contents together is unclear, and it would be beneficial to clarify this and consider separating them into distinct subsections.

Moreover, the paper contains numerous theorem statements that, although likely mathematically correct, pose challenges in interpretation for several reasons. Many of these theorems reference algorithms that are either presented in later parts of the paper or not included in the main text at all. This creates logical inconsistencies and forces the reader to navigate back and forth, hindering comprehension. Furthermore, the authors do not provide explicit remarks on the optimal variance or the origin of the terms used in the error bounds, particularly, neither in close proximity to the theorem statements nor in a designated discussion section. Consequently, it is difficult to grasp the significance of the error bounds in terms of their semantic interpretation and their implications, including how tight or loose they are compared to optimal errors.


II. Concerns regarding soundness

Moreover, there are concerns regarding the soundness of the work, which may be further exacerbated by the clarity issues. The authors claim that their proposed method successfully addresses the aforementioned three challenges of previous approaches, but this claim lacks adequate support. To strengthen their arguments, the authors should provide explicit discussions or remarks, either following relevant technical propositions or in a dedicated discussion section. Specifically, they should elaborate on the following aspects in clear and comprehensible language:

1.  Clarify why previous approaches, such as [16], require the entire population in the experiment, while the proposed method does not.

2.  Discuss the optimal values or lower bounds of variance and how the authors' error bounds can be considered "comparable to the optimal."

3.  Explain how the proposed method allows for adaptation to the online experiment design setting.

Additionally, it would be beneficial to appropriately compare the proposed method to baseline approaches, such as classical regression adjustment estimators, Lin's interacted regression adjustment [20], or the crude difference-in-means estimator. Conducting such comparisons, particularly when the sample size is equal to the entire population, would provide a clearer understanding of the strengths and limitations (if any) of the proposed approach.


Addressing these weaknesses would significantly enhance the clarity, soundness, and overall quality of the paper.


**Questions:**

1.  I would like to request the authors to address the the concerns raised in the “Weaknesses” section.

2.  It appears that the error bound in Theorem 2 may be trivial and not particularly informative because $ d \| y^{(1)} + y^{(0)} \|_{\infty}^{2} \geq \| y^{(1)} + y^{(0)}\|_{2}^{2} \geq \| y^{(1)} - y^{(0)}\|_{2}^{2} = \| t \|_{2}^{2} $ assuming $\|  \| y^{(1)} - y^{(0)}\|_{2} \| \leq \|  \| y^{(1)} + y^{(0)}\|_{2} \| $. I wonder if the authors clarify the value of Theorem 2? t would be beneficial for the authors to clarify the significance and value of Theorem 2, particularly in relation to its practical implications.

3.  In Section 4.1, it would be helpful to provide information about the population size (n) for each dataset used in the experiments. Additionally, it would be valuable for the authors to evaluate the methods by quantifying the required sample size (i.e., the size of the sub-population) to achieve comparable performance to the method based on the entire population. This analysis would provide insights into the practical efficiency of the proposed method and its advantages in terms of minimizing the required sample size.

4.  In Section 4.2, it appears that the classic regression adjustment method outperforms all the other methods, including the authors' proposed method. In light of these results, it would be important for the authors to provide a clear justification for the value and effectiveness of their proposed method, especially in comparison to the superior performance of the classic regression adjustment method.

5.  In line 267, the authors mention that the classic regression adjustment technique is biased. While this is true, it is worth noting that there are at least two standard unbiased techniques available, namely the difference-in-means method and Lin's interacted regression adjustment [20]. It would be valuable for the authors to include these unbiased techniques in their experiments to provide a comprehensive comparison and analysis of the proposed method against these established approaches.

Addressing these concerns would greatly enhance the understanding, validity, and practical applicability of the research presented in the paper.

**Limitations:**

It would be valuable for the authors to provide a short discussion on the technical limitations of their methods and offer insights into the potential adverse consequences that may arise when applying these approaches in real-world settings, specifically in experiment design and treatment effect estimation. Nonetheless, it is important to note that given the paper's primary focus on theoretical aspects, a comprehensive and extensive examination of these limitations may not be deemed critical.

---

> ### Author Rebuttal · Authors · 2023-08-10
>
> We thank the reviewer for their in depth review and valuable suggestions, especially those regarding increasing clarity of the writing and paper organization, which we will incorporate in a revised version.
>
> > To improve clarity, the authors should clearly identify the specific problems they are focusing on and indicate which parts of the paper address these issues.
>
> We have stated the specific problems in Section 1.2. We consider 4 problems: 1) mean estimation; 2) ITE; 3) ATE with full observation (i.e., for each unit, exactly one of the outcomes are observed); 4) ATE with partial observation (i.e., for each unit, at most one of the outcomes are observed).
>
> > Furthermore, the authors do not provide explicit remarks on the optimal variance or the origin of the terms used in the error bounds…
>
> We will add a discussion about terms in error bounds. One general way of looking at our bounds is that regression adjustment replaces the population variance with the error of best linear fit.
>
> We note that the Bernoulli design is min-max optimal (see Lemma 4.1 of [16]).
>
> > Clarify why previous approaches, such as [16], require the entire population in the experiment, while the proposed method does not.
>
> For [16], this is essentially because of the random walk approach and is inherent to the algorithm. Generally, for previous methods in the literature, including [16], one can uniformly subsample the population and then design an experiment on the subsample and make an estimation, but this leads to worse bounds compared to ours — see the discussion in Section 2.4. Essentially the main reason that our subsampling approach works is the guarantees that leverage score sampling provides for minimizing linear regression error, and the fact that we show the variance of the population can be replaced with the error of the best linear fit.
>
> > Explain how the proposed method allows for adaptation to the online experiment design setting.
>
> Note that in leverage score sampling, we only require upper bounds on leverage scores (not the exact leverage score), as stated in Theorem 7. Then by the constants we have picked in our algorithms/theorems we make sure that the probability of picking a unit for outcome 0 or outcome 1 is less than 0.5. Then essentially, the algorithm only needs to toss a fair coin for each unit arriving in the online setting. The only requirement here is to have the largest singular value of the covariate matrix to be bounded during the online experiment (because of our constant). We note that this is a very mild assumption. We will add a discussion about this. A similar argument applies to our subsampling (partial observation) approach as well.
>
> > It appears that the error bound in Theorem 2 may be trivial and not particularly informative because $d||y^{(1)}+y^{(0)}||\_{\infty}^2\geq||y^{(1)}+y^{(0)}||\_2^2\geq||y^{(1)}-y^{(0)}||\_2^2=||t||\_2^2$ assuming $||y^{(1)}-y^{(0)}||\_2^2\leq||y^{(1)}+y^{(0)}||\_2^2 $. I wonder if the authors clarify the value of Theorem 2?
>
> Note that $d$ is the number of covariates/features. $y^{(1)}$ and $y^{(0)}$ are n-dimensional vectors, where n is the population size. Therefore the first inequality you mentioned does not hold. Typically, we have $d \ll n$. Thus, $d||y^{(1)}+y^{(0)}||\_{\infty}^2$ is much smaller than $||y^{(1)}+y^{(0)}||\_2^2$. As an example, pick $\alpha=\frac{d}{\sqrt{n}}$. In this case, the number of observed units will be $O(d\log(d/\delta)/\epsilon^2+d\sqrt{n})$ and the mentioned term in the variance bound is $\sqrt{n}||y^{(1)}+y^{(0)}||\_{\infty}^2$ which could be much smaller than $||y^{(1)}+y^{(0)}||\_{2}^2$ when n is large.
>
> In addition, note that $y^{(1)}+y^{(0)}$ is indeed the vector $\mu$ which is the vector appearing in Horvitz-Thompson estimator and Gram-Schmidt walk.
>
> In practice, the size of the population is usually much larger than the number of features/covariates, and we believe in such settings, our bound is very useful.
>
> > In Section 4.2, it appears that the classic regression adjustment method outperforms all the other methods, including the authors' proposed method...
>
> We would like to note that as opposed to classic regression adjustment, we present non-asymptotic bounds for our approach. To our knowledge, the guarantees on classic regression adjustment are asymptotic. Moreover, as you mentioned, classic regression adjustment is biased. Finally, note that it performs very poorly on our synthetic dataset. In general, we believe that no single approach performs well on all datasets and all settings (similar to the no-free-lunch theorem in machine learning). Our approach is most effective in scenarios where the regression errors are small, i.e., the treatment effect is linearly predicted by covariates well.
>
> > In line 267, the authors mention that the classic regression adjustment technique is biased. While this is true, it is worth noting that there are at least two standard unbiased techniques available, namely the difference-in-means method and Lin's interacted regression adjustment [20]...
>
> Thank you for pointing this out. We have added experiments for comparing the performance of the unbiased versions of classic regression adjustment to the pdf accompanying our response. Interestingly, Lin’s approach works very well on all except one of the datasets. However, on the twins dataset, it performs very poorly (but this might be due to numerical stability issues). Note that Lin’s approach is only asymptotically unbiased, while the focus of our work is to provide Non-asymptotic Guarantees and unbiasedness.

---

> > ### Comment · Reviewer_HGb9 · 2023-08-16
> >
> > I appreciate the authors' efforts in providing a comprehensive rebuttal and clarifications to address the concerns raised by both myself and other reviewers.  As a result of this process, I have gained a deeper understanding of the paper's contributions, notably aided by the insightful inquiries posed by Reviewer UdXn and the subsequent elucidations provided by the authors. It appears that the paper holds promise as a valuable addition to the conference. However, I maintain that substantial rewriting is necessary for the paper to effectively meet the required standards and convey its messages to the audience. Consequently, my assessment has shifted slightly towards the positive side, while still retaining some reservations, should a definitive stance be required.

---

### Official Review · Reviewer_zpw6 · 2023-07-07

**Soundness:** 3 good
**Presentation:** 2 fair
**Contribution:** 2 fair
**Rating:** 4
**Confidence:** 3

**Summary:**

The paper addresses the problem of ATE and ITE estimation in the presence of covariates. In particular, the authors provide finite-sample variance bounds for regression adjustment method-based estimators and novel variants thereof. The core of the methodology is using leverage scores, a randomized numerical linear algebra technique. This approach has been previously employed in selecting a sample of the population on which to perform an experiment.


**Strengths:**

To the best of my knowledge, using leverage scores for regression adjustment and offering finite-sample variance guarantees are novel contributions to the ATE and ITE estimation literature. The paper is strong in its technical aspects, as many of the theoretical contributions are robustly presented and accompanied by thorough proofs.

**Weaknesses:**

The idea of the paper is overall good, but the execution is poor. The main issue is that the paper very technically dense and is largely unreadable without the appendix. It reads like a stream of (highly technical) consciousness rather than a scientific report. It also looks like the authors sacrificed intuition in favor of technical details of questionable relevance.

Other notes:

* The algorithm links are broken (as they are in the appendix, but that is not mentioned anywhere)
* The algorithms referenced from the appendix contain quantities not yet defined which makes the paper difficult to read
* Many details in experimental section are relegated to the appendix so it's difficult to assess its soundness.

In my opinion, while the paper has potential to be a valuable addition to the conference, it would require substantial rewriting in order to meet the required standards and expectations.


**Questions:**

* How is the random vector in Key Challenge 2 built given that we don't observe both $\textbf{y}^{(0)}$ and $\textbf{y}^{(1)}$?
* It looks like both potential outcomes are observed in the experiments, is that correct? Again, the experimental section is difficult to read.
* It appears that, in several experiments, the performance of the proposed method is comparable to that of the leverage scores method. In light of this observation, what is the significance or unique contribution of this approach?

**Limitations:**

I was unable to find a discussion by the authors about the limitations of this work.

---

> ### Author Rebuttal · Authors · 2023-08-10
>
> We thank the reviewer for positive assessment of our technical contributions and for recognizing the novelty of applying leverage scores to regression adjustment. We agree with the reviewer that the paper can be made more readable and appreciate the reviewer’s suggestions towards doing so, which we will address in the revised version.
>
>
>
> > How is the random vector in Key Challenge 2 built given that we don't observe both $y^{(0)}$ and $y^{(1)}$?
>
> To construct the random vector v, we only need to observe one of the outcomes $y^{(0)}$ or $y^{(1)}$ for each unit. In particular, for each unit $i$, we toss a fair coin. If it lands on heads, we observe $y^{(0)}_i$ and set $v_i=-2y^{(0)}_i$. If it lands on tails, we observe $y^{(1)}_i$ and set $v_i=2y^{(1)}_i$.
>
> > It looks like both potential outcomes are observed in the experiments, is that correct?
>
> No, for each unit, we observe at most one of the potential outcomes. The only method that uses both observations is the baseline for ITE which is used for comparison.
>
> > It appears that, in several experiments, the performance of the proposed method is comparable to that of the leverage scores method. In light of this observation, what is the significance or unique contribution of this approach?
>
>
> Our goal was to develop a treatment effect estimation approach that is fast and unbiased, works on finite-population, gives non-asymptotic guarantees, allows only the use of a subset of the population (instead of the whole population), and can be used in different settings (such as online experimental design).
>
> As we mention in the introduction, each previous method has some shortcomings. So the goal of our method is to address all of these shortcomings simultaneously. For example, the GSW-design is very slow, as our experiments (presented in the appendix) suggest — on the twins dataset with about 32 thousand samples, the GSW design is about 500 times slower than our algorithm. Another example is that classic regression adjustment and using leverage scores to learn two different vectors are biased approaches.

---

### Author Rebuttal · Authors · 2023-08-10

We would like to thank all reviewers for the positive assessment of our contributions to finite-population treatment effect estimation and its non-asymptotic analysis. We appreciate the comments on the presentation and organization of the paper. We believe these would make the paper stronger, and we will address these comments in the next version of the paper.

Below we address the comments and questions of reviewers individually. The attached PDF contains some extra experiments requested by the reviewers. More specifically, it contains the comparison with the difference-in-means method and Lin's interacted regression adjustment approach.

---

> ### Author Response · Authors · 2023-08-21
> **Presentation**
>
> We would like to thank all the reviewers for their in-depth review, engagement, and discussion. Their comments on the writing and presentation will certainly help us to reach a broader audience. We would like to reassure the reviewers that we will do our best to improve the quality of the writing and presentation of the paper in the next version, incorporating their feedback.

---

### Decision · Program_Chairs · 2023-09-21

**Decision:**

Accept (poster)

**Comment:**

This paper presents the first finite sample analysis for regression-adjusted estimators under the Bernoulli design in the finite population causal inference setting. The reviewers unanimously agree that these contributions are significant and substantive. The quality of the feedback has been exceptional, I won't repeat them in detail in my meta-review and instead highlight the most important concern raised by the reviewing team: presentation.

Even with the challenges of writing for an interdisciplinary community (causal inference, TCS, ML), the current submission falls short of providing a clear exposition by most measures. The reviewing team represented experienced and mature researchers with diverse backgrounds, and it is concerning that all of them initially found the manuscript inscrutable. They have provided thoughtful feedback on how to improve the presentation, and I strongly urge the authors to carefully follow them to maximize the impact of their work.